# Rethinking the Flow-Based Gradual Domain Adaptation: A Semi-Dual Optimal Transport Perspective

**Zhichao Chen** [1]  **Zhan Zhuang** [2]  **Yunfei Teng** [1]  **Hao Wang** [3]  **Fangyikang Wang** [4]  **Zhengnan Li** [5]  **Tianqiao Liu** [6]
**Haoxuan Li** [✉ 1]  **Zhouchen Lin** [✉ 1]

## Abstract

Gradual domain adaptation (GDA) aims to mitigate domain shift by progressively adapting models from the source domain to the target domain via intermediate domains. However, real intermediate domains are often unavailable or ineffective, necessitating the synthesis of intermediate samples. Flow-based models have recently been used for this purpose by interpolating between source and target distributions. Notably, their training typically relies on sample-based log-likelihood estimation, which can discard useful information and thus degrade GDA performance. The key to addressing this limitation is constructing the intermediate domains via samples directly. To this end, we propose an Entropy-regularized Semi-dual Unbalanced Optimal Transport (E-SUOT) framework to construct intermediate domains. Specifically, we reformulate flow-based GDA as a Lagrangian dual problem and derive an equivalent semi-dual objective that circumvents the need for likelihood estimation. However, the dual problem leads to an unstable min–max training procedure. To alleviate this issue, we further introduce the entropy regularization to convert it into a more stable sequential optimization procedure. Based on this, we propose a novel GDA training framework and provide theoretical analysis in terms of stability and generalization. Finally, extensive experiments are conducted to demonstrate the efficacy of the E-SUOT framework.

---

[1]State Key Lab of General AI, School of Intelligence Science and Technology, Peking University [2]City University of Hong Kong [3]Zhejiang University [4]Mohamed bin Zayed University of Artificial Intelligence [5]The Chinese University of Hong Kong, Shenzhen [6]TAL Education Group. Correspondence to: Haoxuan Li <hxli@stu.pku.edu.cn>, Zhouchen Lin <zlin@pku.edu.cn>.

*Proceedings of the 43$^{rd}$ International Conference on Machine Learning*, Seoul, South Korea. PMLR 306, 2026. Copyright 2026 by the author(s).

## 1. Introduction

Unsupervised Domain Adaptation (UDA) (Courty et al., 2014; 2017a; Long et al., 2015; Pan & Yang, 2010; Tzeng et al., 2017; Yang et al., 2026; Zhang et al., 2023; 2024; 2025), which transfers knowledge from a well-trained source domain to a related yet unlabeled target domain, is of great importance across fundamental application areas. For example, in recommender systems (Chen et al., 2024a;b; Li et al., 2024a;b; Liu et al., 2023; 2025; Zheng et al., 2025a;b; 2026; 2024), a cold-start user has no interaction history with new items, so domain adaptation helps transfer user and item knowledge from an existing system to improve recommendations. Similar scenarios occur in machine translation, where a model trained on high-resource language pairs like English-French can be adapted to translate between English and low-resource languages with limited parallel data (Cheng et al., 2025; Gazdieva et al., 2023; 2025). These scenarios highlight the importance of conducting UDA to bridge domain gaps and ensure reliable performance in real-world applications.

Despite these methodological advances, directly performing UDA can be brittle when the source–target shift is substantial or class overlap is weak. In such cases, one-shot alignment often degrades discriminability and amplifies pseudo-label errors during self-training. This challenge motivates a transition from the traditional UDA setting to the Gradual Domain Adaptation (GDA) setting (He et al., 2024), where adaptation proceeds through a sequence of intermediate distributions that progressively bridge the domain gap. A key aspect of generating intermediate domains in GDA is to interpolate between the source and target domains. Various methods have been proposed to construct such intermediate domains, among which flow-based approaches (Kobyzev et al., 2020; Papamakarios et al., 2021; Yan et al., 2026; Zeng et al., 2026) have attracted increasing attention, primarily due to their property of preserving probability density along the transformation path, thereby enabling consistent and stable probability densities without distortion or loss of information. To drive the samples from the source domain towards those of the target domain, it is necessary to design an appropriate driving force, typically

derived from a discrepancy metric. Among these metrics, $f$-divergence (Sason & Verdú, 2016) is most widely used due to its computational efficiency, empirical effectiveness, and principled formulation within the framework of geometry for probability distributions (Amari, 2016).

Despite the success of flow-based approaches in GDA (Sagawa & Hino, 2025; Zeng et al., 2026; Zhuang et al., 2024), we argue that directly applying standard flow-based models leads to suboptimal performance. Specifically, existing flow-based frameworks utilizing $f$-divergence often require the explicit estimation of target domain probability density functions (PDFs) from available target samples (Ambrosio et al., 2005; Santambrogio, 2017; Vincent, 2011), whereas the subsequent GDA process relies on these estimated PDFs to drive the source-to-target transfer. For example, Zhuang et al. (2024) estimate the target domain PDF in the score function form and generate intermediate domains via Langevin dynamics. Consequently, the quality of the intermediate domain heavily depends on the accuracy of the estimated PDF; if this estimation is inaccurate, the performance of downstream task may suffer significantly.

To address these limitations, we propose a novel flow-based GDA framework termed "Entropy-regularized Semi-dual Unbalanced Optimal Transport" (E-SUOT), which leverages the semi-dual formulation of gradient flows. Rather than explicitly estimating PDFs, we recast flow evolution as an optimization problem that combines an $f$-divergence term with a Wasserstein distance regularization term, enabling sample transport toward the target domain without reliance on PDF estimation. However, as the semi-dual reformulation inherently leads to an adversarial training paradigm that can compromise stability and performance, we introduce entropy regularization to the objective to guarantee the stability of the training process. Based on this, we summarize the algorithm for E-SUOT-based intermediate domain generation, and prove the convergence of our E-SUOT framework. Extensive experiments on the GDA and UDA tasks validate that E-SUOT achieves superior performance compared with existing methods.

**Contributions.** We summarize our contributions as follows:

1. We develop a semi-dual formulation for intermediate domain generation in flow-based GDA, which eliminates the need for explicit estimation of the target domain PDF.
2. We introduce an entropy regularization term to address the unstable issue inherent in the semi-dual formulation, resulting in the novel and stable E-SUOT framework.
3. We conducted various experiments to demonstrate the superiority of the E-SUOT framework compared to prevalent approaches.

**Conflicts of Interest Disclosure.** No potential conflict of interest was reported by the authors.

## 2. Preliminaries

**Preliminary Note.** In this manuscript, we focus on analyzing the flow-based GDA *per se*, rather than establishing that GDA is superior to UDA. We inherit the standard assumptions commonly adopted in GDA and do not discuss when these assumptions hold in practice. A related unbalanced optimal transport (UOT) formulation will be derived as a direct consequence of standard definitions in the flow-based GDA setting; we use it as an analytical characterization, not as an additional assumption, and we do not study the inherent superiority of UOT compared with the vanilla optimal transport (OT) formulation.

### 2.1. Settings and Notations

In GDA, we consider a labeled source domain, $T - 1$ unlabeled intermediate domains, and an unlabeled target domain. Let the input space be $\mathcal{X}$ and the label space be $\mathcal{Y}$. We denote inputs as $x \in \mathcal{X}$ and labels as $y \in \mathcal{Y}$. We index the domains by $t \in \{0, \dots, T\}$, where $t = 0$ denotes the source domain and $t = T$ denotes the target domain. Each domain induces a marginal distribution $p_t$ over $\mathcal{X}$. Let $\mathcal{H}$ be a hypothesis class of classifiers $h : \mathcal{X} \to \mathcal{Y}$. The GDA task assumes that each domain admits a labeling function $q_t \in \mathcal{H}$. Given a loss function $\mathcal{L} : \mathcal{Y} \times \mathcal{Y} \to \mathbb{R}_{\geq 0}$, the generalization error of $h$ on domain $t$ is defined as $\varepsilon_{p_t}(h) = \mathbb{E}_{p_t(x)}\big[\mathcal{L}\big(h(x), q_t(x)\big)\big]$. A source classifier $q_0 \in \mathcal{H}$ can be learned via supervised learning on the source domain with minimal error $\varepsilon_{p_0}(q_0)$. The objective of GDA is to evolve $q_0$ through the intermediate domains to a classifier $h_T$ so as to minimize the target error $\varepsilon_{p_T}(h_T)$.

### 2.2. Flows for Intermediate Domain Generation

A flow describes the time-dependent evolution of particles induced by a smooth invertible map. Based on this, the intermediate domains can be seen as a discretization of a continuous flow linking source and target distributions. This motivates flow-based models (Chen et al., 2025b; 2026), which evolve a distribution over a fixed time horizon while preserving normalization, and are thus well-suited for GDA. From the flow perspective, intermediate domains are generated by the following ordinary differential equation:

$$\frac{\mathrm{d}x_t}{\mathrm{d}t} = v_t(x_t) = -\nabla \frac{\delta \mathbb{D}[p(x_t), p_T(x)]}{\delta p(x_t)}, \ x_{t=0} = x_0, \ (1)$$

where $p(x_t)$ is the PDF induced by $\{x_{t,i}\}_{i=1}^{\mathrm{N}}$, and we desire the law $p(x_T)$ to approximate the target $p_T(x)$. Here $v_t : \mathcal{X} \to \mathcal{X}$ is the velocity field. Notably, $\frac{\delta}{\delta p}$ denotes the first variation with respect to $p$, and the second equality sign is called "gradient flow". The core design problem is to choose $v_t$ so that $p(x_t) \xrightarrow[t \to T]{} p_T(x)$. A principled approach is to define $v_t$ as the steepest descent direction of some discrepancy functional $\mathbb{D}[p(x_t), p_T(x)]$ between

$p(x_t)$ and $p_T(x)$ as demonstrated in the second equal sign in Equation (1).

Among various choices, $f$-divergences are favored in GDA for their task-aligned objectives, stable probability-preserving dynamics, and efficient computation when compared to alternatives such as Sinkhorn divergence (Glaser et al., 2021). For an $f$-divergence, we have:

$$\mathbb{D}_f[p(x_t), p_T(x)] = \int p_T(x) f\left(\frac{p(x_t)}{p_T(x)}\right) \mathrm{d}x, \quad (2)$$

with $f : (0, \infty) \to \mathbb{R}$ convex and $f(x) = 0$ if and only if $x = 1$. A canonical example is the Kullback–Leibler (KL) divergence with $f(u) = u \log u$. In this case, we have:

$$v_t(x_t) = \nabla \log p_T(x) - \nabla \log p(x_t), \quad (3)$$

and, in the weak solution to the partial differential equations (Chen et al., 2024c; Evans, 2022; Liu, 2017), the induced dynamics yield the classical *Langevin dynamic* (Santambrogio, 2017; Wang et al., 2025a).

Intuitively, applying the forward Euler scheme with step size $\eta$ to the gradient flow in Equation (1) under an $f$-divergence yields a discrete-time generation for the intermediate domain, which is equivalent to solving the following optimization problem with the squared 2-Wasserstein distance as the regularization term (Jordan et al., 1998); the derivation is provided in Section C.1:

$$p(x_{t+\eta}) = \arg\min_{\rho \in \mathcal{P}_2(\mathbb{R}^D)} \frac{1}{2\eta} \mathcal{W}_2^2(\rho(x), p(x_t)) + \mathbb{D}_f[\rho(x), p_T(x)], \quad (4)$$

where $\mathcal{P}_2(\mathbb{R}^D)$ denotes the Wasserstein space (Villani et al., 2009), which is the set of the distributions with finite second moment. Here, $\mathcal{W}_2^2$ is the squared 2-Wasserstein distance, whose definition is given as follows:

$$\mathcal{W}_2^2(\rho, \xi) = \inf_{\pi \in \Pi(\rho, \xi)} \iint \|x - y\|_2^2 \, \pi(x, y) \, \mathrm{d}x \, \mathrm{d}y, \quad (5)$$

and $\Pi(\rho, \xi)$ is the set of joint distribution on $\mathbb{R}^D \times \mathbb{R}^D$ with marginal distributions $\rho$ and $\xi$.

## 3. Methodology

### 3.1. Motivation Analysis

Flow-based approaches, exemplified by gradient-flow methods, interpolate between the source and target distributions by gradually minimizing a discrepancy measure, typically an $f$-divergence, between the two domains. The success of these methods in GDA tasks critically depends on accurately estimating the target distribution's (normalized/unnormalized) PDF. Given a reliable estimate, one can construct a velocity field that progressively pushes source samples toward the target distribution.

However, estimating the target PDF from samples alone is ill-posed (Song et al., 2020; Vincent et al., 2010). When the estimate is inaccurate, the resulting flow may drive samples into low-density regions, producing a noticeable mismatch between generated and true target samples and degrading downstream performance. To highlight this issue, we compare two transport strategies: (1) EstTrans, which first estimates the target PDF via denoised score matching (Vincent, 2011) and then applies gradient flow-based transport, and (2) DirTrans, which directly transports samples using a learned optimal transport map. The qualitative results and the Wasserstein distance to ground-truth target samples are reported in Figures 1(a) and 1(b). As shown, inaccurate PDF estimates lead EstTrans to produce samples that deviate from the target distribution and incur a larger Wasserstein distance, whereas DirTrans better recovers target samples and achieves a smaller distance. These observations motivate the following questions: *How can we generate intermediate domains without estimating the PDF of target domain? How can robust intermediate domain generation be achieved within this framework? Does this approach improve the performance for GDA task?*

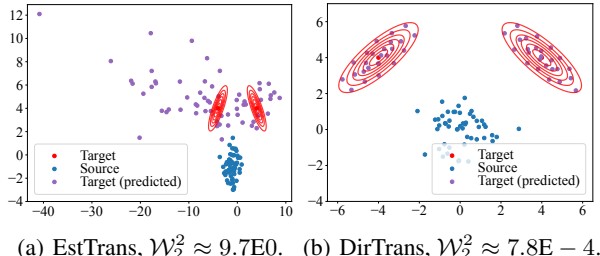

(a) EstTrans, $\mathcal{W}_2^2 \approx 9.7\mathrm{E}0$.   (b) DirTrans, $\mathcal{W}_2^2 \approx 7.8\mathrm{E}-4$.

*Figure 1.* Comparison between EstTrans and DirTrans.

### 3.2. Dual-Form Optimal Transport for GDA Task

As shown in Equation (4), simulating the gradient flow to generate intermediate domains is exactly equivalent to solving an optimization problem regularized by the Wasserstein distance. This insight opens up a practical alternative: *instead of explicitly estimating the target domain's probability density, one can guide source samples by directly tackling this optimization formulation*. Thus, we have the following proposition regarding the solution property of the problem defined in Equation (4):

**Proposition 3.1.** *Consider the following primal problem:*

$$\mathcal{L}^{Primal} = \arg\min_{\rho(x) \in \mathcal{P}_2(\mathbb{R}^D)} \frac{1}{2\eta} \mathcal{W}_2^2(\rho(x), p(x_t)) + \mathbb{D}_f[\rho(x), p_T(x)]. \quad (6)$$

*This problem is equivalent to the following semi-dual formulation:*

$$\mathcal{L}^{SemiDual} = \sup_w \mathbb{E}_{p(x_t)}\left[\inf_{\boldsymbol{T}} \frac{1}{2\eta} \|\boldsymbol{T}(x_t) - x_t\|_2^2 - w(\boldsymbol{T}(x_t))\right] \\ - \mathbb{E}_{p_T(x)}[f^\star(-w(x))], \quad (7)$$

where $w : \mathbb{R}^D \rightarrow \mathbb{R}$ is a measurable continuous function, $\boldsymbol{T} : \mathbb{R}^D \rightarrow \mathbb{R}^D$ is the transport map, and $f^\star := \sup_{y \geq 0}\{zy - f(y)\}$ denotes the convex conjugate of $f$.

Importantly, the structure of the semi-dual problem ensures that both $p_t(x)$ and $p_T(x)$ are involved only through expectation operators, rather than through explicit density evaluations. This enables the use of Monte Carlo methods to approximate all necessary integrals, thereby eliminating the need for access to the density function, particularly for the target domain, when constructing intermediate distributions. On this basis, following prior works (Choi et al., 2023; 2024; Korotin et al., 2023), we can parameterize both the dual potential $w$ and the transport map $\boldsymbol{T}$ by neural networks, denoted as $w_\phi$ and $\boldsymbol{T}_\theta$ respectively. The models are trained in an alternating adversarial scheme to learn the sequence of maps $\{\boldsymbol{T}_{\theta,t}\}_{t=0}^{T-1}$, which can be applied to generate intermediate domains progressively.

### 3.3. Robust Training for Semi-Dual Formulation

While Section 3.2 provides a semi-dual form of the gradient flow problem that avoids explicit PDF estimation in target domain, naively training $\mathcal{L}^{\text{SemiDual}}$ in Equation (7) is intrinsically unstable because of its composite 'sup–inf' structure. This instability is not merely algorithmic: the objective itself may be non-identifiable. We formalize this phenomenon by proving that the dual problem can have non-unique optima, as the following proposition shows:

**Proposition 3.2.** *Consider the semi-dual objective in Equation* (7) *and assume it admits at least one optimizer $w$. There exist $a \in \mathbb{R}^D$ and two points $b_1, b_2 \in \mathbb{R}^D$ such that $\|a - b_1\|_2 = \|a - b_2\|_2$. Let $p_\gamma(x_t) = \mathcal{N}(a, \gamma I)$ and $p_{T,\gamma}(x) = \frac{1}{2}\mathcal{N}(b_1, \gamma I) + \frac{1}{2}\mathcal{N}(b_2, \gamma I)$. Then, for all sufficiently small $\gamma > 0$, there exists a semi-dual optimizer $w_\gamma^*$ and a point $x_t^\gamma \in \mathbb{R}^D$ (with $x_t^\gamma \rightarrow a$ as $\gamma \rightarrow 0$) such that $\arg\min_{x \in \mathbb{R}^D}\{c(x_t^\gamma, x) - w_\gamma^*(x)\}$ is not a singleton. Equivalently, the inner $c$-transform (and thus $\boldsymbol{T}^*(x_t^\gamma)$) is non-unique.*

To address this issue, we incorporate an entropy regularization term into the primal objective Equation (6), which leads to the following proposition:

**Proposition 3.3.** *Let $\kappa(x_t, x) := p(x_t)\,p_T(x)$ denote the reference joint PDF. The entropy-regularized primal problem can be formulated as follows:*

$$\mathcal{L}^{\text{E-Primal}} = \arg\min_{\rho \in \mathcal{P}_2(\mathbb{R}^D)} \frac{1}{2\eta}\mathcal{W}_2^2(\rho(x), p(x_t)) + \mathbb{D}_f[\rho(x), p_T(x)]$$
$$+ \epsilon \iint \pi(x_t, x)[\log\frac{\pi(x_t, x)}{\kappa(x_t, x)} - 1]\mathrm{d}x_t\mathrm{d}x, \tag{8}$$

and is equivalent to the semi-dual optimization problem

$$\mathcal{L}^{\text{E-SemiDual}} = \inf_w \mathbb{E}_{p_T(x)}[f^\star(-w(x))]$$
$$+ \epsilon\mathbb{E}_{p(x_t)}[\log\mathbb{E}_{p_T(x)}(\exp(\frac{w(x) - \frac{1}{2\eta}\|x - x_t\|_2^2}{\epsilon}))], \tag{9}$$

where $w$ and $f^\star$ are as defined in Proposition 3.1.

We defer a more detailed discussion of the primal-dual relationship and the influence of the entropic regularizer to Section C.4.2 and Section C.4.3, respectively. Here, we focus on the theoretical analysis and practical implications of the entropic semi-dual formulation in Equation (9). Notably, this formulation admits the following uniqueness guarantee:

**Proposition 3.4.** *The semi-dual formulation in Equation* (9) *admits a unique optimal solution under standard regularity conditions on $f^\star$ and suitable normalization of $w$.*

Notably, as seen in Equation (9), the semi-dual objective depends solely on the potential $w$. Consequently, we can optimize a single model, which lowers the computational burden. We therefore parameterize $w$ by a neural network $w_\phi$ and carry out the optimization.

Finally, conditioned on the resulting $w_\phi$, we subsequently optimize the transport map $\boldsymbol{T}_\theta(x)$ via the following objective based on Equation (7):

$$\arg\min_\theta \quad \frac{1}{2\eta}\|x_t - \boldsymbol{T}_\theta(x_t)\|_2^2 - w_\phi(\boldsymbol{T}_\theta(x_t)). \tag{10}$$

Notably, we denote our approach as "E-SUOT", as the derivation of $\boldsymbol{T}_\theta$ is grounded in the Entropy-regularized Semi-dual Unbalanced Optimal Transport framework.

### 3.4. Overall Workflow for E-SUOT

Although Sections 3.2 and 3.3 have presented the E-SUOT framework for intermediate domain generation, they do not provide a unified view of the overall workflow for generating intermediate domains. To address this, we summarize the complete procedure in Algorithm 1 (Due to space limitations, other detailed information are summarized in Appendix E) and the corresponding illustration is given in Figure 2. As shown in the algorithm, the construction of $w_\phi$ and $\boldsymbol{T}_\theta$ are performed as separate steps, corresponding to Figure 2(a), and are illustrated in *Lines 3–6* and *Lines 7–10*, respectively. By iteratively executing the procedure described in *Lines 3–10*, we obtain a sequence of transport maps, $\mathcal{T} = \{\boldsymbol{T}_{\theta,t}\}_{t=0}^{T-1}$, which progressively transport samples from the source domain to the target domain, as we demonstrate in Figure 2(b).

Once the transport map sequence $\mathcal{T} = \{\boldsymbol{T}_{\theta,t}\}_{t=0}^{T-1}$ has been obtained, we proceed to train the classifier $h$ in a stage-wise manner along the transport path. Specifically, at each intermediate step $t$, we first map samples $x_t$ from the current

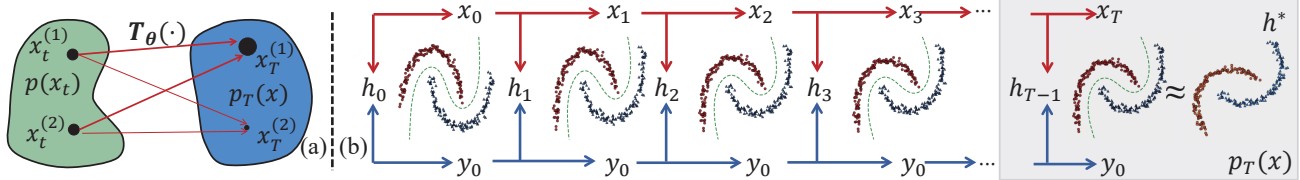

*Figure 2.* The illustration of the proposed E-SUOT: (a) the unbalanced OT formulation used to solve the transport map $\boldsymbol{T}_\theta(\cdot)$ at time $t$, where thicker arrows and larger points indicate higher mass flows, and (b) the evolution process from the source to the target domain. This figure is conceptually inspired by previous works on GDA and OT tasks (He et al., 2024; Wang et al., 2025b; Zhuang et al., 2024).

---

**Algorithm 1** Overall Workflow for E-SUOT

**Input:** Source domain samples: $\{(x_0^{(i)}, y_0^{(i)})\}_{i=1}^{\mathrm{N}}$, target domain samples: $\{(x_T^{(i)}, y_T^{(i)})\}_{i=1}^{\mathrm{N}}$, entropy regularization strength: $\epsilon$, step size: $\eta$, number of intermediate domains $T - 1$, neural network batch size $\mathcal{B}$, and neural network training epochs: $\mathcal{E}$.

**Output:** The set of optimal transportation map: $\mathcal{T} = \{\boldsymbol{T}_{\theta,t}\}_{t=0}^{T-1}$.

1: $\mathcal{T} \leftarrow \varnothing$.
2: **for** $t = 0$ to $T - 1$ **do**
3:     **for** $e = 1$ to $\mathcal{E}$ **do**
4:         Sample a batch $\{x_t^{(i)}\}_{i=1}^{\mathcal{B}} \sim \{(x_t^{(i)}, y_t^{(i)})\}_{i=1}^{\mathrm{N}}$ and $\{x_T^{(i)}\}_{i=1}^{\mathcal{B}} \sim \{(x_T^{(i)}, y_T^{(i)})\}_{i=1}^{\mathrm{N}}$.
5:         Update $w_{\phi,t}$ by: $\phi \leftarrow \arg\min_\phi \frac{\epsilon}{\mathcal{B}} \times \sum_{j=1}^{\mathcal{B}} \log \frac{1}{\mathcal{B}}$
        $\sum_{i=1}^{\mathcal{B}} [\exp(\frac{w_{\phi,t}(x_T^{(j)}) - \frac{1}{2\eta}\|x_t^{(j)} - x_T^{(i)}\|_2^2}{\epsilon})] + \frac{1}{\mathcal{B}}$
        $\sum_{j=1}^{\mathcal{B}} f^\star(-w_{\phi,t}(x_T^{(j)}))$.
6:     **end for**
7:     **for** $e = 1$ to $\mathcal{E}$ **do**
8:         Sample a batch $\{x_t^{(i)}\}_{i=1}^{\mathcal{B}} \sim \{(x_t^{(i)}, y_t^{(i)})\}_{i=1}^{\mathrm{N}}$.
9:         Update $\boldsymbol{T}_{\theta,t}$ by: $\theta \leftarrow \arg\min_\theta \frac{1}{\mathcal{B}} \sum_{i=1}^{\mathcal{B}} \frac{1}{2\eta}$
        $\|x_t^{(i)} - \boldsymbol{T}_{\theta,t}(x_t^{(i)})\|_2^2 - w_{\phi,t}(\boldsymbol{T}_{\theta,t}(x_t^{(i)}))$.
10:     **end for**
11:     $x_{t+1}^{(i)} \leftarrow \boldsymbol{T}_{\theta,t}(x_t^{(i)}), \forall i \in \{1, \ldots, \mathrm{N}\}$.
12:     $\mathcal{T} \leftarrow \mathcal{T} \cup \{\boldsymbol{T}_{\theta,t}\}$
13: **end for**

---

domain to the next intermediate domain $x_{t+1}$ using the corresponding transport map $\boldsymbol{T}_{\theta,t}$. We then update or train the model $h_t$ using the mapped data $x_{t+1}$ as input. By iteratively applying this procedure for $t = 0, \ldots, T - 1$, the model is progressively adapted along the sequence of intermediate domains, ultimately bridging the source and target domains.

### 3.5. Theoretical Analysis

Notably, our derivation sidesteps the explicit estimation of the PDF of the target domain by leveraging the semi-dual formulation. This leads to two important questions: (1) Can the proposed E-SUOT framework transport the source

domain sufficiently close to the target domain? (2) How does the model perform on the target domain after transport?

To address the first question, we present the following proposition, which qualitatively characterizes the discrepancy between $\rho(x)$ and $p_T(x)$:

**Proposition 3.5.** *The optimal solution $\rho^*(x)$ to the problem defined in Equation* (8) *satisfies the following bound when* $\mathcal{W}_2(p(x_t), p_T(x)) \leq 2\eta$:

$$\mathbb{D}_f[\rho^*(x), p_T(x)] \leq \mathcal{W}_2(p(x_t), p_T(x)). \quad (11)$$

In view of Proposition 3.5, we note that as $t$ increases, the transported density $\rho(x)$ progressively approaches the target distribution $p_T(x)$. A detailed analysis of this statement is provided in Section C.6. In addition, related discussions on the selection of $\eta$ for the case where $\mathbb{D}_f$ is the KL divergence are given in Section C.8.

Finally, we present the following proposition to demonstrate the model's performance on the target domain:

**Proposition 3.6.** *Under mild assumptions, the E-SUOT-based GDA ensures that the target domain generalization error is upper-bounded by the following inequality:*

$$\varepsilon_{p_T}(h_T) \leq \varepsilon_{p_0}(h_0) + \varepsilon_{p_0}(h_T^*) + \iota\zeta\mathcal{C} + \mathcal{S}_{stat}, \quad (12)$$

*where $\iota$ is the Lipschitz constant of the loss function, $\zeta$ is the Lipschitz constant bound for hypotheses in $\mathcal{H}$, $\mathcal{C}$ aggregates the cumulative domain transportation and label continuity costs along the gradual domain adaptation path, $\mathcal{S}_{stat}$ is the statistical error term, and $h_T^* \in \mathcal{H}$ is a reference hypothesis used to characterize the source-domain approximation gap.*

## 4. Main Experimental Results

### 4.1. Experimental Setup

We conduct case studies on four datasets. Specifically, for the GDA task, we use the Portraits dataset (Kumar et al., 2020) as well as MNIST 45° and MNIST 60° (LeCun, 1998). Following prior work (Sagawa & Hino, 2025; Zhuang et al., 2024), to ensure a fair comparison, we use the UMAP embeddings (McInnes et al., 2018) provided by Zhuang et al. (2024). In addition, to demonstrate the scalability of the proposed E-SUOT, we conduct experiments

on the Office-Home dataset (Venkateswara et al., 2017) for UDA task. We use the classification accuracy (denoted as "Accuracy") as the evaluation metric, defined as the proportion of correctly predicted target samples:

$$\text{Accuracy} = \left[ \frac{1}{N_{tgt}} \sum_{i=1}^{N_{tgt}} \mathbb{I}(\hat{y}_i = y_i) \right] \times 100\%,$$

where $N_{tgt}$ is the number of target test samples, $\hat{y}_i$ denotes the predicted label of the $i$-th sample, $y_i$ is the corresponding ground-truth label, and $\mathbb{I}(\cdot)$ is the indicator function that equals 1 when the condition holds and 0 otherwise. A larger accuracy value corresponds to better adaptation performance. Other details about these datasets are provided in Section E.1. Due to space limitations, additional experimental results are provided in Section F.

## 4.2. Baseline Comparison Results

**GDA Task.** We first compare our proposed approach with several existing GDA-based methods, including Self-training (Wei et al., 2021), GST (with 4 intermediate domains) (Kumar et al., 2020), GOAT (He et al., 2024), CNF (Sagawa & Hino, 2025), MMDSW (Bonet et al., 2025), SMMD (Hertrich et al., 2024), STDW (Wang et al., 2025c), AST (Shi & Liu, 2023), and GGF (Zhuang et al., 2024). The experimental results are listed in Table 1.

*Table 1.* GDA accuracy (%) comparison results.

| Method | Portraits | | MNIST 45° | | MNIST 60° | |
|---|---|---|---|---|---|---|
| | Acc. (%) | Δ | Acc. (%) | Δ | Acc. (%) | Δ |
| Source | 71.2 | - | 58.4 | - | 36.8 | - |
| Self Train | 77.4 | ↑8.8% | 58.7 | ↑0.5% | 39.9 | ↑8.6% |
| GST (4) | 76.1 | ↑6.9% | 59.2 | ↑1.3% | 39.9 | ↑8.5% |
| GOAT | 74.9 | ↑5.3% | 65.0 | ↑11.3% | 37.2 | ↑1.1% |
| CNF | 80.0 | ↑12.4% | 57.6 | ↓1.4% | 41.8 | ↑13.5% |
| MMDSW | 80.2 | ↑12.6% | 57.9 | ↓0.9% | 42.2 | ↑14.7% |
| SMMD | 79.8 | ↑12.0% | 57.9 | ↓0.8% | 42.2 | ↑14.7% |
| STDW | 84.3 | ↑18.4% | 60.3 | ↑3.2% | 43.9 | ↑19.2% |
| AST | 84.2 | ↑18.3% | 58.3 | ↓0.2% | 41.3 | ↑12.2% |
| GGF | 83.4 | ↑17.2% | 57.7 | ↓1.2% | 40.8 | ↑11.0% |
| E-SUOT | **86.4** | ↑21.5% | **72.1** | ↑23.4% | **51.0** | ↑38.6% |

*Kindly Note*: **Bolded** results are the best results. Underlined results are the second best results. Δ denotes performance change percentage of the classifier trained on the source domain.

As shown in Table 1, our proposed E-SUOT consistently achieves the best performance across all evaluated datasets. Specifically, E-SUOT improves over the source-only baseline by 21.5%, 23.4%, and 38.6% on Portraits, MNIST 45°, and MNIST 60°, respectively, demonstrating its effectiveness in constructing reliable intermediate domains for gradual domain adaptation. Compared with recent GDA approaches, E-SUOT also shows clear advantages. Although STDW achieves the second-best performance on Portraits and MNIST 60°, and GOAT obtains the second-best result on MNIST 45°, their performance gains are not consistent across all datasets. Similarly, several competitive methods,

including CNF, AST, and GGF, perform well on certain tasks but occasionally underperform the source-only baseline or provide only marginal improvements. In particular, the occasional degradation of flow-based methods such as CNF and GGF may be attributed to the difficulty of accurately estimating target domain densities, which is consistent with our motivation analysis in Section 3.1.

**UDA Task.** We further evaluate E-SUOT on the Office-Home dataset under the UDA setting to demonstrate its scalability. We compare E-SUOT with a broad range of representative baselines, including UDA methods such as DANN (Ganin & Lempitsky, 2015), MSTN (Xie et al., 2018), GVB-GD (Cui et al., 2020), RSDA (Gu et al., 2020; 2024), LAMBDA (Le et al., 2021), SENTRY (Prabhu et al., 2021), FixBi (Na et al., 2021), CST (Liu et al., 2021a), and CoVi (Na et al., 2022), as well as GDA methods including MMDSW (Bonet et al., 2025), SMMD (Hertrich et al., 2024), STDW (Wang et al., 2025c), AST (Shi & Liu, 2023), and GGF (Zhuang et al., 2024). The evaluation is conducted on the Office-Home dataset (Venkateswara et al., 2017). In our UDA experiments, we use CoVi as the embedding feature extraction backbone, with additional experimental details provided in Section E.2. The corresponding experimental results are summarized in Table 2.

From Table 2, we observe that E-SUOT achieves the best average accuracy on the Office-Home dataset. Although it is not the top method on every transfer direction, it remains consistently competitive and obtains several second-best results, e.g., on Ar→Cl, Ar→Pr, Pr→Cl, and Pr→Rw, with only small gaps to the best-performing approaches. This observation suggests that the strength of E-SUOT lies in its stable cross-task performance rather than in dominating a few individual transfers.

## 4.3. Ablation Studies

### 4.3.1. ABLATION ON $T_\theta$ TRAINING.

To ensure fair comparisons under the same feature representations (UMAP embeddings from Zhuang et al. (2024)), we conduct the ablation study in this part on the three datasets in the GDA task only. In this part, we perform ablation studies from two perspectives: the training strategy for $T_\theta$ and the choice of $f$-divergence. For the *training strategy*, we 1). examine the effect of removing the entropy regularization term, reducing the method to the adversarial training strategy in Equation (7) to support Proposition 3.4, and 2). evaluate a barycentric projection approach analogous to flow matching (Lipman et al., 2023), where the transport plan is first estimated and then used to project source samples toward the target, subsequently being refined during training. For the objective *functional*, we study different parameterizations of $f^\star$, such as employing non-decreasing convex functions like 1) SftPls (softplus function), and also com-

*Table 2.* UDA accuracy (%) comparison on the Office-Home dataset.

| Method | Ar→Cl | Ar→Pr | Ar→Rw | Cl→Ar | Cl→Pr | Cl→Rw | Pr→Ar | Pr→Cl | Pr→Rw | Rw→Ar | Rw→Cl | Rw→Pr | Avg. |
|---|---|---|---|---|---|---|---|---|---|---|---|---|---|
| DANN | 45.6 | 59.3 | 70.1 | 47.0 | 58.5 | 60.9 | 46.1 | 43.7 | 68.5 | 63.2 | 51.8 | 76.8 | 57.6 |
| MSTN | 49.8 | 70.3 | 76.3 | 60.4 | 68.5 | 69.6 | 61.4 | 48.9 | 75.7 | 70.9 | 55.0 | 81.1 | 65.7 |
| GVB-GD | 57.0 | 74.7 | 79.8 | 64.6 | 74.1 | 74.6 | 65.2 | 55.1 | 81.0 | 74.6 | 59.7 | 84.3 | 70.4 |
| RSDA | 53.2 | 77.7 | 81.3 | 66.4 | 74.0 | 76.5 | 67.9 | 53.0 | 82.0 | 75.8 | 57.8 | 85.4 | 70.9 |
| LAMDA | 57.2 | 78.4 | 82.6 | 66.1 | **80.2** | **81.2** | 65.6 | 55.1 | 82.8 | 71.6 | 59.2 | 83.9 | 72.0 |
| SENTRY | **61.8** | 77.4 | 80.1 | 66.3 | 71.6 | 74.7 | 66.8 | **63.0** | 80.9 | 74.0 | **66.3** | 84.1 | 72.3 |
| FixBi | 58.1 | 77.3 | 80.4 | 67.7 | 79.5 | 78.1 | 65.8 | 57.9 | 81.7 | 76.4 | 62.9 | **86.7** | 72.7 |
| CST | 59.0 | **79.6** | **83.4** | **68.4** | 77.1 | 76.7 | **68.9** | 56.4 | **83.0** | 75.3 | 62.2 | 85.1 | 72.9 |
| CoVi | 58.5 | 78.1 | 80.0 | 68.1 | 80.0 | 77.0 | 66.4 | 60.2 | 82.1 | **76.6** | 63.6 | 86.5 | 73.1 |
| GGF | 59.4 | 75.6 | 81.7 | 67.6 | 77.6 | 78.0 | 67.4 | 61.0 | 82.7 | 75.9 | 62.4 | 85.4 | 72.9 |
| MMDSW | 59.2 | 75.6 | 81.7 | 67.6 | 77.4 | 77.9 | 67.4 | 61.0 | 82.6 | 75.9 | 62.4 | 85.4 | 72.8 |
| SMMD | 58.8 | 74.8 | 81.7 | 67.6 | 77.4 | 77.9 | 67.4 | 61.0 | 82.6 | 75.9 | 62.4 | 85.4 | 72.7 |
| STDW | 59.5 | 75.6 | 81.7 | 67.6 | 77.6 | 78.0 | 67.4 | 61.0 | 82.7 | 75.9 | 62.5 | 85.4 | 72.9 |
| AST | 58.9 | 75.5 | 81.7 | 67.6 | 77.4 | 77.9 | 67.4 | 61.0 | 82.6 | 75.9 | 62.4 | 85.4 | 72.8 |
| E-SUOT | 61.6 | 79.3 | 81.8 | 67.6 | 77.7 | 78.1 | 67.4 | 61.2 | 82.9 | 76.3 | 62.5 | 85.2 | **73.5** |
| Win Counts | 13 | 13 | 12 | 6 | 11 | 12 | 7 | 13 | 13 | 12 | 10 | 6 | 14 |

*Kindly Note*: **Bolded** and underlined results are the first and second best results, respectively. "Avg." is short for "Average."

pare the 2) $\chi^2$ divergence and the 3) identity function. More detailed information on these experiments' implementation is provided in Appendix E.3. The ablation study results are summarized in Table 3.

*Table 3.* Ablation study results on GDA setting.

| | Dataset | | Portraits | | MNIST 45° | | MNIST 60° | |
|---|---|---|---|---|---|---|---|---|
| | Metric | | Acc. (%) | Δ | Acc. (%) | Δ | Acc. (%) | Δ |
| Training | Adversarial | KL | 74.8 | ↓13.4% | 52.0 | ↓27.8% | 34.9 | ↓31.5% |
| | Barycentric | KL | 83.9 | ↓3.0% | 62.5 | ↓13.3% | 38.3 | ↓24.8% |
| Functional | Entropy | SftPls | 80.1 | ↓7.3% | 59.7 | ↓17.2% | 38.2 | ↓25.1% |
| | Entropy | $\chi^2$ | 79.8 | ↓7.7% | 60.2 | ↓16.5% | 42.4 | ↓16.9% |
| | Entropy | Identity | 81.2 | ↓6.1% | 59.6 | ↓17.4% | 39.6 | ↓22.3% |
| | Entropy | KL | 86.4 | - | 72.1 | - | 51.0 | - |

*Kindly Note*: Δ denotes performance change percentage compared to E-SUOT with entropy regularization and KL divergence.

From Table 3, we find that adversarial training performs the worst, underscoring the importance of entropy regularization for robust model training in Section 3.3. While barycentric mapping is competitive, it struggles on complex datasets such as MNIST 45° and MNIST 60°, highlighting the need for the semi-dual formulation. Additionally, alternatives to KL divergence, especially SftPls, cause significant performance drops, emphasizing the importance of proper divergence selection for conducting GDA task. We also observe that replacing KL divergence with alternatives such as $\chi^2$ divergence, the identity function, or particularly Softplus results in substantial performance degradation, further illustrating that choosing a suitable discrepancy to drive the evolution of source domain to target domain is critical for promising the performance of GDA.

### 4.3.2. ABLATION ON EMBEDDING FEATURE

Since our E-SUOT uses UMAP embeddings obtained from backbones pretrained with different UDA algorithms, we further conduct ablations on the Office-Home dataset from two aspects: (1) the pretraining strategy and (2) the UMAP embedding dimensionality. The corresponding results are reported in the following contents.

**Pretraining Strategy.** The ablation study on the pretraining strategy is reported in Table 2. We evaluate the proposed E-SUOT by integrating it with four representative UDA algorithms, namely MSTN, RSDA, FixBi, and CoVi, on the Office-Home dataset. For a fair comparison, we apply UMAP to obtain 8-dimensional embeddings. The corresponding results are listed in Table 4.

As shown in Table 4, E-SUOT consistently improves all baselines, with average gains of 8.1%, 3.3%, 1.0%, and 0.5% over MSTN, RSDA, FixBi, and CoVi, respectively. The largest improvement (+16.1%) occurs on the challenging Ar→Cl transfer, suggesting that E-SUOT is particularly beneficial under large domain shifts. Notably, E-SUOT remains effective when combined with strong recent methods (e.g., FixBi and CoVi), indicating that it serves as a complementary module. To gain further insight, we provide additional discussions in Section F.5 regarding the results reported in Table 2. Overall, these consistent gains across different pretrained feature extractors demonstrate the scalability and robustness of E-SUOT for UDA.

*Table 4.* Accuracy (%) improvement over different UDA feature extractor backbones.

| Method | Ar→Cl | Ar→Pr | Ar→Rw | Cl→Ar | Cl→Pr | Cl→Rw | Pr→Ar | Pr→Cl | Pr→Rw | Rw→Ar | Rw→Cl | Rw→Pr | Avg. |
|---|---|---|---|---|---|---|---|---|---|---|---|---|---|
| MSTN | 49.8 | 70.3 | 76.3 | 60.4 | 68.5 | 69.6 | 61.4 | 48.9 | 75.7 | 70.9 | 55.0 | 81.1 | 65.7 |
| E-SUOT+MSTN | 57.8 | 75.9 | 79.6 | 65.5 | 75.9 | 74.8 | 64.5 | 58.5 | 81.4 | 73.7 | 59.5 | 84.4 | 71.0 |
| Δ | ↑16.1% | ↑8.0% | ↑4.3% | ↑8.4% | ↑10.8% | ↑7.5% | ↑5.0% | ↑19.6% | ↑7.5% | ↑3.9% | ↑8.2% | ↑4.1% | ↑8.1% |
| RSDA | 53.2 | 77.7 | 81.3 | 66.4 | 74.0 | 76.5 | 67.9 | 53.0 | 82.0 | 75.8 | 57.8 | 85.4 | 70.9 |
| E-SUOT+RSDA | 61.5 | 78.8 | 81.7 | 67.6 | 77.3 | 77.6 | 67.2 | 61.0 | 82.7 | 76.0 | 62.4 | 85.3 | 73.3 |
| Δ | ↑15.7% | ↑1.4% | ↑0.5% | ↑1.8% | ↑4.4% | ↑1.5% | ↓1.0% | ↑15.2% | ↑0.9% | ↑0.3% | ↑7.9% | ↓0.1% | ↑3.3% |
| FixBi | 58.1 | 77.3 | 80.4 | 67.7 | 79.5 | 78.1 | 65.8 | 57.9 | 81.7 | 76.4 | 62.9 | 86.7 | 72.7 |
| E-SUOT+FixBi | 61.7 | 79.1 | 81.7 | 67.6 | 77.6 | 78.2 | 67.3 | 61.3 | 82.7 | 76.0 | 62.5 | 85.3 | 73.4 |
| Δ | ↑6.2% | ↑2.3% | ↑1.6% | ↓0.1% | ↓2.4% | ↑0.1% | ↑2.3% | ↑5.9% | ↑1.2% | ↓0.5% | ↓0.6% | ↓1.6% | ↑1.0% |
| CoVi | 58.5 | 78.1 | 80.0 | 68.1 | 80.0 | 77.0 | 66.4 | 60.2 | 82.1 | 76.6 | 63.6 | 86.5 | 73.1 |
| E-SUOT+CoVi | 61.6 | 79.3 | 81.8 | 67.6 | 77.7 | 78.1 | 67.4 | 61.2 | 82.9 | 76.3 | 62.5 | 85.2 | 73.5 |
| Δ | ↑5.3% | ↑1.5% | ↑2.2% | ↓0.7% | ↓2.9% | ↑1.4% | ↑1.5% | ↑1.7% | ↑1.0% | ↓0.4% | ↓1.7% | ↓1.5% | ↑0.5% |

*Kindly Note*: Δ denotes the relative accuracy change of E-SUOT over the corresponding vanilla UDA feature extractor backbone.

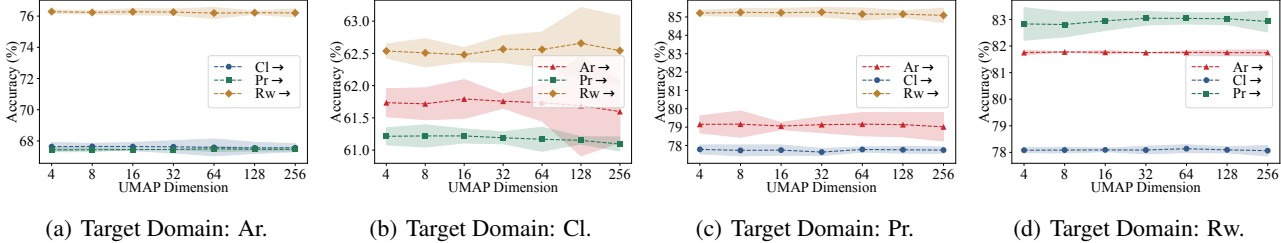

| (a) Target Domain: Ar. | (b) Target Domain: Cl. | (c) Target Domain: Pr. | (d) Target Domain: Rw. |

*Figure 3.* Ablation study of E-SUOT performance with varying UMAP embedding dimensions on the Office-Home dataset. For dimension 256, the vanilla backbone features are used without UMAP. The shaded area indicates the ± 5.0 times standard deviation error.

**UMAP Embedding Dimension.** We further conduct an ablation study to analyze the sensitivity of E-SUOT with respect to the dimensionality of the UMAP embedding. Specifically, we vary the reduced feature dimension from 4 to 256, where 256 corresponds to using the original 256-dimensional backbone features (i.e., without UMAP). Through this experiment, we aim to evaluate the impact of different embedding dimensions on the performance of E-SUOT. The corresponding results are presented in Figure 3.

From Figure 3, we observe that as the embedding dimension changes, the classification accuracy tends to fluctuate within a relatively small range across all target domains. In detail, for each target domain, the performance remains quite stable as we vary the UMAP dimension from 4 to 256, it can be observed that no drastic drops are observed. In some domains (e.g., Ar in Figure 3(a), Pr in Figure 3(c) and Rw in Figure 3(d)), the accuracy curves are almost flat, indicating the proposed method is insensitive to UMAP dimension choices in these cases. For domain Cl in Figure 3(b), although the standard deviation is higher, the main trend is still relatively stable. It suggests that E-SUOT retains strong robustness to the UMAP embedding dimension and does not rely heavily on finetuning this hyperparameter. In summary, the ablation analysis in Figure 3 shows that E-SUOT remains robust to variations in the UMAP embedding dimension.

## 5. Related Works

### 5.1. Gradual Domain Adaptation

GDA seeks to bridge the distributional gap between source and target domains by leveraging a sequence of intermediate domains, thereby enabling more fine-grained adaptation. Early works have explored self-training strategies (Kumar et al., 2020), adversarial objectives (Shi & Liu, 2023; Wang et al., 2020), and provided generalization bounds under gradual distribution shifts (Dong et al., 2022; Kumar et al., 2020; Wang et al., 2022). However, these approaches often depend on the availability of discrete intermediate domains (Chen & Chao, 2021). To address this, optimal transport approaches (Abnar et al., 2021; He et al., 2024) have been leveraged to construct intermediate domains along the Wasserstein geodesic, ensuring minimal distributional discrepancy in the adaptation process. More recently, flow-based GDA has emerged, which explicitly models domain evolution and synthesizes continuous intermediate distributions via parametric flows. For instance, Sagawa & Hino (2025) uses continuous normalizing flows to parameterize domain trajectories as ODEs in the data space, while Zhuang et al. (2024) incorporates label information into this evolution and employs gradient flows to realize the steepest transformation from source to target domain.

Nevertheless, flow-based GDA approaches still require explicit estimation of the target domain's PDF to guide the evolution, and inaccuracies in this estimation can lead to performance drops in target domain. To address this limitation, we reformulate the flow-based approach from a semi-dual formulation (see Proposition 3.1), which unifies the flow-based and optimal transport methods. Building on this, we further propose a convergence-guaranteed approach with the help of entropy regularization (Proposition 3.3) and analyze its generalization error (see Proposition 3.6).

### 5.2. Semi-Dual Formulation of Gradient Flows

Gradient flow (Santambrogio, 2017), which seeks to optimize a specified functional in the space of probability measures, has played a critical role in both sampling and optimization algorithm design. For gradient flows induced by $f$-divergences (with the KL divergence being the notable example), such as Langevin sampling (Welling & Teh, 2011; Xu et al., 2026), have been extensively explored to generate samples that progressively transition from the source domain toward the target domain. However, these methods typically assume access to an exact (unnormalized) PDF for the target distribution (Liu, 2017; Liu & Wang, 2016), which is often infeasible in practice when only samples are available. To overcome this, several approaches have explored dual formulations of $f$-divergence (Nguyen et al., 2007; 2010), which avoid explicit density estimation for the target domain and instead optimize primal formulation (Choi et al., 2023; 2024; Fan et al., 2022; Gazdieva et al., 2023; Korotin et al., 2023; Rout et al., 2022; Xu et al., 2024; 2025). These dual-formulation methods, however, generally require adversarial optimization characterized by a composite "sup-inf" structure in order to properly approximate the dual objective when implemented with neural networks (Arjovsky et al., 2017; Nowozin et al., 2016). In addition, recent works attempt to introduce entropy regularization into UOT-based generative modeling. For example, Choi & Choi (2024) addresses the resulting inner entropy-regularized problem via the Schrödinger bridge formulation (Chen et al., 2016a;b; 2021; Guo et al., 2026).

To our knowledge, the proposed E-SUOT differs from these approaches in three key aspects. First, we provide a theoretical analysis from the perspective of the non-uniqueness of optimal solutions in Proposition 3.2, highlighting that such adversarial formulations can suffer from this issue, which may hinder training stability. Second, building upon this insight, we introduce an entropy regularization term that transforms the adversarial game into a sequential optimization paradigm, as formalized in Proposition 3.3. We further prove that this regularization improves stability by ensuring the uniqueness of the optimum in Proposition 3.4, and establish its convergence in Proposition 3.5. Third, our inner sup problem adopts a relative-entropy regularization with respect to the reference measure $\kappa(x_t, x) = p(x_t)p_t(x)$, rather than the Brownian endpoint reference commonly induced by the classical Schrödinger bridge formulation (Chen et al., 2016a;b; 2021). The Brownian endpoint reference typically has the form $\kappa_{\text{Brown}}(x_t, x) = p(x_t)p(x|x_t)$, where $p(x|x_t)$ is the transition kernel for Brownian motion. Since $p(x|x_t)$ depends on $x_t$, the two endpoints are generally coupled under this reference, and thus $p(x_t)$ does not factorize as $p(x_t)p_t(x)$. In contrast, our reference measure explicitly factorizes into the product of the source and target marginals, corresponding to an independent endpoint reference.

## 6. Conclusions

In this paper, we addressed the challenge in flow-based GDA, namely the reliance on explicit estimation of the target domain PDF inherited from traditional $f$-divergence formulations. To address this issue, we recast the flow simulation as a recursive optimization problem with Wasserstein distance regularization, leading to the Wasserstein proximal recursion. Building on this, we derived a novel semi-dual formulation that avoids explicit estimation of the target density. However, we observed that the resulting semi-dual structure introduces instability due to its composite 'sup-inf' structure. To address this issue, we proposed an entropy regularization term that eliminates the inner inf operator, thereby restoring stability of the optimal solution solving process theoretically. Based on these insights, we developed a new GDA framework called "E-SUOT" and provided theoretical guarantees for its convergence and generalization. Finally, extensive experiments validate the effectiveness and practical advantages of our approach.

**Limitations.** Although E-SUOT shows promising empirical performance, several limitations remain. First, the current formulation mainly performs marginal feature alignment and assumes label invariance along the transport path, without explicitly incorporating pseudo-label information (Courty et al., 2017a). This may limit robustness under strong covariate shift, label shift, or concept drift. Second, while entropy regularization improves computational efficiency, it may blur the sparsity of the transport plan and affect the interpretability of the learned transport potential. Third, this work adopts the Wasserstein distance as the main discrepancy measure; exploring alternative geometries, such as Fisher-Rao (Wang et al., 2023; Zhu, 2025) or Bures-Wasserstein discrepancies (Wang et al., 2026b), may further improve the construction of intermediate domains. Finally, E-SUOT currently relies on a multi-network, multi-stage offline training pipeline, which leaves room for more parameter-efficient designs (Hu et al., 2022). Due to space limitations, detailed discussions and the corresponding possible alleviation strategies are provided in Section G.

## Acknowledgements

Zhouchen Lin was supported by the Beijing Natural Science Foundation under Grant No. L257007, the National Natural Science Foundation of China under Grant No. 62276004, and the Beijing Major Science and Technology Project under Grant No. Z251100008425006. Zhichao Chen was supported by the China Postdoctoral Science Foundation under Grant No. 2025M781449. Yunfei Teng was supported by Beijing Postdoctoral Research Foundation. The first author, Zhichao Chen, would like to express his gratitude to Dr. Chunyuan Zheng from Peking University for his valuable guidance on organizing the rebuttal language during the conference rebuttal stage, and to Dr. Jaemoo Choi of Georgia Institute of Technology, whose relevant works inspired this research. The authors acknowledge the anonymous reviewers for their insightful comments and suggestions.

## Impact Statement

GDA addresses a critical challenge in machine learning: transferring knowledge from a labeled source domain to an unlabeled target domain when there is a substantial gap between the two. Rather than relying on abrupt, one-shot shifts, GDA interpolates through a series of intermediate domains, allowing for a smoother and more effective adaptation process. This paradigm has direct implications for many real-world applications. For example, in recommender systems, GDA enables knowledge transfer to serve cold-start users or to integrate new items, and in language processing it allows models trained on high-resource languages to adapt more robustly to low-resource languages. Our work advances the field of GDA by unifying flow-based methods and optimal transport within the semi-dual formulation, identifying fundamental issues of stability and generalization that have limited previous approaches. We further propose theoretically-grounded regularization strategies that improve the robustness and reliability of the adaptation process. These advances not only deepen the theoretical understanding of GDA but also offer practical benefits for deploying adaptable machine learning systems in diverse settings. We believe our findings will help catalyze the development of more general, stable, and information-preserving domain adaptation methods, with impact across fields ranging from recommendation to broader AI applications.

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

# A. Nomenclature

To facilitate reading our manuscript, we provide the major technical terminology we use during our derivation in Table 5.

*Table 5.* Technical terminology table.

| Symbol | Description | Symbol | Description |
|--------|-------------|--------|-------------|
| $T$ | Intermediate domain number | softmax | Softmax function |
| $\Delta$ | Performance difference | $\mathscr{A}$ | Upper bound on the $L_2$ norm of the first variation of the KL divergence |
| $\|\|\|\|_2$ | $L_2$ norm | $\mathscr{B}$ | Upper bound on the $L_2$ norm of the gradient of the first variation of the KL divergence |
| $\arg\min$ | Argument of the minimum | $\mathscr{H}_0$ | Light-tail constant |
| $\boldsymbol{T}$ | Transportation map | $\nabla$ | Gradient operator |
| $\boldsymbol{T}^{-1}$ | Inverse transformation of the transportation map | $\omega$ | Parameter of classifier |
| $\delta$ | Variation operator | $\phi$ | Parameter of potential function |
| $\det$ | Determinant | $\pi$ | Transportation plan |
| $\epsilon$ | Entropy regularization coefficient | $\rho^*(x)$ | Optimal PDF |
| $\eta$ | Discretization stepsize | $\sup$ | Supremum |
| $\exp$ | Exponential function | $\theta$ | Parameter of transportation map |
| $\inf$ | Infimum | $\delta$ | Dirac delta measure |
| $\iota$ | Lipschitz constant of the loss function | $\varepsilon$ | Generalization error |
| $\kappa(x,y)$ | Reference joint PDF | $\widehat{h}$ | Logit layer of classifier |
| $\lambda_1$ | Coefficient of unbalanced optimal transport | $\widehat{y}$ | Predicted label |
| $\lambda_2$ | Coefficient of unbalanced optimal transport | $\zeta$ | Lipschitz constant bound for hypotheses in $\mathcal{H}$ |
| $\mathbb{D}_f[\rho(x), p_T(x)]$ | $f$-divergence of distribution $q(x)$ with respect to distribution $p(x)$ | $c$ | Cost matrix |
| $\mathbb{D}_{\mathrm{KL}}[\rho(x), p_T(x)]$ | Kullback-Leibler divergence of distribution $q(x)$ with respect to distribution $p(x)$ | $f^\star(x)$ | Convex conjugate function of $f(x)$ |
| $\mathbb{E}_q(x)[f(x)]$ | Expectation of function $f(x)$ with respect to distribution $q(x)$ | $h$ | Classifier |
| $\mathbb{I}$ | indicator function | $t$ | Time index |
| $\mathcal{B}$ | Batch size | $u$ | Kantorovich potential function |
| $\mathcal{C}$ | Cumulative domain transportation and label continuity costs along the adaptation path | $v_t$ | Velocity field |
| $\mathcal{H}$ | Hypothesis class for classifier | $w$ | Kantorovich potential function |
| $\mathcal{K}$ | Kernel Function | $x$ | Input |
| $\mathcal{L}$ | Loss function | $y$ | Label |
| $\mathcal{P}_2(\mathbb{R}^{\mathrm{D}})$ | D-dimensional Wasserstein space | GDA | Gradual domain adaptation |
| $\mathcal{S}_{\mathrm{stat}}$ | Statistical error term. | JKO | Jordan-Kinderlehrer-Otto |
| $\mathcal{T}$ | Set of transportation map | KL divergence | Kullback-Leibler divergence |
| $\mathcal{W}_p$ | $p$-Wasserstein distance | OT | Optimal transport |
| $\mathcal{X}$ | Input space | PDF | Probability density function |
| $\mathcal{Y}$ | Label space | UDA | Unsupervised domain adaptation |
| $\mathrm{N}$ | Sample size | UOT | Unbalanced optimal transport |
| $\mathrm{d}$ | Differential operator | | |

## B. Mathematical Foundations of Optimal Transport

We begin by reviewing the relevant background of optimal transport, based on references (Peyré & Cuturi, 2019; Villani et al., 2009). Assume continuous variables with densities: source $\rho(x)$ supported on $\mathcal{X}$, target $\xi(y)$ supported on $\mathcal{Y}$, and a cost $c(x, y) \geq 0$. We search for a joint probability density function which is called transport plan $\pi(x, y) \geq 0$ such that:

$$\int \pi(x, y)\,\mathrm{d}y = \rho(x), \tag{13a}$$

$$\int \pi(x, y)\,\mathrm{d}x = \xi(y), \tag{13b}$$

and minimize expected cost:

$$\inf_{\pi \geq 0} \iint c(x, y)\,\pi(x, y)\,\mathrm{d}y\,\mathrm{d}x, \tag{14}$$

where $c(x, y)$ is the cost function, for example, squared Euclidean norm: $c(x, y) = \|x - y\|_2^2$. Notably, when $c(x, y)$ is chosen as the squared Euclidean distance, the resulting optimal transport cost corresponds to the squared Wasserstein-2 distance between the two PDFs.

Introducing potentials $u(x)$ and $w(y)$ as Lagrange multipliers for the marginal constraints, we get:

$$\sup_{u,w} \left[ \int u(x)\,\rho(x)\,\mathrm{d}x + \int w(y)\,\xi(y)\,\mathrm{d}y \right] \quad \text{s.t.} \quad u(x) + w(y) \leq c(x, y)\ \forall x, y. \tag{15}$$

Intuitively, $u$ and $w$ are "prices", and the constraint ensures the total "price" never exceeds the cost function. In addition, $u$ and $w$ are also called "(Kantorovich) potential" in optimal transport (Peyré & Cuturi, 2019).

Based on this, we can eliminate one potential via the $c$-transform (Villani et al., 2009) as follows:

$$w^c(x) := \inf_y c(x, y) - w(y). \tag{16}$$

Based on this, we get the semi-dual formulation of optimal transport problem (Choi et al., 2023; 2024; 2025; Korotin et al., 2021; 2023) which maximizes over one potential:

$$\sup_w \int w^c(x)\,\rho(x)\,\mathrm{d}x + \int w(y)\,\xi(y)\,\mathrm{d}y. \tag{17}$$

Notably, when total mass may differ or we allow creation/destruction of mass, we can relax marginal constraints using the $f$-divergence-based penalty terms (Chizat et al., 2018; Zhang et al., 2022). Specifically, we still want to optimize $\pi(x, y) \geq 0$, but we will penalize deviations of the induced marginals $\tilde{\rho}(x) := \int \pi(x, y)\,\mathrm{d}y$ and $\tilde{\xi}(y) := \int \pi(x, y)\,\mathrm{d}x$ from $\rho(x)$ and $\xi(y)$:

$$\inf_{\pi \geq 0} \iint c(x, y)\,\pi(x, y)\,\mathrm{d}y\,\mathrm{d}x + \lambda_1\,\mathbb{D}_f(\tilde{\rho}(x), \rho(x)) + \lambda_2\,\mathbb{D}_f(\tilde{\xi}(y), \xi(y)), \tag{18}$$

where $\mathbb{D}_f(\tilde{\rho}(x), \rho(x)) = \int \rho(x)\,f\big(\frac{\tilde{\rho}(x)}{\rho(x)}\big)\,\mathrm{d}x$ and $\lambda_{1,2} > 0$.

In addition, using the convex conjugate $f^\star$, the dual problem becomes

$$\sup_{u,w} -\int \rho(x)\,f_1^\star\big(-u(x)\big)\,\mathrm{d}x - \int \xi(y)\,f_2^\star\big(-w(y)\big)\,\mathrm{d}y \quad \text{s.t.} \quad u(x) + w(y) \leq c(x, y)\ \forall x, y, \tag{19}$$

where $f_1, f_2$ are the chosen divergences on each side.

Similarly, we can eliminate one potential via the $c$-transform as follows:

$$\sup_w -\int \rho(x)\,f_1^\star\big(-w^c(x)\big)\,\mathrm{d}x - \int \xi(y)\,f_2^\star\big(-w(y)\big)\,\mathrm{d}y, w^c(x) = \inf_y\{c(x, y) - w(y)\}. \tag{20}$$

Based on this, we obtain the semi-dual formulation of the unbalanced optimal transport problem.

## C. Theoretical Derivation

### C.1. Derivation of Equation (4)

In this subsection, we want to derive the following equivalent relationship:

$$x_{t+\eta} = x_t - \eta \nabla \frac{\delta \mathbb{D}_f[p(x_t), p_T]}{\delta p(x_t)} \Rightarrow p(x_{t+\eta}) = \arg\min_{\rho(x) \in \mathcal{P}_2(\mathbb{R}^D)} \frac{1}{2\eta} \mathcal{W}_2^2(\rho(x), p(x_t)) + \mathbb{D}_f[\rho(x), p_T(x)]. \tag{21}$$

Notably, the optimization problem given by the right-hand-side of the abovementioned equation is also called Jordan-Kinderlehrer-Otto canonical form (Caluya & Halder, 2020; 2022; Jordan et al., 1998) or minimum movement scheme (Park et al., 2023). Before conducting the derivation, it is necessary to introduce the definition of Wasserstein distance. The squared 2-Wasserstein distance $\mathcal{W}_2^2$ can be defined by finding a transport map $\boldsymbol{T} : \mathbb{R}^D \to \mathbb{R}^D$ that minimizes the average cost of transporting mass from $\rho(x)$ to $\xi(x)$ as follows:

$$\mathcal{W}_2^2(\rho, \xi) = \inf_{\boldsymbol{T}: \boldsymbol{T}_{\#}\rho(x)=\xi(x)} \int \|x - \boldsymbol{T}(x)\|_2^2 \, \rho(x) \mathrm{d}x, \tag{22}$$

where $\boldsymbol{T}_{\#}$ indicates the pushforward measure, and the expression for $\boldsymbol{T}(x)$ is defined as follows:

$$\boldsymbol{T}(x) = x + \eta v_t(x). \tag{23}$$

Meanwhile, during the transportation, the differential equation that delineates PDF of the evolution process driven by Equation (1) is called *continuity equation* (Ambrosio et al., 2005; Villani et al., 2009), defined as follows:

$$\frac{\partial \rho(x_t)}{\partial t} = -\nabla \cdot [v_t(x_t)\rho(x_t)]. \tag{24}$$

Building on Equations (23) and (24), and discretizing the continuity equation in the time domain using the forward Euler scheme (Butcher, 2016; Evans, 2022), we obtain:

Taking the functional derivative of $\mathbb{D}_f[\rho(x), p_T(x)]$ with respect to $\rho(x)$, we have the following result:

$$
\begin{aligned}
&\frac{\mathrm{d}}{\mathrm{d}\eta} \mathbb{D}_f[\rho(x), p_T(x)] \\
&= \frac{\mathrm{d}}{\mathrm{d}\eta} \int p_T(x) f(\frac{\rho(x)}{p_T(x)}) \mathrm{d}x \\
&= \underbrace{\int \frac{\partial p_T(x)}{\partial \eta} f(\frac{\rho(x)}{p_T(x)}) \mathrm{d}x}_{=0} + \int p_T(x) \frac{\partial}{\partial \eta} f(\frac{\rho(x)}{p_T(x)}) \mathrm{d}x \\
&= \int \cancel{p_T(x)} \frac{1}{\cancel{p_T(x)}} [\frac{\partial}{\partial \rho(x)} f(\frac{\rho(x)}{p_T(x)})] \frac{\partial \rho(x)}{\partial \eta} \mathrm{d}x \\
&\overset{(i)}{=} \int [\frac{\partial}{\partial \rho(x)} f(\frac{\rho(x)}{p_T(x)})][-\nabla \cdot (v_t(x)\rho(x))] \mathrm{d}x \\
&\overset{(ii)}{=} \int [\rho(x)v_t(x)]^\top [\nabla \frac{\partial}{\partial \rho(x)} f(\frac{\rho(x)}{p_T(x)})] - \nabla \cdot [\frac{\partial}{\partial \rho(x)} f(\frac{\rho(x)}{p_T(x)}) v_t(x)\rho(x)] \mathrm{d}x \\
&\overset{(iii)}{=} \int [\rho(x)v_t(x)]^\top [\nabla \frac{\delta \mathbb{D}_f[\rho(x), p_T(x)]}{\delta \rho(x)}] - \nabla \cdot [\frac{\delta \mathbb{D}_f[\rho(x), p_T(x)]}{\delta \rho(x)} v_t(x)\rho(x)] \mathrm{d}x \\
&\overset{(iv)}{=} \int [\rho(x)v_t(x)]^\top [\nabla \frac{\partial}{\partial \rho(x)} f(\frac{\rho(x)}{p_T(x)})] \mathrm{d}x \\
&= \int \rho(x) v^\top(x) \nabla \frac{\delta \mathbb{D}_f[\rho(x), p_T(x)]}{\delta \rho(x)} \mathrm{d}x,
\end{aligned}
\tag{25}
$$

where '(i)' is based on Equation (24), '(ii)' is based on the following chain rule:

$$\nabla \cdot [\frac{\partial}{\partial \rho(x)} f(\frac{\rho(x)}{p_T(x)}) \rho(x) v_t(x)]$$

$$= [\rho(x)][v_t(x)]^\top [\nabla \frac{\partial}{\partial \rho(x)} f(\frac{\rho(x)}{p_T(x)})] + [\nabla \cdot v_t(x)][\rho(x)][\frac{\partial}{\partial \rho(x)} f(\frac{\rho(x)}{p_T(x)})] + [v_t(x)]^\top [\nabla \rho(x)][\frac{\partial}{\partial \rho(x)} f(\frac{\rho(x)}{p_T(x)})]$$

$$\Rightarrow$$

$$\nabla \cdot [\frac{\partial}{\partial \rho(x)} f(\frac{\rho(x)}{p_T(x)}) \rho(x) v_t(x)] - [\rho(x)][v_t(x)]^\top [\nabla \frac{\partial}{\partial \rho(x)} f(\frac{\rho(x)}{p_T(x)})] = [\nabla(v_t(x)\rho(x))][\frac{\partial}{\partial \rho(x)} f(\frac{\rho(x)}{p_T(x)})], \tag{26}$$

'(iii)' is based on the following equation:

$$\frac{\delta \mathbb{D}_f[\rho(x), p_T(x)]}{\delta \rho(x)} = \frac{\partial}{\partial \rho(x)} f(\frac{\rho(x)}{p_T(x)}), \tag{27}$$

and '(iv)' is based on the mild assumption on $\frac{\delta \mathbb{D}_f[\rho(x), p_T(x)]}{\delta \rho(x)} v(x) \rho(x)$ (Abraham et al., 2012; Johnson & Zhang, 2018; Liu et al., 2019; Shi et al., 2022; Zou et al., 2026), for example, rapid decay as $x \to \infty$, so that we have:

$$\int \nabla \cdot [\frac{\delta \mathbb{D}_f[\rho(x), p_T(x)]}{\delta \rho(x)} v_t(x) \rho(x)] \mathrm{d}x = 0. \tag{28}$$

Based on this, $\mathbb{D}_f[\rho(x), p_T(x)]$ can be expanded as follows when $\eta \to 0$:

$$\mathbb{D}_f[\rho(x), p_T(x)] = \mathbb{D}_f[p(x_t), p_T(x)] + \eta \int p(x_t) v^\top(x_t) \nabla \frac{\delta \mathbb{D}_f[p(x_t), p_T(x)]}{\delta p(x_t)} \mathrm{d}x. \tag{29}$$

For the squared 2-Wasserstein distance (Villani et al., 2009), we have the following inequality bound according to the Benamou-Brenier formulation (Ambrosio et al., 2021, Chapter 17):

$$\mathcal{W}_2^2(\rho(x), p(x_t)) = \int p(x_t) \|x - \boldsymbol{T}^*(x_t)\|_2^2 \mathrm{d}x = \eta^2 \int p(x_t) \|v_t^*(x_t)\|_2^2 \mathrm{d}x \le \eta^2 \int p(x_t) \|v_t(x_t)\|_2^2 \mathrm{d}x, \tag{30}$$

where $\boldsymbol{T}^*(x)$ and $v_t^*(x)$ are the optimal transportation map and optimal velocity field. Since $v_t(x)$ is not the optimal velocity filed, we obtain the last inequality. Based on Equations (29) and (30), we finally reach the following result:

$$\mathbb{D}_f[\rho(x), p_T(x)] + \frac{1}{2\eta} \mathcal{W}_2^2(\rho(x), p(x_t)) - \mathbb{D}_f[p(x_t), p_T(x)]$$

$$\le \cancel{\mathbb{D}_f[\rho(x), p_T(x)]} + \frac{\eta}{2} \mathbb{E}_{p(x_t)}[\|v_t(x_t)\|_2^2] + \eta \int \cdot [p(x_t) v_t^\top(x_t) \nabla \frac{\delta \mathbb{D}_f[p(x_t), p_T(x)]}{\delta p(x_t)}] \mathrm{d}x - \cancel{\mathbb{D}_f[\rho(x), p_T(x)]}$$

$$\le \frac{\eta}{2} \underbrace{\mathbb{E}_{p(x_t)}[\|\nabla \frac{\delta \mathbb{D}_f[p(x_t), p_T(x)]}{\delta p(x_t)}]\|_2^2]}_{\ge 0} + \frac{\eta}{2} \mathbb{E}_{p(x_t)}[\|v_t(x_t)\|_2^2] + \eta \int p(x_t) v_t^\top(x_t) \nabla \frac{\delta \mathbb{D}_f[p(x_t), p_T(x)]}{\delta p(x_t)} \mathrm{d}x \tag{31}$$

$$= \frac{\eta}{2} \mathbb{E}_{p(x_t)}\{\|v_t(x_t) + \nabla \frac{\delta \mathbb{D}_f[p(x_t), p_T(x)]}{\delta p(x_t)}\|_2^2\}.$$

Consequently, the optimal velocity field that reduces the upper bound of the optimization problem defined by the right-hand-side of Equation (4) can be given as follows:

$$v_t^*(x_t) = -\nabla \frac{\delta \mathbb{D}_f[p(x_t), p_T(x)]}{\delta p(x_t)}, \tag{32}$$

which implies that the left-hand side of Equation (21) is a sufficient condition for the optimality of its right-hand side.

## C.2. Derivation of Proposition 3.1

*Proposition* (3.1). Consider the following primal problem:

$$\mathcal{L}^{\text{Primal}} = \underset{\rho(x)\in\mathcal{P}_2(\mathbb{R}^D)}{\arg\min} \frac{1}{2\eta} \mathcal{W}_2^2(\rho(x), p(x_t)) + \mathbb{D}_f[\rho(x), p_T(x)]. \tag{33}$$

This problem is equivalent to the following semi-dual formulation:

$$\mathcal{L}^{\text{SemiDual}} = \sup_w \mathbb{E}_{p(x_t)} \left[ \inf_{\boldsymbol{T}} \left( \|\boldsymbol{T}(x_t) - x_t\|_2^2 - w(\boldsymbol{T}(x_t)) \right) \right] - \mathbb{E}_{p_T(x)}[f^\star(-w(x))], \tag{34}$$

where $w : \mathbb{R}^D \to \mathbb{R}$ is a measurable continuous function, $\boldsymbol{T} : \mathbb{R}^D \to \mathbb{R}^D$ is the transport map, and $f^\star := \sup_{y\geq0}\{zy - f(y)\}$ denotes the convex conjugate of $f$.

*Proof.* Equation (33) can be reformulated as follows:

$$\inf_{\pi\in\mathbb{R}_+^{D\times D}} \frac{1}{2\eta} \iint \|x_t - x\|_2^2 \pi(x_t, x)\mathrm{d}x_t\mathrm{d}x + \int f(\frac{\rho(x)}{p_T(x)})p_T(x)\mathrm{d}x, \tag{35a}$$

$$\text{s.t.} \quad p(x_t) = \int \pi(x_t, x)\mathrm{d}x, \quad \rho(x) = \int \pi(x_t, x)\mathrm{d}x_t. \tag{35b}$$

Based on this, we introduce the Lagrange multiplier (Biegler, 2010; Boyd & Vandenberghe, 2004) $u(x_t)$ and $w(x)$ to handle the equality constraints given by Equation (35b) as follows:

$$
\begin{aligned}
\mathcal{L} =& \frac{1}{2\eta} \iint \|x_t - x\|_2^2 \pi(x_t, x)\mathrm{d}x_t\mathrm{d}x + \int f(\frac{\rho(x)}{p_T(x)})p_T(x)\mathrm{d}x \\
&+ \int u(x_t)[p(x_t) - \int \pi(x_t, x)\mathrm{d}x]\mathrm{d}x_t + \int w(x)[\rho(x) - \int \pi(x_t, x)\mathrm{d}x_t]\mathrm{d}x \\
=& \iint [\frac{1}{2\eta}\|x_t - x\|_2^2 - u(x_t) - w(x)]\pi(x_t, x)\mathrm{d}x_t\mathrm{d}x + \int u(x_t)p(x_t)\mathrm{d}x_t + \int w(x)\rho(x) + f(\frac{\rho(x)}{p_T(x)})p_T(x)\mathrm{d}x.
\end{aligned}
\tag{36}
$$

As a result, the following dual function can be given as follows due to the linear independent structure of problem defined by Equation (36):

$$
\begin{aligned}
&g(u, w) \\
&= \inf_\pi \left\{ \iint [\frac{1}{2\eta}\|x_t - x\|_2^2 - u(x_t) - w(x)]\pi(x_t, x)\mathrm{d}x_t\mathrm{d}x + \int u(x_t)p(x_t)\mathrm{d}x_t \right\} \\
&\quad + \inf_\rho \left\{ \int [w(x)\frac{\rho(x)}{p_T(x)} + f(\frac{\rho(x)}{p_T(x)})]p_T(x)\mathrm{d}x \right\} \\
&= \inf_\pi \left\{ \iint [\frac{1}{2\eta}\|x_t - x\|_2^2 - u(x_t) - w(x)]\pi(x_t, x)\mathrm{d}x_t\mathrm{d}x + \int u(x_t)p(x_t)\mathrm{d}x_t \right\} - \int p_T(x) f^\star(-w(x)) \,\mathrm{d}x,
\end{aligned}
\tag{37}
$$

where the last line uses the Legendre–Fenchel conjugate (Caluya & Halder, 2020; Shi et al., 2022; Touchette, 2005). Specifically, without loss of generality, we define the density ratio function $r(x) := \frac{\rho(x)}{p_T(x)}$. On this basis, since $p_T(x) > 0$ always holds, we have the following results based on the definition of conjugate function for $f$:

$$
\begin{aligned}
&\inf_\rho \int [w(x)\frac{\rho(x)}{p_T(x)} + f(\frac{\rho(x)}{p_T(x)})]p_T(x)\mathrm{d}x \\
&= \inf_r \int [w(x)r(x) + f(r(x))]p_T(x)\mathrm{d}x \\
&= \int \inf_r [w(x)r(x) + f(r(x))]p_T(x)\mathrm{d}x \\
&= -\int \sup_r [-w(x)r(x) - f(r(x))]p_T(x)\mathrm{d}x \\
&= -\int p_T(x) f^\star(-w(x)) \,\mathrm{d}x.
\end{aligned}
\tag{38}
$$

Suppose that $\frac{1}{2\eta}\|x_t - x\|_2^2 - u(x_t) - w(x) < 0$ for some pair $(x_t, x)$. In this case, concentrating all the mass of $\pi(x_t, x)$ at this point drives the Lagrangian in Equation (37) to $-\infty$. To avoid such degenerate solutions, it is necessary to impose the condition $\frac{1}{2\eta}\|x_t - x\|_2^2 - u(x_t) - w(x) \geq 0$ almost everywhere. On this basis, the dual problem can be reformulated as

$$\sup_{u,w} \left\{ \int u(x_t)\, p(x_t)\, dx_t - \int p_T(x)\, f^\star(-w(x)) dx \right\}, \quad \text{s.t.} \quad u(x_t) + w(x) \leq \frac{1}{2\eta}\|x_t - x\|_2^2. \tag{39}$$

Equivalently, we can introduce a convex indicator function $\ell(\cdot)$ to incorporate the constraint into the objective. Specifically, we define the convex indicator $\ell$ as follows:

$$\ell[u(x_t) + w(x) \leq \frac{1}{2\eta}\|x_t - x\|_2^2] = \begin{cases} 0, & u(x_t) + w(x) \leq \frac{1}{2\eta}\|x_t - x\|_2^2, \\ +\infty, & \text{otherwise}, \end{cases}$$

so that infeasible pairs are ruled out. After that, Equation (39) can be equivalently written as follows:

$$\sup_{u,w} \left\{ \int u(x_t)p(x_t)\, dx_t - \int p_T(x)f^\star(-w(x))\, dx - \ell[u(x_t) + w(x) \leq \frac{1}{2\eta}\|x_t - x\|_2^2] \right\}. \tag{40}$$

To justify the ensuing strong duality, note that if the inequality constraint is violated at some $(x_t, x)$, then by concentrating the coupling mass at such a point the corresponding primal value becomes unbounded from below, i.e., the dual feasible set must enforce $u(x_t) + w(x) \leq \frac{1}{2\eta}\|x_t - x\|_2^2$ $\pi$-almost everywhere for admissible optimal couplings. In addition, since $\frac{1}{2\eta}\|x_t - x\|^2 \geq 0$, there exists a feasible pair $(u, w) = (-1, -1)$ such that the constraint qualification of Fenchel–Rockafellar theorem (Bauschke & Combettes, 2017) holds (the feasible set is non-empty and the corresponding infimal convolution is well-posed). Therefore, strong duality follows.

Moreover, by complementary slackness, the inequality in (39) is tight on the support of the optimal coupling $\pi^*$. In other words, the slack $\frac{1}{2\eta}\|x_t - x\|_2^2 - u^*(x_t) - w^*(x)$ is $\pi^*$-almost surely zero, which yields the following result:

$$\frac{1}{2\eta}\|x_t - x\|_2^2 = u^*(x_t) + w^*(x) \qquad \pi^*\text{-almost everywhere}. \tag{41}$$

Combining Equation (41) with the feasibility condition $u^*(x_t) + w^*(x) \leq \frac{1}{2\eta}\|x_t - x\|_2^2$, we have:

$$u^*(x_t) \leq \frac{1}{2\eta}\|x_t - x\|_2^2 - w^*(x),$$

and the tightness condition ensures equality is attained on the $\pi^*$-support. Therefore, we arrive at the following result:

$$u^*(x_t) = \inf_x \left\{ \frac{1}{2\eta}\|x_t - x\|_2^2 - w^*(x) \right\}. \tag{42}$$

Substituting Equation (42) into Equation (37), we obtain the semi-dual formulation:

$$\sup_w \left\{ \int \inf_x \left[ \frac{1}{2\eta}\|x_t - x\|_2^2 - w(x) \right] p(x_t)\, dx_t - \int p_T(x)f^\star(-w(x))\, dx \right\}. \tag{43}$$

Defining the transport map $\boldsymbol{T}$ via the $c$-transform as follows:

$$\boldsymbol{T}^*(x_t) \in \arg\min_x \left\{ \frac{1}{2\eta}\|x_t - x\|_2^2 - w(x) \right\},$$

we can reformulate the inner problem of Equation (43) as follows:

$$\inf_x \left\{ \frac{1}{2\eta}\|x_t - x\|_2^2 - w(x) \right\} = \frac{1}{2\eta}\|x_t - \boldsymbol{T}^*(x_t)\|_2^2 - w(\boldsymbol{T}^*(x_t)). \tag{44}$$

As such, substituting Equation (44) into Equation (43), we obtain the final semi-dual objective as follows:

$$\mathcal{L}^{\text{SemiDual}} = \sup_w \mathbb{E}_{p(x_t)}[\|\frac{1}{2\eta}\boldsymbol{T}^*(x_t) - x_t\|_2^2 - w(\boldsymbol{T}^*(x_t))] - \mathbb{E}_{p_T(x)}[f^\star(-w(x))], \tag{45}$$

It should be pointed out that there is no closed-form expression of the optimal $\boldsymbol{T}^*(x_t)$ for each $w(x)$ (Choi et al., 2023; Korotin et al., 2023). Consequently, the optimization $\boldsymbol{T}(x_t)$ for each $w(x)$ is required, and we finally reach the final semi-dual objective as follows based on Equation (45):

$$\mathcal{L}^{\text{SemiDual}} = \sup_w \mathbb{E}_{p(x_t)}\{\inf_{\boldsymbol{T}}[\frac{1}{2\eta}\|\boldsymbol{T}(x_t) - x_t\|_2^2 - w(\boldsymbol{T}(x_t))]\} - \mathbb{E}_{p_T(x)}[f^\star(-w(x))].$$

$\square$

### C.3. Derivation of Proposition 3.2

*Proposition* (3.2). Consider the semi-dual objective in Equation (7) and assume it admits at least one optimizer $w$. There exist $a \in \mathbb{R}^D$ and two points $b_1, b_2 \in \mathbb{R}^D$ such that $\|a - b_1\|_2 = \|a - b_2\|_2$. Let $p_\gamma(x_t) = \mathcal{N}(a, \gamma I)$ and $p_{T,\gamma}(x) = \frac{1}{2}\mathcal{N}(b_1, \gamma I) + \frac{1}{2}\mathcal{N}(b_2, \gamma I)$. Then, for all sufficiently small $\gamma > 0$, there exists a semi-dual optimizer $w_\gamma^*$ and a point $x_t^\gamma \in \mathbb{R}^D$ (with $x_t^\gamma \to a$ as $\gamma \to 0$) such that $\arg\min_{x \in \mathbb{R}^D}\{c(x_t^\gamma, x) - w_\gamma^*(x)\}$ is not a singleton. Equivalently, the inner $c$-transform (and thus $\boldsymbol{T}^*(x_t^\gamma)$) is non-unique.

*Proof.* Let us first consider the discrete case. Fix $a, b_1, b_2$ with $r := \|a - b_1\|_2 = \|a - b_2\|_2$ and consider the following Dirac delta measure $\delta$:

$$p = \delta_a, \qquad \rho = \frac{1}{2}\delta_{b_1} + \frac{1}{2}\delta_{b_2}.$$

The discrete inner problem reduces to $\min\{c(a, b_1) - w(b_1), c(a, b_2) - w(b_2)\}$ up to constants. Let $S$ be the exchange map swapping $b_1 \leftrightarrow b_2$. Since $c(a, b_1) = c(a, b_2)$ and $\rho(b_1) = \rho(b_2)$, the semi-dual is invariant under exchanging $b_1$ and $b_2$. Therefore, for any optimizer $w^*$, the symmetrized potential

$$\bar{w}^*(x) := \frac{1}{2}[w^*(x) + w^*(Sx)]$$

is also an optimizer.

In particular, $\bar{w}^*(b_1) = \bar{w}^*(b_2)$. Hence, we have $c(a, b_1) - \bar{w}^*(b_1) = c(a, b_2) - \bar{w}^*(b_2)$, so both $b_1$ and $b_2$ are minimizers of the inner objective. Thus the discrete inner $\arg\min$ is not a singleton.

Now consider the continuous case and define

$$p_\gamma(x_t) = \mathcal{N}(a, \gamma I), \qquad p_{T,\gamma}(x) = \frac{1}{2}\mathcal{N}(b_1, \gamma I) + \frac{1}{2}\mathcal{N}(b_2, \gamma I).$$

Let $x_t^\gamma \sim p_\gamma$ and let $w_\gamma^*$ be a semi-dual optimizer. Since $p_\gamma, p_{T,\gamma}$ converge weakly to the discrete measures as $\gamma \to 0$, the expected semi-dual objective is a continuous perturbation of the discrete one when $w$ is continuous. Moreover, for a fixed $w$, the function $x \mapsto c(x_t^\gamma, x) - w(x)$ is continuous in $x$, so its minimizers cannot disappear abruptly under the small perturbations induced by $\gamma$: there exist small neighborhoods $U_1$ and $U_2$ of $b_1$ and $b_2$ such that, for all sufficiently small $\gamma$, every global minimizer must lie in $U_1 \cup U_2$ and the two symmetric wells remain nearly tied.

To make this explicit, take the symmetric discrete optimizer $\bar{w}^*$ from above and build a smooth (e.g., mollified) approximation $w_\gamma^*$ that remains approximately symmetric, so that $w_\gamma^*(b_1) \approx w_\gamma^*(b_2)$. Then for $x_t^\gamma$ close to $a$, the inner values at $x \approx b_1$ and $x \approx b_2$ stay approximately equal, implying that the inner minimizer set contains at least one point in $U_1$ and at least one point in $U_2$. Therefore, $\arg\min_x\{c(x_t^\gamma, x) - w_\gamma^*(x)\}$ contains at least two minimizers. Hence the inner $c$-transform is non-unique, which implies that $\boldsymbol{T}^*(x_t^\gamma)$ is non-unique. $\square$

### C.4. Analysis and Discussions on the Entropic Problem

#### C.4.1. DERIVATION OF PROPOSITION 3.3

*Proposition* (3.3). Let $\kappa(x_t, x) := p(x_t)p_T(x)$ denote the reference joint PDF. The entropy-regularized primal problem is

$$\mathcal{L}^{\text{E-Primal}} = \arg\min_{\rho \in \mathcal{P}_2(\mathbb{R}^D)} \frac{1}{2\eta}\mathcal{W}_2^2(\rho(x), p(x_t)) + \mathbb{D}_f[\rho(x), p_T(x)] + \epsilon \iint \pi(x_t, x)[\log\frac{\pi(x_t, x)}{\kappa(x_t, x)}]dx_t dx, \tag{46}$$

and is equivalent to the semi-dual optimization problem

$$\mathcal{L}^{\text{E-SemiDual}} = \sup_w - \epsilon \, \mathbb{E}_{p(x_t)} [\log \mathbb{E}_{p_T(x)} (\exp(\frac{w(x) - \frac{1}{2\eta} \| x - x_t \|_2^2}{\epsilon}))] - \mathbb{E}_{p_T(x)} [f^\star(-w(x))], \tag{47}$$

where $f^\star$ denotes the convex conjugate of $f$.

*Proof.* Without loss of generality, we first define the quadratic function $c(x_t, x)$ as follows to facilitate derivation:

$$c(x_t, x) := \frac{1}{2\eta} \| x_t - x \|_2^2. \tag{48}$$

Based on this, Equation (46) can be reformulated as follows:

$$\underset{\rho, \pi}{\arg\min} \iint c(x_t, x) \pi(x_t, x) \mathrm{d}x \mathrm{d}x_t + \int f(\frac{\rho(x)}{p_T(x)}) p_T(x) \mathrm{d}x + \epsilon \iint \pi(x_t, x) \log \frac{\pi(x_t, x)}{\kappa(x_t, x)} \mathrm{d}x_t \mathrm{d}x,$$

$$\overset{(i)}{\Rightarrow}$$

$$\underset{\rho, \pi}{\arg\min} \iint c(x_t, x) \pi(x_t, x) \mathrm{d}x \mathrm{d}x_t + \int f(\frac{\rho(x)}{p_T(x)}) p_T(x) \mathrm{d}x + \epsilon \iint \pi(x_t, x)[\log \frac{\pi(x_t, x)}{\kappa(x_t, x)} - 1] \mathrm{d}x_t \mathrm{d}x,$$

$$\overset{(ii)}{\Rightarrow}$$

$$\underset{\rho, \pi}{\arg\min} \iint c(x_t, x) \pi(x_t, x) \mathrm{d}x \mathrm{d}x_t + \int f(\frac{\rho(x)}{p_T(x)}) p_T(x) \mathrm{d}x + \epsilon \iint \pi(x_t, x)[\log \frac{\pi(x_t, x)}{\kappa(x_t, x)} - 1] \mathrm{d}x_t \mathrm{d}x \tag{49}$$

$$+ \int u(x_t)[p(x_t) - \int \pi(x_t, x) \mathrm{d}x] \mathrm{d}x_t + \int w(x)[\rho(x) - \int \pi(x_t, x) \mathrm{d}x_t] \mathrm{d}x,$$

$$\Rightarrow$$

$$\underset{\rho, \pi}{\arg\min} \iint [c(x_t, x) - u(x_t) - w(x) + \epsilon \log \frac{\pi(x_t, x)}{\kappa(x_t, x)} - \epsilon] \pi(x_t, x) \mathrm{d}x \mathrm{d}x_t$$

$$+ \int u(x_t) p(x_t) \mathrm{d}x_t + \int [w(x)\rho(x) + f(\frac{\rho(x)}{p_T(x)}) p_T(x)] \mathrm{d}x,$$

where '(i)' is based on the following constraint for transportation plan $\pi(x_t, x)$:

$$\iint \pi(x_t, x) \mathrm{d}x \mathrm{d}x_t = 1,$$

'(ii)' is the Lagrangian of Equation (46), $u : \mathbb{R}^D \to \mathbb{R}$ is the Lagrange multiplier for margin $p(x_t)$, and $v : \mathbb{R}^D \to \mathbb{R}$ is the Lagrange multiplier for margin $\rho(x)$.

To facilitate analysis, we define Lagrangian $\mathcal{L}$ as follows:

$$\mathcal{L} := \iint [c(x_t, x) - u(x_t) - w(x) + \epsilon \log \frac{\pi(x_t, x)}{\kappa(x_t, x)} - \epsilon] \pi(x_t, x) \mathrm{d}x \mathrm{d}x_t$$

$$+ \int u(x_t) p(x_t) \mathrm{d}x_t + \int [w(x)\rho(x) + f(\frac{\rho(x)}{p_T(x)}) p_T(x)] \mathrm{d}x. \tag{50}$$

For $\mathcal{L}$, we have the following results for the optimal transportation plan $\pi^*(x_t, x)$:

$$\frac{\delta \mathcal{L}}{\delta \pi} = [c(x_t, x) - u(x_t) - w(x) + \epsilon \log \frac{\pi(x_t, x)}{\kappa(x_t, x)} - \epsilon] + \epsilon = 0 \Rightarrow \pi^*(x_t, x) = \kappa(x_t, x) \exp(-\frac{c(x_t, x) - u(x_t) - w(x)}{\epsilon}). \tag{51}$$

Consequently, we can define the objective function for $u$ and $w$ according to Equations (38), (49) and (51):

$$g(u, w) := -\epsilon \iint \kappa(x_t, x) \exp(-\frac{c(x_t, x) - u(x_t) - w(x)}{\epsilon}) \mathrm{d}x_t \mathrm{d}x + \int u(x_t) p(x_t) \mathrm{d}x_t - \int p_T(x) f^\star(-w(x)) \, \mathrm{d}x. \tag{52}$$

On this basis, $u(x_t)$ and $w(x)$ are determined by solving the following duality problem:

$$u^*, w^* = \sup_{u,w} \; g(u,w). \tag{53}$$

To facilitate derivation, we define $Z(x_t)$ as follows:

$$Z(x_t) := \mathbb{E}_{p_T(x)}[\exp(\frac{w(x) - c(x_t, x)}{\epsilon})] = \int p_T(x) \exp(\frac{w(x) - c(x_t, x)}{\epsilon}) \mathrm{d}x. \tag{54}$$

On this basis, $g(u,w)$ can be reformulated as follows:

$$g(u,w) = -\epsilon \int p(x_t) \exp(\frac{u(x_t)}{\epsilon}) Z(x_t) \mathrm{d}x_t + \int u(x_t) p(x_t) \mathrm{d}x_t - \int p_T(x) f^\star(-w(x)) \, \mathrm{d}x. \tag{55}$$

For $u^*(x_t)$, we have:

$$u^*(x_t) = \sup_u \int p(x_t)[-\epsilon \exp(\frac{u(x_t)}{\epsilon}) Z(x_t) + u(x_t)] \Rightarrow -\exp(\frac{u(x_t)}{\epsilon}) Z(x_t) + 1 = 0 \Rightarrow u^*(x_t) = -\epsilon \log Z(x_t). \tag{56}$$

As such, $g(u,w)$ can be further reformulated as follows:

$$\begin{aligned}
g(u,w) &= -\epsilon \underbrace{\int p(x_t) \cancel{\exp[-\log Z(x_t)]} \cancel{Z(x_t)} \mathrm{d}x_t}_{=1} + \int [-\epsilon \log Z(x_t)] p(x_t) \mathrm{d}x_t - \int p_T(x) f^\star(-w(x)) \mathrm{d}x, \\
\Rightarrow g(u,w) &= -\epsilon \int [\log \mathbb{E}_{p_T(x)}(\exp(\frac{w(x) - c(x_t, x)}{\epsilon}))] p(x_t) \mathrm{d}x_t - \int p_T(x) f^\star(-w(x)) \mathrm{d}x, \\
\Rightarrow g(u,w) &= -\epsilon \mathbb{E}_{p(x_t)}[\log \mathbb{E}_{p_T(x)}(\exp(\frac{w(x) - c(x_t, x)}{\epsilon}))] - \int p_T(x) f^\star(-w(x)) \mathrm{d}x, \\
\Rightarrow g(u,w) &= -\epsilon \mathbb{E}_{p(x_t)}[\log \mathbb{E}_{p_T(x)}(\exp(\frac{w(x) - \frac{1}{2\eta}\|x - x_t\|_2^2}{\epsilon}))] - \mathbb{E}_{p_T(x)}[f^\star(-w(x))].
\end{aligned} \tag{57}$$

By taking the supremum to the last line of Equation (57), we obtain the entropic semi-dual objective function, denoted as $\mathcal{L}^{\text{E-SemiDual}}$, which is defined in Equation (47). $\qquad\square$

### C.4.2. COMPARISON OF PRIMAL AND DUAL FORMULATIONS

In this part, we perform a comparative experiment to investigate the relationship between different primal and dual formulations. To this end, we consider a two-dimensional toy example consisting of two Gaussian clusters. The source samples are generated from a Gaussian mixture distribution, whereas the target samples are obtained by rotating the source distribution, enlarging the variance of one target component, and applying a global translation. The source and target samples are illustrated in Figure 4. Based on this toy example, we consider the following entropically regularized OT problem:

$$\inf_\pi \iint \pi^\top(x_t, x_s) \|x_s - x_t\|_2^2 \mathrm{d}x_s \mathrm{d}x_t + \epsilon \iint \pi_{ij} (\log \pi_{ij} - 1) \mathrm{d}x_s \mathrm{d}x_t, \tag{58a}$$

$$\text{s.t.} \quad \int \pi(x_t, x_s) \mathrm{d}x_s = p(x_t), \qquad \int \pi(x_t, x_s) \mathrm{d}x_t = p(x_s). \tag{58b}$$

Notably, the term $\iint \pi(x_t, x_s) \log[p(x_s)p(x_t)] \, \mathrm{d}x_s \mathrm{d}x_t$ is omitted from the objective in Equation (58a). Since the marginals $p(x_s)$ and $p(x_t)$ are fixed, this term is constant with respect to the transport plan $\pi$ and therefore does not affect the optimization. In practice, we solve the discrete empirical counterpart in Equations (59a) and (59b), where $n$ denotes the empirical sample size.

$$\inf_\pi \quad \sum_{i=1}^n \sum_{j=1}^n \pi_{ij} \|x_s^{(i)} - x_t^{(j)}\|_2^2 + \epsilon \sum_{i=1}^n \sum_{j=1}^n \pi_{ij} (\log \pi_{ij} - 1), \tag{59a}$$

$$\text{s.t.} \quad \sum_{j=1}^n \pi_{ij} = \frac{1}{n}, \quad \forall i \in \{1, \ldots, n\}, \qquad \sum_{i=1}^n \pi_{ij} = \frac{1}{n}, \qquad j \in \{1, \ldots, n\}. \tag{59b}$$

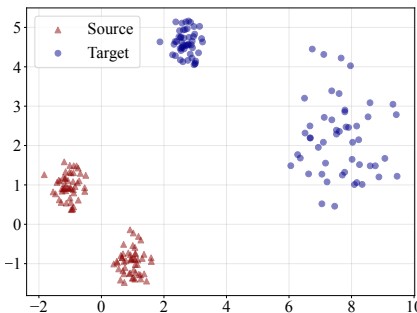

*Figure 4.* The illustration for the source and target samples.

We compare the solutions of the entropic OT problem in Equations (59a) and (59b) obtained from primal and dual perspectives. For the primal formulation, we employ the Sinkhorn iteration, as summarized in Algorithm 2, following Cuturi (2013), where $\oslash$ denotes element-wise division. In Algorithm 2, we set $\mathcal{E}$ as 1000. For the dual formulation, we directly optimize the dual objective in Equation (60) using the Adam optimizer (Kingma & Ba, 2015). To assess the quality of the primal and dual solutions, we take the Earth Mover's Distance (EMD) solution as the ground-truth reference. The solution accuracy and computational time across different sample sizes are summarized in Table 6. Furthermore, the convergence behavior of the dual optimization, measured by the iteration loss under different sample sizes, is illustrated in Figure 5.

---

**Algorithm 2** Sinkhorn Iteration for Entropic Optimal Transport

---

**Input:** Source samples $\{x_s^{(i)}\}_{i=1}^n$, target samples $\{x_t^{(j)}\}_{j=1}^n$, marginal weights $a \in [\frac{1}{n}, \ldots, \frac{1}{n}]^\top$ and $b \in [\frac{1}{n}, \ldots, \frac{1}{n}]^\top$, regularization parameter $\epsilon > 0$, maximum iterations $\mathcal{E}$.
**Output:** Transportation plan $\pi$.

1: Construct the Gibbs kernel: $\mathcal{K}(x_s, x_t) \leftarrow \exp(\frac{\|x_s - x_t\|_2^2}{\epsilon})$

2: Initialize $v^{(0)} \leftarrow \mathbf{1}_{n_t}$ and $u^{(0)} \leftarrow \mathbf{1}_{n_t}$.
3: **for** $t = 0, 1, \ldots, \mathcal{E} - 1$ **do**
4:    $u_{t+1}(x_s) = a \oslash [\mathcal{K}(x_s, x_t) w_t(x_t)]$.
5:    $w_{t+1}(x_t) = b \oslash [\mathcal{K}^\top(x_s, x_t) u_{t+1}(x_s)]$.
6: **end for**
7: $\pi^\epsilon \leftarrow \mathrm{diag}[u_\mathcal{E}(x_s)] \mathcal{K} \mathrm{diag}[w_\mathcal{E}(x_t)]$.

---

$$\underset{u,v}{\arg\max} \, \mathcal{L}^{\text{Dual}}(u, v) = \frac{1}{n} \sum_{i=1}^n u^{(i)}(x_s) + \frac{1}{n} \sum_{j=1}^n w^{(j)}(x_t) - \epsilon \frac{1}{n \times n} \sum_{i=1}^n \sum_{j=1}^n \exp(\frac{u^{(i)}(x_s) + w^{(j)}(x_t) - \|x_s^{(i)} - x_t^{(j)}\|_2^2}{\epsilon}).$$

(60)

From Table 6, we observe that the computational time for the dual strategy is slightly higher than that for the primal strategy. This is expected because the primal problem is solved by Sinkhorn iteration, which is specifically designed for entropically regularized OT. By exploiting the scaling form of the optimal coupling, Sinkhorn iteration only requires alternating matrix-vector products and element-wise divisions. By contrast, the dual strategy is optimized with Adam, a general-purpose gradient-based optimizer, which involves gradient computation and may require more iterations to achieve stable convergence. Meanwhile, the OT cost obtained by the dual strategy is lower than that obtained by the primal strategy. Although the primal and dual formulations are theoretically equivalent under strong duality, their practical solutions differ due to numerical precision and optimization effects. In the primal approach, the Sinkhorn algorithm operates directly on the Gibbs kernel $\mathcal{K} = \exp(\frac{\|x_s - x_t\|_2^2}{\epsilon})$, and each iteration involves repeated element-wise multiplication and division of exponentially scaled quantities. When the regularization parameter $\epsilon$ is small relative to the entries of the cost matrix $C$, the kernel entries $\mathcal{K}_{ij} = \exp(\frac{\|x_s^{(i)} - x_t^{(j)}\|_2^2}{\epsilon})$ approach zero and suffer from floating-point underflow, which prevents the marginal constraints from being exactly satisfied at termination. Consequently, the Sinkhorn algorithm terminates with a

small but nonzero marginal violation, which can introduce bias into the recovered transport plan and inflate the resulting OT cost. In contrast, the dual approach directly optimizes the dual potentials $u$ and $v$ using optimizer, updating them in an additive manner that avoids the multiplicative scaling instability inherent in Sinkhorn iteration. Although exponential terms still appear in the dual objective, the optimization operates in the non-exponentiated space of the potentials, which is numerically better conditioned. As a result, the dual strategy can potentially achieve a more accurate and numerically stable solution, which may explain the lower OT cost observed in Table 6.

*Table 6.* Comparison of Transport Plans: Sinkhorn, Dual Solver, and Exact Earth Moving Distance

| $n$ | Primal Time (ms) | Dual Time (ms) | Marginal Err (Primal) | Marginal Err (Dual) | Cost (Primal) | Cost (Dual) | Cost (EMD) | $\Delta_{\text{EMD}}$ (Primal) | $\Delta_{\text{EMD}}$ (Dual) |
|---|---|---|---|---|---|---|---|---|---|
| 50 | $0.98 \pm 0.08$ | $1014.7 \pm 69.1$ | $2.11\times10^{-18}$ | $7.54\times10^{-6}$ | 0.3971 | 0.3381 | 0.3350 | 0.1398 | 0.1343 |
| 100 | $2.95 \pm 2.08$ | $1043.9 \pm 65.5$ | $1.24\times10^{-18}$ | $3.42\times10^{-6}$ | 0.3667 | 0.3109 | 0.3078 | 0.0994 | 0.0972 |
| 200 | $9.12 \pm 5.23$ | $1005.0 \pm 80.5$ | $6.36\times10^{-19}$ | $1.62\times10^{-6}$ | 0.3224 | 0.2734 | 0.2706 | 0.0705 | 0.0696 |
| 500 | $25.50 \pm 17.38$ | $1068.1 \pm 44.7$ | $2.28\times10^{-19}$ | $5.53\times10^{-7}$ | 0.3298 | 0.2801 | 0.2770 | 0.0447 | 0.0444 |

*Kindly Note:* "Marginal Err." is short for "Marginal Error," and "Sink." is short for "Sinkhorn." $\Delta$ indicates the relative performance change of each solution strategy with respect to the EMD solver.

Additionally, from Figure 5, we observe that the negative dual loss consistently decreases as the number of iterations increases and gradually reaches a plateau. This indicates that the dual objective converges stably across different sample sizes. Moreover, the curves tend to plateau at similar levels, suggesting that the dual optimization nearly saturates and reaches a comparable stationary objective value under different sample sizes. The shaded regions, which represent $\pm 1.0$ standard deviation over multiple runs, remain relatively narrow, suggesting that the optimization process is robust to random initialization. In summary, although larger sample sizes introduce a slightly higher computational burden, the overall convergence pattern and final objective level remain similar for all values of $n$, further demonstrating the stability and scalability of the dual optimization strategy for constructing OT plan.

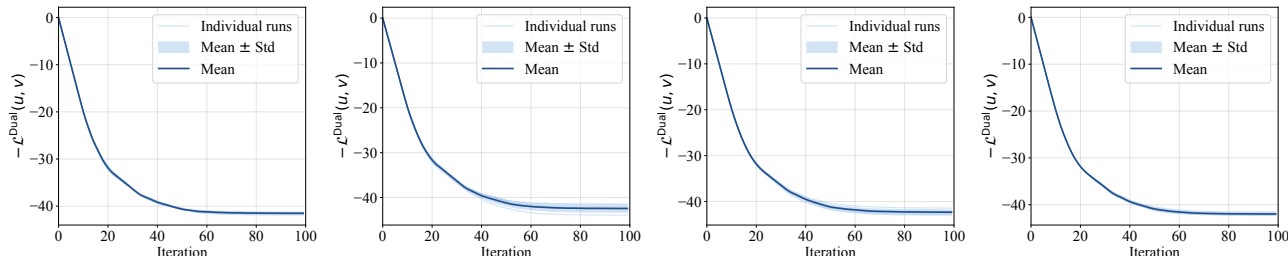

(a) Negative dual loss, $n = 50$. (b) Negative dual loss, $n = 100$. (c) Negative dual loss, $n = 200$. (d) Negative dual loss, $n = 500$.

*Figure 5.* Convergence of the negative dual loss over iterations under different sample sizes. Shaded regions indicate $\pm 1.0$ standard deviation across runs.

### C.4.3. EFFECT OF ENTROPY REGULARIZATION STRENGTH

In this section, we provide a more systematic discussion of the influence of the entropic regularization strength on the learned transport plan. Specifically, we consider a two-dimensional toy example similar to that in Section C.4.2. We generate a total of 100 samples for this experiment.

The corresponding experimental results are presented in Figure 6. As shown in the figure, the entropy regularization strength has a clear influence on the structure of the transport plan. When $\epsilon$ is small, as demonstrated in Figures 6(b) and 6(c), the transport plan is relatively concentrated and mainly assigns mass to low-cost source-target pairs, resulting in a sharper and sparser matching pattern. As $\epsilon$ increases, as demonstrated in Figures 6(d) to 6(f), the entropy term encourages the plan to avoid zero entries, and the resulting coupling becomes denser, smoother, and more diffuse. This behavior is consistent with the Sinkhorn form of the optimal plan, i.e., $\pi = \text{diag}[u(x_t)] \exp(-\frac{\|x_t - x_s\|_2^2}{\epsilon})\text{diag}[v(x_s)]$, where a larger $\epsilon$ weakens the sensitivity to the transport cost $\|x_t - x_s\|_2^2$ and spreads the mass over more feasible paths. Therefore, increasing $\epsilon$ leads to a smoother transport plan, while decreasing $\epsilon$ makes the solution closer to the unregularized optimal transport plan.

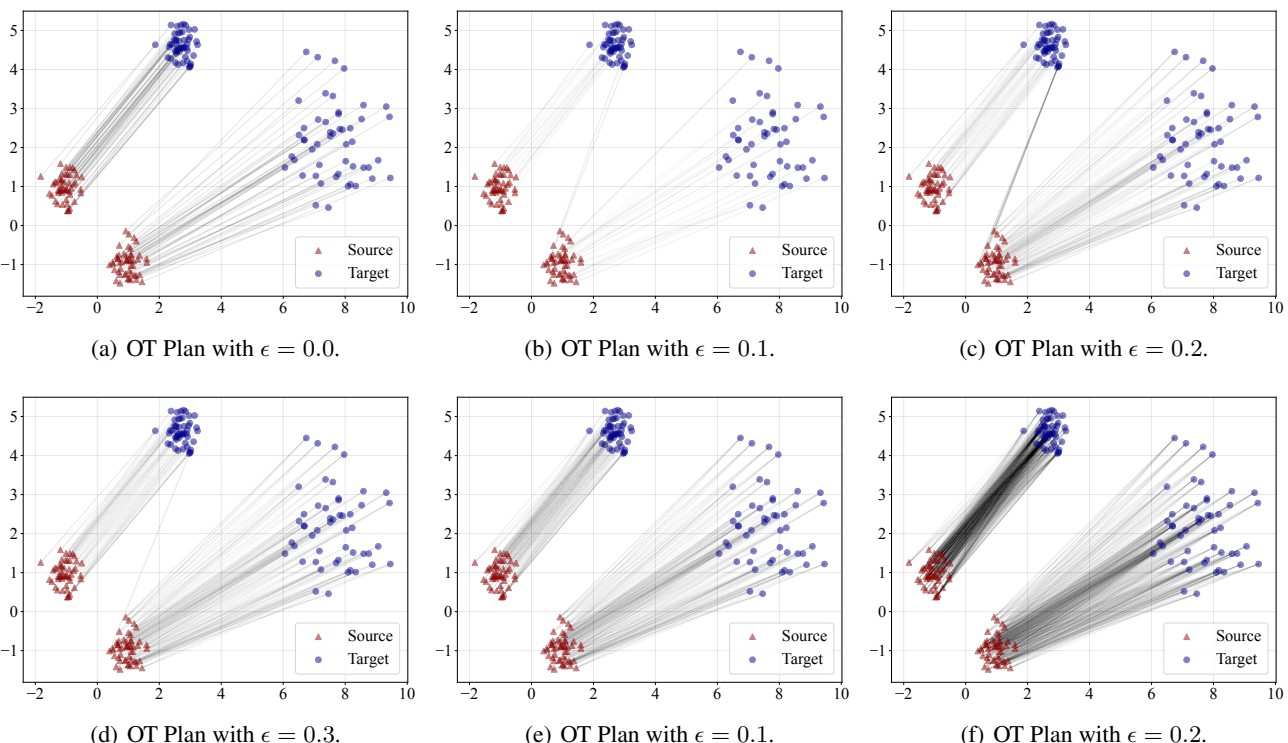

*Figure 6.* OT plan visualization vary entropic regularization strength.

## C.5. Derivation of Proposition 3.4

*Proposition* (3.4). The semi-dual formulation in Equation (9) admits a unique optimal solution under standard regularity conditions on $f^\star$ and suitable normalization of $w$.

*Proof.* Let the entropy-regularized dual objective in Equation (9) be

$$g(w) := -\epsilon \mathbb{E}_{p(x_t)}[\log \mathbb{E}_{p_T(x)}(\exp(\frac{w(x) - \frac{1}{2\eta}\|x - x_t\|_2^2}{\epsilon}))] - \mathbb{E}_{p_T(x)}[f^\star(-w(x))], \epsilon > 0.$$

Assume the admissible space $\mathcal{A}$ consists of measurable functions $w$ satisfying:

(A1) Both expectations in $g(w)$ are finite;
(A2) $w$ is normalized such that $\mathbb{E}_{p_T}[w(x)] = 0$;
(A3) $f^\star$ is strictly convex on the range $\{-w(x) : w \in \mathcal{A}\}$.

Then $g(w)$ is strictly concave on $\mathcal{A}$ and admits at most one maximizer. The corresponding justifications are listed as follows:

1) **Log-sum-exp term:** Fix $x_t$ and we first define $\Phi_\epsilon(w; x_t)$ as follows:

$$\Phi_\epsilon(w; x_t) := -\epsilon \log \mathbb{E}_{p_T(x)}[\exp(\frac{w(x) - \frac{1}{2\eta}\|x - x_t\|_2^2}{\epsilon})].$$

For $w_1 \neq w_2$ with $w_1(x) \neq w_2(x)$ on a set of positive $p_T$-measure, strict convexity of $\exp(\cdot)$ and Hölder's inequality (Hardy et al., 1952) imply the following result:

$$\Phi_\epsilon((1 - \lambda)w_1 + \lambda w_2; x_t) > (1 - \lambda)\Phi_\epsilon(w_1; x_t) + \lambda\Phi_\epsilon(w_2; x_t) \tag{61}$$

for $\lambda \in (0, 1)$, with equality only if the difference between $w_1$ and $w_2$, $w_1 - w_2$, is constant $p_T$-almost everywhere Since the admissible set imposes normalization (excluding constant functions), taking expectation over $x_t$ yields a strictly concave first term.

2) $f$-**divergence related term:** The term $-\mathbb{E}_{p_T(x)}[f^\star(-w(x))]$ is strictly concave since $f^\star$ is strictly convex, for $w_1 \neq w_2$ and $\lambda \in (0,1)$, we have the following result:

$$-f^\star(-(1-\lambda)w_1(x) - \lambda w_2(x)) > -(1-\lambda)f^\star(-w_1(x)) - \lambda f^\star(-w_2(x))$$

holds on the set where $w_1(x) \neq w_2(x)$, which has positive $p_T$-measure.

As a sum of two strictly concave functions, $g$ is strictly concave on the admissible set and admits a unique maximizer. □

### C.6. Derivation and Discussions of Proposition 3.5

C.6.1. DERIVATION OF PROPOSITION 3.5

*Proposition* (3.5). The optimal solution $\rho^*(x)$ to problem defined in Equation (8) satisfies the following bound when $\mathcal{W}_2(p(x_t), p_T(x)) \leq 2\eta$:

$$\mathbb{D}_f[\rho^*(x), p_T(x)] \leq \mathcal{W}_2(p(x_t), p_T(x)). \tag{62}$$

*Proof.* At the beginning of the proof, let us first recall related concepts. Specifically, let $\Pi(p(x_t), \rho)$ denote the set of couplings $\pi(x_t, x)$ such that $\pi$ is a probability joint density with fixed marginals:

$$\int \pi(x_t, x)\mathrm{d}x = p(x_t), \tag{63a}$$

$$\int \pi(x_t, x)\mathrm{d}x_t = \rho(x). \tag{63b}$$

Based on this, we have:

$$\iint \pi(x_t, x)\mathrm{d}x_t\mathrm{d}x = 1.$$

In addition, $\kappa(x_t, x)$ is our reference joint probability density defined as follows:

$$\kappa(x_t, x) := p(x_t)p_T(x). \tag{64}$$

On this basis, Let us consider the entropy-regularized primal objective defined by Equation (8). For any feasible solution $(\rho, \pi)$, we can first introduce functional $J$ as follows:

$$J(\rho, \pi) := \frac{1}{2\eta}\mathcal{W}_2^2(\rho(x), p(x_t)) + \mathbb{D}_f[\rho(x), p_T(x)] + \epsilon\iint \pi(x_t, x)[\log\frac{\pi(x_t, x)}{\kappa(x_t, x)} - 1]\mathrm{d}x_t\mathrm{d}x. \tag{65}$$

Let $(\rho^*, \pi^*)$ be the optimal solution to the primal problem. Then $(\rho^*, \pi^*)$ satisfies the following inequality for all feasible $(\rho, \pi)$

$$J(\rho^*, \pi^*) \leq J(\rho, \pi). \tag{66}$$

Based on the definition of KL divergence,

$$\mathbb{D}_{\mathrm{KL}}[\pi(x_t, x), \kappa(x_t, x)] = \iint \pi(x_t, x)\log\frac{\pi(x_t, x)}{\kappa(x_t, x)}\mathrm{d}x_t\mathrm{d}x$$

and the fact that

$$\iint \pi(x_t, x)\mathrm{d}x_t\mathrm{d}x = 1,$$

we arrive at the following results:

$$\epsilon\iint \pi(x_t, x)[\log\frac{\pi(x_t, x)}{\kappa(x_t, x)} - 1]\mathrm{d}x_t\mathrm{d}x = \epsilon\mathbb{D}_{\mathrm{KL}}[\pi(x_t, x), \kappa(x_t, x)] - \epsilon.$$

Since

$$\mathbb{D}_{\mathrm{KL}}[\pi(x_t, x), \kappa(x_t, x)] \geq 0,$$

for the optimal solution $(\rho^*, \pi^*)$, we obtain the following inequality:

$$J(\rho^*, \pi^*) = \frac{1}{2\eta} \mathcal{W}_2^2(\rho^*(x), p(x_t)) + \mathbb{D}_f[\rho^*(x), p_T(x)] + \epsilon \mathbb{D}_{\text{KL}}[\pi^*(x_t, x), \kappa(x_t, x)] - \epsilon \geq \mathbb{D}_f[\rho^*(x), p_T(x)] - \epsilon.$$

Since both $\mathcal{W}_2^2(\cdot, \cdot) \geq 0$ and $\mathbb{D}_{\text{KL}}[\cdot, \cdot] \geq 0$, we have the following inequality:

$$\mathbb{D}_f[\rho^*, p_T] \leq J(\rho^*, \pi^*) + \epsilon. \tag{67}$$

As such, take the following conditions:

$$\rho(x) = p_T(x), \tag{68a}$$
$$\pi(x_t, x) = \kappa(x_t, x) = p(x_t)p_T(x), \tag{68b}$$

we can observe that $\pi \in \Pi(p(x_t), p_T)$ is feasible, and clearly we have the following results:

$$\mathbb{D}_f[p_T(x), p_T(x)] = 0, \tag{69a}$$
$$\mathbb{D}_{\text{KL}}[\kappa(x_t, x), \kappa(x_t, x)] = 0. \tag{69b}$$

Hence, we arrive at the following results:

$$J(p_T, \kappa) = \frac{1}{2\eta} \mathcal{W}_2^2(p_T(x), p(x_t)) + 0 + \epsilon(0 - 1) = \frac{1}{2\eta} \mathcal{W}_2^2(p(x_t), p_T) - \epsilon. \tag{70}$$

Based on Equations (66) and (70), we have:

$$J(\rho^*, \pi^*) \leq \frac{1}{2\eta} \mathcal{W}_2^2(p(x_t), p_T) - \epsilon. \tag{71}$$

Plugging Equation (71) into Equation (67) yields the following result:

$$\mathbb{D}_f[\rho^*(x), p_T(x)] \leq \frac{1}{2\eta} \mathcal{W}_2^2(p(x_t), p_T) - \cancel{\epsilon} + \cancel{\epsilon} = \frac{1}{2\eta} \mathcal{W}_2^2(p(x_t), p_T). \tag{72}$$

In addition, since $\mathcal{W}_2(p(x_t), p_T(x)) \leq 2\eta$, we have:

$$\mathcal{W}_2(p(x_t), p_T(x)) \leq 2\eta \iff \frac{1}{2\eta} \mathcal{W}_2(p(x_t), p_T(x)) \leq 1 \iff \frac{1}{2\eta} \mathcal{W}_2^2(p(x_t), p_T(x)) \leq \mathcal{W}_2(p(x_t), p_T(x)). \tag{73}$$

Substituting Equation (73) into Equation (72), we obtain the following result:

$$\mathbb{D}_f[\rho^*(x), p_T(x)] \leq \mathcal{W}_2(p(x_t), p_T(x)),$$

as desired. $\qquad\square$

## C.6.2. THEORETICAL ANALYSIS

After proving Proposition 3.5, we provide some justification for the statement in the main contents: "In view of Proposition 3.5, we note that as $t$ increases, the transported density $\rho(x)$ progressively approaches the target distribution $p_T(x)$." Specifically, according to the definition of the JKO proximal recursion Equations (6) and (8), each update $p(x_{t+1})$ satisfies (see Theorem 4.0.4 in reference (Ambrosio et al., 2005)):

$$\frac{1}{2\eta} \mathcal{W}_2^2(p(x_{t+1}), p(x_t)) + \mathbb{D}_f[p(x_{t+1}), p_T(x)] \leq \mathbb{D}_f[p(x_t), p_T(x)],$$

where $\mathbb{D}_f[p(x_t), p_T(x)] \geq 0$ achieves its minimum at $p_T(x)$. Besides, when $\mathbb{D}_f$ is geodesically convex, this inequality implies that the divergence decreases at each step, yielding:

$$\mathcal{W}_2(p(x_{t+1}), p_T(x)) \leq \mathcal{W}_2(p(x_t), p_T(x)) - \Delta_t, \quad \Delta_t \geq 0. \tag{74}$$

Therefore, by monotonic decrease of the divergence (see Equations (62) and (74)), we arrive at the following conclusion:

$$\mathbb{D}_f[\rho^*(x_{t+1}), p_T(x)] \leq \mathbb{D}_f[\rho^*(x_t), p_T(x)] - \Delta_t,$$

which demonstrates that the transported density $\rho(x)$ progressively converges toward the target $p_T(x)$.

### C.6.3. TOY CASE STUDY

To gain further insight into Proposition 3.5, we conduct a toy case study. The target distribution $p_T(x)$ is defined as

$$p_T(x) \propto \frac{1}{2}\mathcal{N}(-2.5, 0.8^2) + \frac{1}{2}\mathcal{N}(2.5, 0.8^2).$$

The initial particles are independently sampled according to

$$x_0^{(i)} \sim \mathcal{N}(0, 1), \quad i = 1, \dots, M,$$

where M denotes the number of particles. In this experiment, we set $M = 100$ and $\eta = 0.005$. Additionally, since the KL divergence between the particle approximation and the underlying density cannot be computed in closed form, we adopt the k-nearest-neighbor-based (kNN-based) estimator. Specifically, we estimate the differential entropy $\mathbb{H}(\rho^*)$ of the particle distribution using the Kozachenko–Leonenko kNN entropy estimator (Kozachenko & Leonenko, 1987), and approximate the cross-entropy term $\mathbb{E}_{\rho^*}[\log \rho^*(x)]$ by Monte Carlo averaging over the particles. Consequently, we compute the KL divergence by the following equation:

$$\mathbb{D}_{\mathrm{KL}}[\rho^*, p_T] = -\widehat{\mathbb{H}}(\rho^*) - \frac{1}{M}\sum_{i=1}^{M}\log p_T(x_t^{(i)}),$$

where $\widehat{\mathbb{H}}(\rho^*)$ denotes the estimated differential entropy. The visualization of the particle evolution and the corresponding quantitative comparisons are reported in Figure 7(a) and Figure 7(b), respectively.

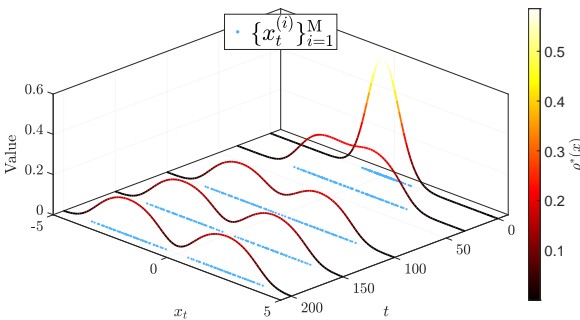

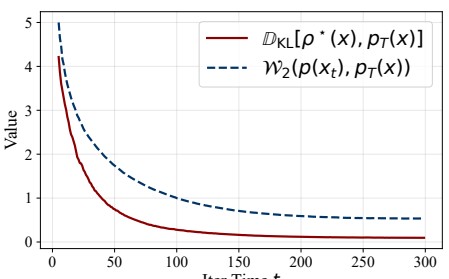

(a) Density Evolution Results, where the red line is estimated via kernel density estimation via median bandwidth.

(b) Evolution of $\mathbb{D}_{\mathrm{KL}}[\rho^*(x), p_T(x)]$ and $\mathcal{W}_2(p(x_t), p_T(x))$ along $t$.

*Figure 7.* Toy case study for solving problem $\inf_{\rho(x)} \mathbb{D}_{\mathrm{KL}}[\rho(x), p(x_T)] + \frac{1}{2\eta}\mathcal{W}_2^2(\rho(x), p(x_t))$.

From Figure 7(a), we observe that as the evolution time increases, the density $p(x_t)$ progressively approaches the target distribution $p_T(x)$. This suggests that solving the proximal recursion effectively yields a progressive approximation of $p_T(x)$ along time $t$, supporting our claim. Moreover, as shown in Figure 7(b), both the KL divergence $\mathbb{D}_{\mathrm{KL}}(\rho^*(x), p_T(x))$ and the Wasserstein distance $\mathcal{W}_2(p(x_t), p_T(x))$ decrease with iteration time $t$. Notably, the Wasserstein distance remains slightly larger than the KL divergence throughout, which further corroborates the result of Proposition 3.4.

### C.7. Derivation of Proposition 3.6

*Proposition* (3.6). Under mild assumptions, the E-SUOT-based GDA ensures that the target domain generalization error is upper-bounded by the following inequality:

$$\varepsilon_{p_T}(h_T) \leq \varepsilon_{p_0}(h_0) + \varepsilon_{p_0}(h_T^*) + \iota\zeta\mathcal{C} + \mathcal{S}_{\mathrm{stat}}, \tag{75}$$

where $\iota$ is the Lipschitz constant of the loss function, $\zeta$ is the Lipschitz constant bound for hypotheses in $\mathcal{H}$, $\mathcal{C}$ aggregates the cumulative domain transportation and label continuity costs along the gradual domain adaptation path, $\mathcal{S}_{\mathrm{stat}}$ is the statistical error term, and $h_T^* \in \mathcal{H}$ is a reference hypothesis used to characterize the source-domain approximation gap.

Before formally proving the proposition, we introduce the following assumptions, which are mild and commonly satisfied in practical domain adaptation scenarios:

**(A. 1)** The loss function $\mathcal{L}(\cdot, y)$ is $\iota$-Lipschitz with respect to its first argument; that is, for any $a, a'$ and fixed $y$, we have:

$$|\mathcal{L}(a, y) - \mathcal{L}(a', y)| \leq \iota |a - a'|. \tag{76}$$

**(A. 2)** Each hypothesis $h \in \mathcal{H}$ is $\zeta$-Lipschitz, i.e., for any $x, x'$, we have:

$$|h(x) - h(x')| \leq \zeta \|x - x'\|. \tag{77}$$

**(A. 3)** The labeling function $q_t$ along the adaptation path is such that $|q_t(x) - q_{t-1}(x)|$ is small for most $x$, to ensure local continuity.

**(A. 4)** The sequence of domains $(p_0, p_1, \ldots, p_T)$ is induced by E-SUOT-based GDA transport, so that the total cumulative cost $\mathcal{C}$ as defined below is finite.

**(A. 5)** At every step, empirical risk minimization over sufficient samples ensures a small empirical-to-expected error gap, leading to a statistical error term $\mathcal{S}_{\text{stat}}$.

**(A. 6)** The sample size for each domain is large enough to make $\mathcal{S}_{\text{stat}}$ negligible in the asymptotic regime.

Notably, Assumption (A.1) is standard and well-justified for classification tasks employing the categorical cross-entropy loss. More specifically, the gradient of $\mathcal{L}_{\text{CE}}$ with respect to the logits $a$ is bounded as

$$\left| \frac{\partial \mathcal{L}_{\text{CE}}}{\partial a_j} \right| = \left| [\text{softmax}(a)]_j - \mathbb{I}(j = y) \right| \leq 1,$$

for any class index $j$, since each softmax component lies within $[0, 1]$, and the indicator function $\mathbb{I}(j = y)$ takes values in $\{0, 1\}$. By the classical Lagrange mean value theorem (Theorem 4 in (Thomas et al., 2014)), this bounded-gradient property implies that $\mathcal{L}_{\text{CE}}$ is Lipschitz continuous with respect to its first argument on any bounded input domain, with a finite Lipschitz constant depending on the chosen norm. In practice, this assumption can be further facilitated by applying weight normalization or spectral-norm regularization, which help maintain bounded network outputs and thereby make the Lipschitz condition more readily satisfied during optimization.

Furthermore, Assumptions (A.2), (A.5) and (A.6) are standard and generally hold for commonly used loss functions and hypothesis classes. Unless the loss or model is exceptionally non-standard, these can be stated directly with the proposition and do not require additional justification.

Assumption (A.3) holds in cases where the labeling function changes smoothly along the adaptation path. For our construction, since the intermediate domains are generated by incremental, continuous transformations (e.g., gradual style or environmental shifts), the underlying semantics of inputs remain stable, and thus $\mathbb{E}_{p_{t-1}(x)}[|q_t(x) - q_{t-1}(x)|]$ is small for every $t$. This situation typically occurs under covariate shift, where only the input distribution evolves while class definitions stay fixed. However, this assumption may fail in fine-grained recognition tasks or tasks with abrupt semantic boundaries, where visually similar samples can belong to distinct categories. Hence, our analysis mainly applies when the domain evolution does not induce significant concept shift, or locally within regions where the labeling function remains approximately smooth.

As for Assumption (A.4), in our E-SUOT-based GDA, each domain is generated via an iterative unbalanced optimal transport step that progressively reduces the transport cost as we proved in Proposition 3.5. This guarantees that the cumulative cost $\mathcal{C}$ is finite, as can be bounded analytically. In summary, all the above assumptions are justified in our setting. Based on these assumptions, we now proceed with the formal proof.

*Proof.* Our goal is to bound the target risk $\varepsilon_{p_T}(h_T)$. Consider the telescoping sum along the domain adaptation path:

$$\varepsilon_{p_T}(h_T) = \varepsilon_{p_0}(h_0) + [\varepsilon_{p_T}(h_T) - \varepsilon_{p_0}(h_0)]. \tag{78}$$

To make the recursion explicit, rewrite this as:

$$\varepsilon_{p_T}(h_T) = \varepsilon_{p_0}(h_0) + \sum_{t=1}^{T} [\varepsilon_{p_t}(h_t) - \varepsilon_{p_{t-1}}(h_{t-1})]. \tag{79}$$

For each $t \in \{1, \ldots, T\}$, we decompose the one-step risk difference as follows:

$$\varepsilon_{p_t}(h_t) - \varepsilon_{p_{t-1}}(h_{t-1}) = \underbrace{[\varepsilon_{p_t}(h_t) - \varepsilon_{p_t}(h_{t-1})]}_{\text{hypothesis update term}} + \underbrace{[\varepsilon_{p_t}(h_{t-1}) - \varepsilon_{p_{t-1}}(h_{t-1})]}_{\text{domain shift term}}. \tag{80}$$

The first term reflects the effect of updating the hypothesis from $h_{t-1}$ to $h_t$. Since $h_t$ is obtained by empirical risk minimization on samples from $p_t$, this term can be controlled up to an optimization and statistical deviation term, denoted by $s_t$. We therefore absorb this contribution into $s_t$.

When the labeling function is fixed, or when its variation is accounted for separately, the composition $x \mapsto \mathcal{L}(h(x), q_{t-1}(x))$ is $\iota\zeta$-Lipschitz with respect to $x$. Hence, by the Kantorovich–Rubinstein duality, we have the following inequality:

$$|\varepsilon_{p_t}(h) - \varepsilon_{p_{t-1}}(h)| \leq \iota\zeta \cdot \mathcal{W}_1(p_{t-1}, p_t). \tag{81}$$

Suppose the true label function $q_t$ changes along the path, which gives an additional cost due to the label discrepancy:

$$\iota \, \mathbb{E}_{p_t(x)} |q_t(x) - q_{t-1}(x)|. \tag{82}$$

Therefore, each step can be bounded by

$$|\varepsilon_{p_t}(h_t) - \varepsilon_{p_{t-1}}(h_{t-1})| \leq \iota\zeta\mathcal{W}_1(p_{t-1}, p_t) + \iota \, \mathbb{E}_{p_t(x)} |q_t(x) - q_{t-1}(x)| + s_t \tag{83}$$

where $s_t$ denotes the statistical error at step $t$.

Let

$$\mathcal{C} := \sum_{t=1}^{T} [\mathcal{W}_1(p_{t-1}, p_t) + \frac{1}{\zeta} \mathbb{E}_{p_t(x)} |q_t(x) - q_{t-1}(x)|] \tag{84}$$

and

$$\mathcal{S}_{\text{stat}} := \sum_{t=1}^{T} s_t. \tag{85}$$

Sum these bounds for all $t \in \{1, \ldots, T\}$, we get:

$$\sum_{t=1}^{T} |\varepsilon_{p_t}(h_t) - \varepsilon_{p_{t-1}}(h_{t-1})| \leq \iota\zeta\mathcal{C} + \mathcal{S}_{\text{stat}}. \tag{86}$$

As the final classifier $h_T$ may not be optimally trained with respect to $p_0$, include the approximation gap:

$$\varepsilon_{p_0}(h_0) + \varepsilon_{p_0}(h_T^*) - \varepsilon_{p_0}(h_0) \tag{87}$$

where $h_T^*$ is the risk minimizer in $\mathcal{H}$ for $p_0$.

Finally, to account for the source-domain approximation gap induced by the hypothesis class, we include the reference risk $\varepsilon_{p_0}(h_T^*)$, where $h_T^* \in \mathcal{H}$ denotes a reference hypothesis evaluated on $p_0$. This yields the following inequality:

$$\varepsilon_{p_T}(h_T) \leq \varepsilon_{p_0}(h_0) + \varepsilon_{p_0}(h_T^*) + \iota\zeta\mathcal{C} + \mathcal{S}_{\text{stat}},$$

as desired. □

While Proposition 3.6 provides a clean decomposition, the last two terms are not directly computable from data. To bridge this gap between theory and practice, we attempt to estimate them using the following strategies:

- **Loss Lipschitz constant $\iota$:** According to the analysis of Assumption (A.1), we treat $\iota$ as a bounded constant in practice, which can be further controlled with weight normalization or spectral-norm regularization.

- **Hypothesis Lipschitz constant** $\zeta$**.** This bounds how sensitively hypotheses $h \in \mathcal{H}$ react to input perturbations. In our implementation, we use a multi-layer perceptron with ReLU activation functions. Since ReLU is 1-Lipschitz, the Lipschitz constant $\zeta$ of the classifier has the following upper bound:

$$\zeta \leq \prod_{\ell=1}^{L} \|W_\ell\|_2, \tag{88}$$

where $W_\ell$ denotes the weight matrix of the $\ell$-th linear layer and $\|W_\ell\|_2$ is its spectral norm. Similarly, $\zeta$ can be controlled by applying spectral normalization or weight normalization to the layers.

- **Cumulative cost** $\mathcal{C}$**:** Recall Equation (84). The Wasserstein-1 term $\mathcal{W}_1(p_{t-1}, p_t)$ can be approximated using sample-based optimal transport distances, such as the Sinkhorn distance (Cuturi, 2013), computed on intermediate feature representations. The label-continuity term $\mathbb{E}_{p_t(x)}[|q_t(x) - q_{t-1}(x)|]$ can be approximated using pseudo-labels predicted by the models at two consecutive steps, which measures how much the pseudo-labels change along the adaptation path.

- **Statistical error** $\mathcal{S}_{\text{stat}}$**:** The term $\mathcal{S}_{\text{stat}}$ collects the statistical deviations between empirical and population risks along the adaptation path. Based on the Rademacher complexity bound in Section 9.2 of (Bach, 2024), for a $\iota$-Lipschitz loss and a hypothesis class whose complexity is controlled by $D$, the step-wise statistical error can be bounded as follows:

$$s_t \leq \frac{2C_{\text{rad}}\iota D}{\sqrt{N}}, \tag{89}$$

where $N$ is the training sample size, $D$ controls the complexity of the function class, and $C_{\text{rad}}$ is a universal constant arising from the Rademacher complexity bound. Therefore, for all $t \in \{1, \ldots, T\}$, we can estimate $\mathcal{S}_{\text{stat}}$ as follows:

$$\mathcal{S}_{\text{stat}} = \sum_{t=1}^{T} s_t \leq \frac{2C_{\text{rad}}\iota D}{\sqrt{N}} T. \tag{90}$$

### C.8. Discussions on the Selection of Step Size $\eta$

Let us recall Equations (6) and (18) as follows:

$$\begin{cases} \mathcal{L}^{\text{Primal}} = \arg\min_{\rho(x) \in \mathcal{P}_2(\mathbb{R}^D)} \frac{1}{2\eta} \mathcal{W}_2^2(\rho(x), p(x_t)) + \mathbb{D}_f[\rho(x), p_T(x)], \\ \min_{\pi \geq 0} \iint c(x, y)\, \pi(x, y)\, \mathrm{d}y\, \mathrm{d}x + \lambda_1 \mathbb{D}_f(\tilde{\rho}(x), \rho(x)) + \lambda_2 \mathbb{D}_f(\tilde{\xi}(y), \xi(y)). \end{cases} \tag{91}$$

Since Equation (6) is a variant of Equation (18), where $\lambda_1 \equiv 0$, one natural question is how to select $\lambda_2$, i.e. How to select $\eta$? Since our target is decreasing the functional $\mathbb{D}_f[\rho(x), p_T(x)]$ along the simulation process, one key factor is whether the choice of $\eta$ can decrease the functional $\mathbb{D}_f[\rho(x), p_T(x)]$. Take the KL divergence, the $f$-divergence we mainly consider in the proposed E-SUOT approach, we have the following proposition for selecting the $\eta$:

**Proposition C.1.** *Suppose that the* $\|\frac{\delta\mathbb{D}_{\text{KL}}[\rho(x), p_T(x)]}{\delta\rho(x)}\| \leq \mathscr{A}$ *and* $\|\nabla\frac{\delta\mathbb{D}_{\text{KL}}[\rho(x), p_T(x)]}{\delta\rho(x)}\| \leq \mathscr{B}$, *there exists a constant* $\mathscr{H}_0$ *that controls the tailness of* $p_T(x)$, *and let* $\{\rho_t(x)|t = 1, \ldots, T\}$ *denote the sequence of empirical PDF of the intermediate domain generated by the JKO recursion. When* $\eta$ *satisfies the following condition, the sequence of KL divergence* $\{\mathbb{D}_{\text{KL}}[\rho_t(x), p_T(x)]|t = 1, \ldots, T\}$ *converges to a finite value as* $t \to \infty$:

$$0 < \eta < \min\left(\frac{1}{\mathscr{B}}, \frac{\mathscr{H}_0}{\mathscr{A}}\right). \tag{92}$$

Before presenting the proof, we should introduce the light-tailness property on the target distribution $p_T(x)$ in order to ensure the validity of our Taylor expansion and to control higher-order discretization errors during the proof process. Specifically, we say that $p_T(x)$ is light-tailed (Ambrosio et al., 2005; Johnson & Zhang, 2018; 2021) if there exists a universal constant $\mathscr{H}_0 < \infty$ such that

$$\int \|\nabla \log p_T(x)\|\, p_T(x)\mathrm{d}x < \mathscr{H}_0. \tag{93}$$

We call $\mathscr{H}_0$ the "light-tail constant" of $p_T(x)$. This condition requires that the expectation (under $p_T(x)$) of the norm of the score function $\nabla p_T(x)$ is finite. Intuitively, this ensures that $p_T(x)$ decays sufficiently rapidly in the tails so that the gradients do not blow up at infinity. On this basis, the proof is articulated as follows motivated by Johnson & Zhang (2021):

*Proof.* Suppose at time $t$ and time $t + \eta$, we have:

$$x_{t+\eta} = \boldsymbol{T}(x) := x_t + \eta v_t(x_t). \tag{94}$$

We denote the probability distributions, before and after applying Equation (94) as $\rho_t(x)$ and $\rho_{t+\eta}(x)$, respectively. The aim is to Taylor expand the evolution of

$$\mathbb{D}_{\mathrm{KL}}[\rho_{t+\eta}(x), p_T(x)] = \int \rho_{t+\eta}(x) \log \frac{\rho_{t+\eta}(x)}{p_T(x)} \mathrm{d}x, \tag{95}$$

with respect to $\eta$ around $\eta = 0$. The new probability distribution, for small $\eta$, can be given as follows according to the Liouville's theorem:

$$\rho_{t+\eta}(x) = \rho_t(\boldsymbol{T}^{-1}(x)) \cdot |\det \boldsymbol{J}_{\boldsymbol{T}^{-1}}(z)| \tag{96}$$

where $\boldsymbol{J}_{\boldsymbol{T}^{-1}}(z)$ is the Jacobian matrix of the inverse map, and when $\eta$ is small enough, the inverse function $\boldsymbol{T}^{-1}(x)$ of function $\boldsymbol{T}(x)$ can be given as follows:

$$\boldsymbol{T}^{-1}(x) \approx x - \eta v_t(x). \tag{97}$$

Hence, expanding to the first order in $\eta$ can be given as follows:

$$\rho_{t+\eta}(x) \approx \rho_t(x - \eta v_t(x)) [1 - \eta \nabla \cdot \phi(z)] \approx \rho_t(x) - \eta \nabla [\rho_t(x) v_t(x)]. \tag{98}$$

Define $F(\eta)$:

$$F(\eta) := \mathbb{D}_{\mathrm{KL}}[\rho_{t+\eta}(x), p_T(x)] = \int \rho_{t+\eta}(x) \log \frac{\rho_{t+\eta}(x)}{p_T(x)} \mathrm{d}x. \tag{99}$$

Applying Taylor's expansion at $\eta = 0$, we can obtain the following result:

$$F(\eta) = F(0) + \eta F'(0) + \mathcal{O}(\eta^2). \tag{100}$$

Now, we start deriving $F'(0)$. When we take the derivative inside the integral, we have:

$$F'(\eta) = \frac{\mathrm{d}}{\mathrm{d}\eta} \int \rho_{t+\eta}(x) \log \frac{\rho_{t+\eta}(x)}{p_T(x)} \mathrm{d}x = \int \frac{\mathrm{d}}{\mathrm{d}\eta} \rho_{t+\eta}(x) [1 + \log \frac{\rho_{t+\eta}(z)}{p_T(x)}] \mathrm{d}x \tag{101}$$

At $\eta = 0$, $\rho_{t+\eta}(x) = \rho_t(x)$:

$$F'(0) = \int \frac{\mathrm{d}}{\mathrm{d}\eta} \rho_{t+\eta}(x) \Big|_{\eta=0} [1 + \log \frac{\rho_t(x)}{p_T(x)}] \mathrm{d}x. \tag{102}$$

Now, using the result from the calculus of variations:

$$\frac{\mathrm{d}}{\mathrm{d}\eta} \rho_{t+\eta}(x) \Big|_{\eta=0} = -\nabla \cdot (\rho_t(x) v_t(x)) \tag{103}$$

Thus,

$$F'(0) = -\int \nabla \cdot (\rho_t(x) v_t(x)) [1 + \log \frac{\rho_t(x)}{p_T(x)}] \mathrm{d}x. \tag{104}$$

Now, use integration by parts:

$$\int -\nabla \cdot [\rho_t(x) v_t(x)] g(x) \mathrm{d}z = \int [\rho_t(x) v_t(x)]^\top \nabla g(x) \mathrm{d}x. \tag{105}$$

Set $g(x) = 1 + \log \frac{\rho_t(x)}{p_T(x)}$. Its gradient is:

$$\nabla g(x) = \nabla \log \rho_t(x) - \nabla \log p_T(x). \tag{106}$$

Hence,

$$F'(0) = \int [\rho_t(x) v_t(x)]^\top [\nabla \log \rho_t(x) - \nabla \log p_T(x)] \mathrm{d}x = \mathbb{E}_{\rho_t(x)} \left[ v_t^\top(x)(\nabla \log \rho_t(x) - \nabla \log p_T(x)) \right]. \tag{107}$$

But with a negative sign because the original derivative is minus divergence:

$$F'(0) = -\mathbb{E}_{\rho_t(x)}\left[v_t^\top(x)(\nabla \log \rho_t(x) - \nabla \log p_T(x))\right]. \tag{108}$$

Putting all together, we get:

$$\mathbb{D}_{\mathrm{KL}}[\rho_{t+\eta}(x), p_T(x)] = \mathbb{D}_{\mathrm{KL}}[\rho_t(x), p_T(x)] - \eta\mathbb{E}_{\rho_t(x)}\left[v_t^\top(x)(\nabla \log p_T(x) - \nabla \log \rho_t(x))\right] + \mathcal{O}(\eta^2). \tag{109}$$

Notably, the optimal velocity field $v_t^*(x)$ for KL divergence is $-\nabla \frac{\delta \mathbb{D}_{\mathrm{KL}}[\rho_t(x)]}{\delta \rho_t(x)}$. Thus, we have:

$$\mathbb{D}_{\mathrm{KL}}[\rho_{t+\eta}(x), p_T(x)] = \mathbb{D}_{\mathrm{KL}}[\rho_t(x), p_T(x)] \underbrace{-\eta\mathbb{E}_{\rho_t(x)}\left[(\nabla \log p_T(x) - \nabla \log \rho_t(x))^\top v_t^*(x)\right]}_{\leq 0} + \mathcal{O}(\eta^2). \tag{110}$$

Since $\|\nabla \frac{\delta \mathbb{D}_{\mathrm{KL}}[\rho(x), p_T(x)]}{\delta \rho(x)}\| \leq \mathscr{B}$, there exists a positive constant $C$ such that:

$$\begin{aligned}
&\mathbb{D}_{\mathrm{KL}}[\rho_t(x), p_T(x)] - \eta\mathbb{E}_{\rho_t(x)}\left[(\nabla \log p_T(x) - \nabla \log \rho_t(x))^\top v_t^*(x)\right] + \mathcal{O}(\eta^2) \\
\leq &\mathbb{D}_{\mathrm{KL}}[\rho_t(x), p_T(x)] - \eta\mathbb{E}_{\rho_t(x)}\left[(\nabla \log p_T(x) - \nabla \log \rho_t(x))^\top v_t^*(x)\right] + C\eta^2,
\end{aligned} \tag{111}$$

where constant $C$ satisfies the following condition:

$$C \propto \mathscr{B}^2. \tag{112}$$

To avoid $C\eta^2$ dominating the right-hand-side of Equation (111), we should satisfy the following condition:

$$\frac{1}{\eta^2} \gg \mathscr{B}^2 \Rightarrow \eta \ll \frac{1}{\mathscr{B}} \Rightarrow \eta < \frac{1}{\mathscr{B}}. \tag{113}$$

According to the log–Sobolev inequality (Ambrosio et al., 2005; Durmus et al., 2019; Guo et al., 2025), for the target distribution $p_T(x)$ satisfying the curvature condition controlled by $\mathscr{H}_0 > 0$ (i.e., strong log–concavity or equivalent tailness control), there exists a log–Sobolev constant $\mathscr{H}_0$ such that for any smooth density $\rho_t(x)$,

$$\mathbb{D}_{\mathrm{KL}}[\rho_t(x), p_T(x)] \leq \frac{\mathscr{H}_0}{2}\,\mathbb{E}_{\rho_t(x)}\left[\|\nabla \log \frac{\rho_t(x)}{p_T(x)}\|^2\right]. \tag{114}$$

Equivalently, the following lower-bound form holds:

$$\mathbb{E}_{\rho_t(x)}\left[\|v_t^*(x)\|^2\right] \geq \frac{1}{\mathscr{H}_0}\,\mathbb{D}_{\mathrm{KL}}[\rho_t(x), p_T(x)], \tag{115}$$

where we used $v_t^*(x) = \nabla \log p_T(x) - \nabla \log \rho_t(x)$.

When the variational derivative $\frac{\delta \mathbb{D}_{\mathrm{KL}}[\rho_t(x), p_T(x)]}{\delta \rho_t(x)}$ is bounded by $\mathscr{A}$, and the spatial gradient of this functional derivative is bounded by $\mathscr{B}$, the log–Sobolev inequality admits the following perturbation-corrected version:

$$\mathbb{E}_{\rho_t(x)}[\|v_t^*(x)\|^2] \geq \frac{1}{\mathscr{H}_0}\{\mathbb{D}_{\mathrm{KL}}[\rho_t(x), p_T(x)] - \frac{\mathscr{A}}{\mathscr{H}_0}\}, \tag{116}$$

where the correction term $\frac{\mathscr{A}}{\mathscr{H}_0}$ compensates for the bounded $\|\nabla \frac{\delta \mathbb{D}_{\mathrm{KL}}}{\delta \rho}\|$ and ensures dimensional consistency of the energy inequality. Plugging Equation (116) into Equation (111) gives:

$$\begin{aligned}
&\mathbb{D}_{\mathrm{KL}}[\rho_{t+\eta}(x), p_T(x)] \\
\leq &\mathbb{D}_{\mathrm{KL}}[\rho_t(x), p_T(x)] - \eta\,\mathbb{E}_{\rho_t}[\|v_t^*(x)\|^2] + C\eta^2 \\
\leq &\mathbb{D}_{\mathrm{KL}}[\rho_t(x), p_T(x)] - \frac{\eta}{\mathscr{H}_0}\{\mathbb{D}_{\mathrm{KL}}[\rho_t(x), p_T(x)] - \frac{\mathscr{A}}{\mathscr{H}_0}\} + C\eta^2,
\end{aligned} \tag{117}$$

Rearranging Equation (117), we have:

$$\mathbb{D}_{\mathrm{KL}}[\rho_{t+\eta}(x), p_T(x)] \leq (1 - \tfrac{\eta}{\mathscr{H}_0})\,\mathbb{D}_{\mathrm{KL}}[\rho_t(x), p_T(x)] + \tfrac{\eta\mathscr{A}}{\mathscr{H}_0^2} + C\eta^2. \tag{118}$$

To ensure that the iteration gradually reduces $\mathbb{D}_{\mathrm{KL}}[\rho_t(x), p_T(x)]$, we should require:

$$(1 - \tfrac{\eta}{\mathscr{H}_0}) \in (0, 1) \quad \Rightarrow \quad 0 < \eta < \mathscr{H}_0. \tag{119}$$

According to Equation (113), ignoring $C\eta^2$, we obtain the equilibrium point, corresponding to the steady state as $t \to \infty$:

$$\mathbb{D}_{\mathrm{KL}}[\rho_\infty(x), p_T(x)] = \frac{\mathscr{A}}{\mathscr{H}_0}. \tag{120}$$

Next, to ensure that each discrete update indeed decreases the KL divergence, we impose

$$\mathbb{D}_{\mathrm{KL}}[\rho_{t+\eta}(x), p_T(x)] < \mathbb{D}_{\mathrm{KL}}[\rho_t(x), p_T(x)]. \tag{121}$$

Substituting Equation (118) into Equation (121) yields the following result:

$$-\frac{\eta}{\mathscr{H}_0}\mathbb{D}_{\mathrm{KL}}[\rho_t(x), p_T(x)] + \frac{\eta\mathscr{A}}{\mathscr{H}_0^2} + C\eta^2 < 0. \tag{122}$$

Dividing both sides by $\eta > 0$ and rearranging terms gives the following equation:

$$\mathbb{D}_{\mathrm{KL}}[\rho_t(x), p_T(x)] > \frac{\mathscr{A}}{\mathscr{H}_0} + C\eta\,\mathscr{H}_0. \tag{123}$$

In the late stage of GDA task, $\mathbb{D}_{\mathrm{KL}}[\rho_t(x), p_T(x)]$ approaches its equilibrium value $\mathbb{D}_{\mathrm{KL}}[\rho_\infty(x), p_T(x)] = \frac{\mathscr{A}}{\mathscr{H}_0}$, so that:

$$\mathbb{D}_{\mathrm{KL}}[\rho_t, p_T] - \mathbb{D}_{\mathrm{KL}}[\rho_\infty, p_T] \approx \mathbb{D}_{\mathrm{KL}}[\rho_\infty, p_T] = \frac{\mathscr{A}}{\mathscr{H}_0}. \tag{124}$$

Hence, the typical contraction strength per update is of order:

$$\frac{\eta}{\mathscr{H}_0}\big(\mathbb{D}_{\mathrm{KL}}[\rho_t, p_T] - \mathbb{D}_{\mathrm{KL}}[\rho_\infty, p_T]\big) \approx \frac{\eta}{\mathscr{H}_0}\frac{\mathscr{A}}{\mathscr{H}_0}. \tag{125}$$

To guarantee that the quadratic residual $C\eta^2$ does not dominate the contraction term in Equation (125), we require:

$$C\eta^2 \ll \frac{\eta}{\mathscr{H}_0}\frac{\mathscr{A}}{\mathscr{H}_0} \quad \Longrightarrow \quad \eta \ll \frac{\mathscr{H}_0}{\mathscr{A}}. \tag{126}$$

That is, the discretization step must satisfy the stability condition since $\mathscr{A} > 0$:

$$0 < \eta < \frac{\mathscr{H}_0}{\mathscr{A}} < \mathscr{H}_0, \tag{127}$$

which ensures that the numerical update is dominated by the contraction term rather than by the additive bias or high-order error. Based on Equations (113) and (127), we arrive at the desired result. $\qquad\square$

## D. Detailed Algorithm of the E-SUOT Framework

While Algorithm 1 outlines the general workflow for generating the intermediate domain, it does not specify how E-SUOT can be applied to the GDA task. To bridge this gap, we first present the complete workflow for E-SUOT-based GDA in Algorithm 3.

Building on this foundation, the complete workflow for E-SUOT-based gradual domain adaptation is summarized in Algorithm 3 based on Algorithm 1. Notably, our algorithm decouples the training of the transport function $T_\theta$ from the fine-tuning of the classifier $h_\omega$. This separation allows the intermediate domain to be generated offline and subsequently used for online inference, potentially reducing overall computation time comparable to traditional GDA approaches.

---

**Algorithm 3** Overall Workflow for Constructing E-SUOT-based Gradual Domain Adaptation

---

**Input:** Source domain samples: $\{(x_0^{(i)}, y_0^{(i)})\}_{i=1}^{\mathrm{N}}$, target domain samples: $\{(x_T^{(i)}, y_T^{(i)})\}_{i=1}^{\mathrm{N}}$, entropy regularization strength: $\epsilon$, step size: $\eta$, number of intermediate domains $T-1$, neural network batch size $\mathcal{B}$, and neural network training epochs: $\mathcal{E}$.
**Output:** Classifier in target domain $h_{\omega,T}$.

1: Initialize the classifier $h_{\omega,0}$: $h_{\omega,0} \leftarrow \arg\min_\omega \mathcal{L}_{\mathrm{CE}}(x_0, h_{\omega,t}, y_0)$.
2: Train $\mathcal{T} = \{\boldsymbol{T}_{\theta,t}\}_{t=1}^{T-1}$: $\mathcal{T} \leftarrow$Algorithm 1.
3: **for** $t = 0$ to $T - 1$ **do**
4:   Obtain the intermediate domain data $\{(x_{t+1}^{(i)}, y_{t+1}^{(i)})\}$: $x_{t+1}^{(i)} \leftarrow \boldsymbol{T}_{\theta,t}(x_t^{(i)})$ and $y_{t+1}^{(i)} \leftarrow y_t^{(i)}$ for all $i \in \{1, \ldots, \mathrm{N}\}$.
5:   Finetune the classifier $h_{\omega,t+1}$: $h_{\omega,t+1} \leftarrow \arg\min_\omega \mathcal{L}_{\mathrm{CE}}(x_{t+1}, h_{\omega,t}, y_{t+1})$.
6: **end for**

---

# E. Detailed Information for Experiments

## E.1. Dataset Descriptions

- **Portraits:** Portraits is a binary gender classification dataset comprising 37,921 front-facing portrait images collected between 1905 and 2013. Following the chronological split protocol of (Kumar et al., 2020), we divide the data into a source domain (the earliest 2,000 images), intermediate domains (14,000 images not utilized in this work), and a target domain (the subsequent 2,000 images), similar to the setting in reference (Zhuang et al., 2024).
- **Rotated MNIST:** Rotated MNIST is variant of the standard MNIST dataset (Deng, 2012) in which images are rotated to create domain adaptation challenges. As described in (He et al., 2024; Kumar et al., 2020), we use 4,000 source images and 4,000 target images, with the target images rotated by $45°$ to $60°$.
- **Office-Home:** Office-Home is a domain adaptation benchmark dataset consisting of approximately 15,500 images categorized into 65 object classes commonly found in office and home environments (Venkateswara et al., 2017). The dataset encompasses four visually distinct domains—*Artistic (Ar)*, *Clipart (Cl)*, *Product (Pr)*, and *Real-World (Rw)*. Following common domain adaptation protocols, one domain is selected as the source domain while another serves as the target domain, resulting in a total of 12 domain transfer tasks (e.g., Ar→Rw, Cl→Pr, etc.)

## E.2. Experimental Settings

### E.2.1. PRELIMINARIES OF SELF-TRAINING METHOD

Self-training is a classical semi-supervised learning strategy that leverages the model's own predictions on unlabeled data to iteratively improve its performance. Given a model $h_\omega$ trained on labeled source data, we use it to generate pseudo-labels for unlabeled samples in a target or auxiliary dataset $\mathcal{D}_{\mathrm{aux}}$. Each unlabeled input $x_i \in \mathcal{D}_{\mathrm{aux}}$ is assigned a pseudo-label $\tilde{y}_i = \mathrm{sign}(h_\omega(x_i))$, indicating a positive or negative prediction. A new model $h_{\omega'}$ is then trained to minimize the empirical loss on this pseudo-labeled dataset:

$$\mathrm{ST}(\theta, \mathcal{D}_{\mathrm{aux}}) = \arg\min_{h'_\omega \in \mathcal{H}} \frac{1}{\mathrm{N}_{\mathrm{aux}}} \sum_{x_i \in \mathcal{D}_{\mathrm{aux}}} \mathcal{L}(h'_\omega(x_i), \mathrm{sign}(h_\omega(x_i))), \tag{128}$$

where ST is the abbreviation of self-training, $\mathrm{N}_{\mathrm{aux}}$ denotes the sample size of auxiliary dataset $\mathcal{D}_{\mathrm{aux}}$. This procedure can be iteratively repeated, replacing $\theta$ with the newly optimized $\theta'$ to refine the pseudo-labels over time.

Intuitively, self-training alternates between the following two steps:

1) Producing pseudo-labels using the current classifier.
2) Retraining the model on these pseudo-labels, thereby progressively refining the decision boundary.

### E.2.2. PRELIMINARIES OF SMMD AND MMDSW

In our manuscript, we employ the SMMD (Hertrich et al., 2024) and MMDSW (Bonet et al., 2025) approaches. SMMD and MMDSW stand for Sliced Maximum Mean Discrepancy and Maximum Mean Discrepancy Sliced Wasserstein, respectively.

For SMMD, we use a sliced MMD formulation with the one-dimensional Riesz kernel $\mathcal{K}(s,t) = -|s-t|$:

$$\mathrm{SMMD}(\mu, \nu) = c_{\mathrm{D}} \mathbb{E}_{\vartheta \sim \mathbb{S}^{\mathrm{D}-1}}[\mathbb{E}_{X \sim \mu, Y \sim \nu}|\vartheta^\top X - \vartheta^\top Y| - \frac{1}{2}\mathbb{E}_{X,X' \sim \mu}|\vartheta^\top X - \vartheta^\top X'| - \frac{1}{2}\mathbb{E}_{Y,Y' \sim \nu}|\vartheta^\top Y - \vartheta^\top Y'|], \tag{129}$$

where $\vartheta$ is uniformly sampled from the unit sphere $\mathbb{S}^{D-1}$, $c_D$ is defined as follows:

$$c_D = \sqrt{\pi} \frac{\Gamma\left(\frac{D+1}{2}\right)}{\Gamma\left(\frac{D}{2}\right)}, \tag{130}$$

and $\Gamma(\cdot)$ is the gamma function. Based on SMMD, we choose the Riesz kernel as the kernel function and utilize the MMDSW defined as follows to avoid choosing the bandwidth compared with the vanilla version:

$$\text{MMDSW}(\mu, \nu) = -(\mathcal{SW}_p(\mu, \nu))^{r/2}, \tag{131}$$

where $\mathcal{SW}$ is defined as follows:

$$\mathcal{SW}_p(\mu, \nu) = (\mathbb{E}_{\vartheta \sim \mathbb{S}^{d-1}}[\mathcal{W}_p^p(\vartheta_{\#}\mu, \vartheta_{\#}\nu)])^{1/p}. \tag{132}$$

Here, $\vartheta_{\#}\mu$ and $\vartheta_{\#}\nu$ denote the one-dimensional projected distributions of $\mu$ and $\nu$ along direction $\vartheta$. In our implementation, we use $p = 2$ and $r = 1$.

### E.2.3. TRAINING AND EVALUATION PROTOCOLS

For GDA and UDA task, we follow the standard domain adaptation protocol, where model training and hyperparameter tuning are performed using only labeled source data, since validation on the target domain is infeasible in the unsupervised adaptation setting. All results are reported on the target domain dataset without using target labels for validation or early stopping.

For classifier training under this protocol, let $\widehat{h}_{\omega,t}(x_t)$ denote the logits produced by the classifier parameterized by $\omega$ at time step $t$. We define

$$h_{\omega,t}(x_t) = \text{softmax}(\widehat{h}_{\omega,t}(x_t)) \tag{133}$$

as the corresponding class-probability vector. Given the ground-truth label $y_t$, the categorical cross-entropy loss is formulated as follows:

$$\mathcal{L}_{\text{CE}}(x_t, \omega, y_t) = -\sum_{i=1}^{\mathcal{B}} y_t^{(i)} \log h_{\omega,t}^{(i)}(x_t) = -\sum_{i=1}^{\mathcal{B}} y_t^{(i)} \log[\text{softmax}(\widehat{h}_{\omega,t}(x_t))^{(i)}]. \tag{134}$$

### E.2.4. GDA TASK SETTINGS

The official implementations of GOAT (He et al., 2024) and CNF (Sagawa & Hino, 2025) are used in our experiments. Additionally, we employ UMAP (McInnes et al., 2018) to reduce the dimensionality of the three GDA datasets to 8 in order to align with the experimental tuple provided by Zhuang et al. (2024). The experiments are conducted on a workstation equipped with two NVIDIA RTX 4090 GPUs and repeated under at least three random seeds. The overall hyper-parameters we use in our GDA task are summarized in Table 7.

*Table 7.* Hyperparameters for E-SUOT on GDA task.

| Datasets | $\eta$ | $\mathcal{B}$ | $\epsilon$ | $T$ |
|---|---|---|---|---|
| Portraits | 0.5 | 1024 | 0.1 | 5 |
| MNIST 45° | 0.5 | 1024 | 0.01 | 5 |
| MNIST 60° | 0.5 | 2048 | 0.005 | 5 |

In all experiments, we parameterize the classifier $h_\phi$ as a three-layer multi-layer perceptron (MLP) at each step, utilizing ReLU activation functions and a hidden dimension of 100 for each layer. For both $T_\theta$ and $w_\phi$, we employ a two-layer MLP with the SiLU activation function and incorporate a skip connection to enable a residual structure (He et al., 2016). All models are optimized by the Adam optimizer (Kingma & Ba, 2015) with a learning rate of 0.0001. For all three GDA datasets, we apply UMAP (McInnes et al., 2018) to reduce their dimensionality to eight. For the classifier $h_\omega$, we use a two-layer MLP with ReLU activations and 128 hidden units. All baseline models are trained on features embedded by the UMAP.

### E.2.5. UDA TASK SETTINGS

For the UDA task, we adopt the Office-Home dataset (Venkateswara et al., 2017) as the benchmark to evaluate the performance of the proposed E-SUOT framework. Following the standard unsupervised domain adaptation (UDA) protocol, model training and hyperparameter tuning are performed solely using the labeled source data, without access to target labels for validation or early stopping. All results are reported on the target domain dataset.

We compare E-SUOT with a diverse set of representative UDA approaches, including DANN (Ganin & Lempitsky, 2015), MSTN (Xie et al., 2018), GVB-GD (Cui et al., 2020), RSDA (Gu et al., 2020; 2024), LAMBDA (Le et al., 2021), SENTRY (Prabhu et al., 2021), FixBi (Na et al., 2021), CST (Liu et al., 2021a), CoVi (Na et al., 2022), MMDSW (Bonet et al., 2025), SMMD (Hertrich et al., 2024), STDW (Wang et al., 2025c), AST (Shi & Liu, 2023), and GGF (Zhuang et al., 2024). For baselines including DANN, MSTN, GVB-GD, RSDA, LAMBDA, SENTRY, FixBi, CST, and CoVi, we directly report the publicly available results from their original papers under identical experimental settings (i.e., the same dataset and evaluation protocol).

For SMMD and MMDSW, we adopt the discrepancy terms defined in Section E.2.2. For AST, the adversarial perturbation budget is set to 0.1, with the number of AST steps set to 5. For STDW, the number of inter-domain migration steps is set to 4. For GGF, we follow the experimental protocol described in the original paper and conduct the experiments locally to maintain consistency with the original setup. For all the aforementioned methods, including SMMD, MMDSW, AST, STDW, and GGF, we adopt CoVi as the backbone feature extractor. The extracted features are then projected into an eight-dimensional space using UMAP. In addition, our E-SUOT framework also builds upon CoVi as the backbone feature extractor. The extracted features are embedded into an eight-dimensional space using UMAP. The classifiers $h_\omega$ for GGF and E-SUOT are implemented as two-layer ReLU MLPs with 256 hidden units. We set the $\eta$, $\mathcal{B}$, $\epsilon$, and $T$ as 0.5, 1024, 0.001, and 4, respectively. Both GGF and E-SUOT models are trained under the same conditions for fair comparison.

### E.3. Detailed Information for Ablation Studies

For the ablation study, we ablate two modules namely, the training strategy of $\boldsymbol{T}_\theta$ and the objective functional. The detailed information is provided in this part.

For "Training Strategy", the detailed experimental protocols are as follows:

- **Adversarial Training:** In our adversarial training scheme, we optimize Equation (7). Building on (Choi et al., 2023; 2024; Korotin et al., 2021; 2023), the training of $\boldsymbol{T}_\theta$ is formulated adversarially, as summarized in Algorithm 4. In *Line 5*, the penalty term $\frac{1}{2\eta}\|x_{t-1} - \boldsymbol{T}_{\theta,t-1}(x_{t-1})\|_2^2$ is omitted since it is constant with respect to $w_{\phi,t-1}$.
- **Barycentric-based Training:** We propose the algorithm for barycentric-based training in Algorithm 5. For barycentric-based training, rather than first compute the transport map, we attempt to compute the optimal transport map $\pi^*$ between $\rho(x_{t-1})$ and $p_T(x)$ as we demonstrate in *Line 4*. Based on this, we make barycentric projection (Courty et al., 2017b; Perrot et al., 2016) using this $\pi^*$ to obtain the proxy points (Liu et al., 2021b; 2023) for transport map learning as we demonstrate in *Line 5*. Finally, the transport map $\boldsymbol{T}_{\theta,t-1}$ is constructed based on these points, similar to the flow matching (Lipman et al., 2023), as we demonstrate in *Line 6*.

---

**Algorithm 4** Adversarial Training for $\{\boldsymbol{T}_{\theta,t}\}_{t=0}^{T-1}$.

---

**Input:** Intermediate domain samples: $\{(x_{t-1}^{(i)}, y_{t-1}^{(i)})\}_{i=1}^N$ for all $t \in \{1, \ldots, T\}$, target domain samples: $\{(x_T^{(i)}, y_T^{(i)})\}_{i=1}^N$, entropy regularization strength: $\epsilon$, step size: $\eta$, neural network batch size $\mathcal{B}$, and neural network training epochs: $\mathcal{E}$.
**Output:** The transportation map at $t-1$: $\boldsymbol{T}_{\theta,t-1}$.

1: Initialize
2: **for** $e = 1$ to $\mathcal{E}$ **do**
3:     Sample a batch $\{x_{t-1}^{(i)}\}_{i=1}^{\mathcal{B}} \sim \{(x_{t-1}^{(i)}, y_{t-1}^{(i)})\}_{i=1}^N$ and $\{x_T^{(i)}\}_{i=1}^{\mathcal{B}} \sim \{(x_T^{(i)}, y_T^{(i)})\}_{i=1}^N$.
4:     Update $w_{\phi,t-1}$ by: $\phi \leftarrow \arg\min_\phi \frac{1}{\mathcal{B}}\sum_{i=1}^{\mathcal{B}} -\frac{1}{2\eta}\|x_{t-1} - \boldsymbol{T}_{\theta,t-1}(x_{t-1})\|_2^2 + w_{\phi,t-1}(\boldsymbol{T}_\theta(x_t^{(i)})) + \frac{1}{\mathcal{B}}\sum_{j=1}^{\mathcal{B}} f^\star(-w_{\phi,t-1}(x_T^{(j)}))$.
5:     Sample a batch $\{x_{t-1}^{(i)}\}_{i=1}^{\mathcal{B}} \sim \{(x_{t-1}^{(i)}, y_{t-1}^{(i)})\}_{i=1}^N$.
6:     Update $\boldsymbol{T}_{\theta,t-1}$ by: $\theta \leftarrow \arg\min_\theta \frac{1}{\mathcal{B}}\sum_{i=1}^{\mathcal{B}} \frac{1}{2\eta}\|x_{t-1}^{(i)} - \boldsymbol{T}_{\theta,t-1}(x_{t-1}^{(i)})\|_2^2 - w_{\phi,t-1}(\boldsymbol{T}_{\theta,t-1}(x_{t-1}^{(i)}))$.
7: **end for**

---

For "Objective Functional", the detailed experimental protocols are given as follows:

---

**Algorithm 5** Barycentric-based training for $\{\boldsymbol{T}_{\theta,t}\}_{t=0}^{T-1}$.

---

**Input:** Intermediate domain samples: $\{(x_{t-1}^{(i)}, y_{t-1}^{(i)})\}_{i=1}^{N}$ for all $t \in \{1, \ldots, T\}$, target domain samples: $\{(x_T^{(i)}, y_T^{(i)})\}_{i=1}^{N}$, entropy regularization strength: $\epsilon$, step size: $\eta$, neural network batch size $\mathcal{B}$, and neural network training epochs: $\mathcal{E}$.

**Output:** The transportation map at $t-1$: $\boldsymbol{T}_{\theta,t-1}$.

1: Initialize
2: **for** $e = 1$ to $\mathcal{E}$ **do**
3:     Sample a batch $\{x_{t-1}^{(i)}\}_{i=1}^{\mathcal{B}} \sim \{(x_{t-1}^{(i)}, y_{t-1}^{(i)})\}_{i=1}^{N}$ and $\{x_T^{(i)}\}_{i=1}^{\mathcal{B}} \sim \{(x_T^{(i)}, y_T^{(i)})\}_{i=1}^{N}$.
4:     Obtain the optimal transport map $\pi^*(x_{t-1}, x_T)$ by: $\pi^*(x_{t-1}, x_T) \leftarrow \inf_\pi \frac{1}{2\eta} \mathcal{W}_2^2(\rho(x_{t-1}), p_T(x)) + \epsilon \iint \pi(x_{t-1}, x_T)[\log \pi(x_{t-1}, x_T) - 1]\mathrm{d}x_{t-1}\mathrm{d}x_T + \mathbb{D}_f[\rho(x_{t-1}), p_T(x)].$
5:     Obtain the projected samples $\tilde{x}_t$ via $\pi^*(x_{t-1}, x_T)$: $\tilde{x}_t = x_{t-1}\pi^*(x_{t-1}, x_T)$:
6:     Update $\boldsymbol{T}_{\theta,t-1}$ by: $\theta \leftarrow \frac{1}{\mathcal{B}} \sum_{i=1}^{\mathcal{B}} \|\tilde{x}_t^{(i)} - \boldsymbol{T}_{\theta,t-1}(x_{t-1}^{(i)})\|_2^2$
7: **end for**

---

- $\chi^2$ **Divergence:** The expression for $\chi^2$ divergence can be given as follows:

$$\mathbb{D}_{\chi^2}[\rho(x_t), p_T(x)] = \int p_T(x)[\frac{\rho(x_t)}{p_T(x)} - 1]^2 \mathrm{d}x_t \tag{135a}$$

$$f(x) = (x - 1)^2. \tag{135b}$$

Based on this, the corresponding conjugate function $f^\star$ can be given as follows:

$$f^\star(x) = \begin{cases} \frac{1}{4}x^2 + x, & \text{if } x \geq -2 \\ -1, & \text{if } x < -2 \end{cases}. \tag{136}$$

- **Identity:** For the identity function, we remove the $f$-divergence-based regularization term during the construction of E-SUOT framework. Based on this, the training objective for $w_\phi$ is reformulated as follows:

$$\mathcal{L}_{\text{Identity}}^{\text{E-SemiDual}} = \sup_w -\epsilon\mathbb{E}_{p(x_t)}\{\log \mathbb{E}_{p_T(x)}[\exp(\frac{w(x) - \frac{1}{2\eta}\|x - x_t\|_2^2}{\epsilon})]\} + \mathbb{E}_{p_T(x)}[w(x)], \tag{137}$$

- **SftPls:** We directly parameterize the $f^\star(x)$ by the following smooth, convex, and non-decreasing softplus function:

$$f^\star(x) = \log[1 + \exp(x)]. \tag{138}$$

## F. Additional Experimental Results

### F.1. Additional Results on Motivation Analysis

In our toy example analysis, we originally estimate the target domain PDF using denoising score matching (Vincent, 2011). To provide a more comprehensive analysis, we further incorporate several recent target domain PDF estimation approaches, including RealNVP (Dinh et al., 2017), Flow Matching (FM) (Lipman et al., 2023), the Kernel Stein Score Estimator (Li & Turner, 2018), the Spectral Stein Gradient Estimator (SSGE) (Shi et al., 2022), Gaussian Kernel Density Estimation (GKDE) (Li & Turner, 2018), variational mixtures of normalizing flows (VMoNF) (Pires & Figueiredo, 2020), and Gaussian Mixture Flow Matching Models (GMFlow) (Chen et al., 2025a). The results are shown in Figure 8.

As shown in Figure 8, most EstTrans approaches are unable to effectively transport source-domain samples to the target domain. In particular, Figures 8(a) to 8(e) and 8(g) demonstrate that the transported samples tend to be dispersed around the target distribution, leading to poor alignment. In contrast, approaches that perform sample-wise estimation, without explicitly estimating the PDF or directly smoothing the empirical samples, yield relatively lower Wasserstein distances, as shown in Figures 8(f) and 8(h). This phenomenon further validates our motivation for constructing the velocity field for sample transportation directly from samples.

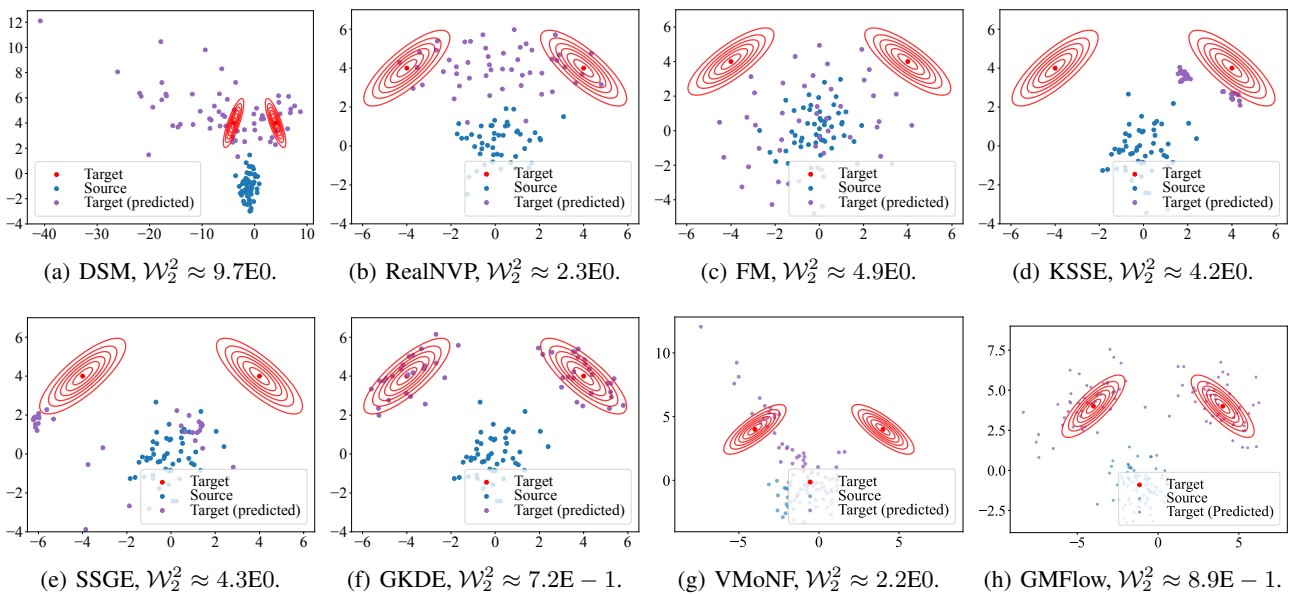

*Figure 8.* Comparison between different EstTrans approaches with respect to DirTrans.

## F.2. Sensitivity Analysis

From Figures 9(a) to 9(d), we systematically investigate the sensitivity of our E-SUOT model with respect to key hyperparameters, including batch size $\mathcal{B}$, discretization step size $\eta$, simulation steps $T$, and entropy regularization strength $\epsilon$ on the Portraits dataset.

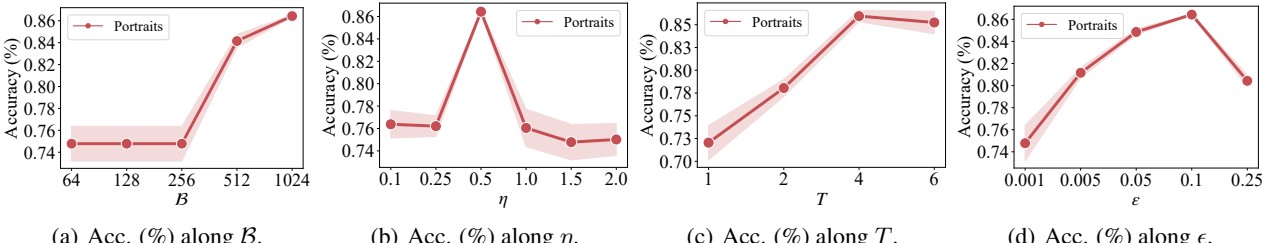

*Figure 9.* Sensitivity analysis results on Portrait Dataset. The shaded area indicates the $\pm 1.0$ times standard deviation error.

Specifically, as shown in Figure 9(a), increasing the batch size $\mathcal{B}$ consistently improves model performance. This suggests that, in simulations of WGF-based approaches (including ours), using a sufficiently large batch size is beneficial, as it reduces stochastic sampling noise and yields a more stable approximation of the underlying distributions.

A non-monotonic trend is observed when varying the discretization step size $\eta$, as illustrated in Figure 9(b). When $\eta$ is too small, the simulation trajectory may not reach the target distribution within a finite number of steps, limiting learning efficiency. In contrast, an overly large $\eta$ introduces substantial discretization error, which also degrades performance. These empirical observations are consistent with our theoretical guidance on step-size selection provided in Section C.8.

Moreover, as demonstrated in Figure 9(c), increasing the number of simulation steps $T$ likewise yields a non-monotonic effect: after a certain threshold, additional steps actually deteriorate performance. A plausible explanation is that enforcing excessively strict alignment between feature and target distributions does not necessarily correspond to optimal performance in the target domain. This observation further supports our introduction of divergence-based regularization, which relaxes strict alignment constraints compared with traditional OT-based methods.

Finally, as shown in Figure 9(d), the entropy regularization parameter $\epsilon$ has a pronounced impact on the results. Different values of $\epsilon$ can lead to markedly different performance, underscoring the need to carefully examine and tune the entropy

regularization strength in practice.

In summary, this sensitivity analysis highlights the importance of properly choosing the batch size $\mathcal{B}$, step size $\eta$, and end time $T$ for achieving good E-SUOT performance, and indicates that the optimal entropy regularization strength $\epsilon$ is dataset-dependent, calling for systematic validation on the target dataset.

### F.3. Impact of Early Transport Steps on Adaptation Performance

The multi-step structure of E-SUOT involves a series of learned transport maps $\{T_{\theta,0}, T_{\theta,1}, \ldots\}$, where each step aims to progressively align intermediate feature distributions between domains. While this design promotes smooth domain alignment, it also raises an important question: how sensitive the overall adaptation performance is to the quality of early transport maps, and whether suboptimal early mappings introduce cumulative errors that affect subsequent steps.

To investigate this, we conduct an experiment on the Portraits dataset, where we selectively disable the training of certain transport maps to simulate incomplete or inaccurate early-stage optimization. Two complementary training strategies are designed:

- **Forward strategy:** Progressively remove the training of early transport maps $(T_{\theta,0}, T_{\theta,1}, \ldots)$ while keeping the later ones active, thereby testing whether missing early steps hinder later GDA performance.
- **Backward strategy:** progressively disable the training of later maps $(T_{\theta,4}, T_{\theta,3}, \ldots)$ while retaining the trained early steps, examining whether well-trained initial stages are sufficient to sustain strong performance.

Table 8 summarizes the results. We observe that in the "Forward" direction, excluding early transport steps causes a substantial accuracy drop (up to 17.7%), indicating that early-stage mappings are crucial for forming a reliable transport foundation. In contrast, the "Backward" experiments show that once these early steps are properly optimized, subsequent refinements yield consistent performance gains, suggesting that later transport maps mainly provide fine adjustments on top of an already well-aligned feature space.

*Table 8.* Performance of the E-SUOT under different training stage on the Portraits dataset.

| Direction | | | Forward | | | | | | | Backward | | | | |
|---|---|---|---|---|---|---|---|---|---|---|---|---|---|---|
| Time Index | $t=0$ | $t=1$ | $t=2$ | $t=3$ | $t=4$ | Acc. (%) | $\Delta$ | $t=0$ | $t=1$ | $t=2$ | $t=3$ | $t=4$ | Acc. (%) | $\Delta$ |
| | ✗ | ✓ | ✓ | ✓ | ✓ | 76.4 | ↑7.2% | ✓ | ✗ | ✗ | ✗ | ✗ | 81.5 | ↑14.4% |
| | ✗ | ✗ | ✓ | ✓ | ✓ | 75.0 | ↑5.3% | ✓ | ✓ | ✗ | ✗ | ✗ | 83.2 | ↑16.8% |
| Training Status | ✗ | ✗ | ✗ | ✓ | ✓ | 74.4 | ↑4.4% | ✓ | ✓ | ✓ | ✗ | ✗ | 83.9 | ↑17.8% |
| | ✗ | ✗ | ✗ | ✗ | ✓ | 74.0 | ↑3.9% | ✓ | ✓ | ✓ | ✓ | ✗ | 84.0 | ↑17.9% |
| | ✗ | ✗ | ✗ | ✗ | ✗ | 58.6 | ↓17.7% | ✓ | ✓ | ✓ | ✓ | ✓ | 86.4 | ↑21.5% |

*Kindly Note*: $\Delta$ denotes performance change percentage of the initial classifier accuracy.

Overall, the results highlight that the early transport steps are the key drivers of successful adaptation. When the early mappings are well trained, the rest of the chain benefits from a stabilized feature representation, leading to larger and more consistent improvements. This behavior parallels diffusion-like processes, where the early transport transformations largely determine the shape and quality of the final distribution, as also illustrated in Figure 3 in reference (Caluya & Halder, 2020) and Figure 1 in reference (Liu & Wang, 2016).

### F.4. Investigation of the UOT Formulation

While our main contribution lies in introducing the semi-dual UOT formulation to analyze and improve the flow-based GDA approach, we further investigate "why the UOT-based method performs better than the vanilla OT formulation". To this end, we conduct experiments under label-shift and missing-class scenarios using the Portrait dataset (binary classification task). Specifically, we resample the target domain to vary the class prior $p(y = 1)$; when $p(y = 1) = 0.0$ or $p(y = 1) = 1.0$, a missing-class situation is realized. The comparison results between vanilla OT and UOT are reported in Table 9. Here, "E-SOT" denotes entropy-regularized semi-dual optimal transport. From the table, we observe that in the missing-class cases $(p(y = 1) = 0.0$ or $p(y = 1) = 1.0)$, the vanilla OT formulation not only fails to improve but even degrades the classifier's performance in the target domain. Moreover, under moderate label shift ($p(y = 1)$ between 0.3 and 0.9), the performance of

*Table 9.* Comparison of vanilla and unbalanced OT formulations for GDA task on Portraits dataset.

| Method | $p(y=1)=0.0$ | | $p(y=1)=0.1$ | | $p(y=1)=0.2$ | | $p(y=1)=0.3$ | | $p(y=1)=0.4$ | |
|---|---|---|---|---|---|---|---|---|---|---|
| | Acc. (%) | $\Delta$ | Acc. (%) | $\Delta$ | Acc. (%) | $\Delta$ | Acc. (%) | $\Delta$ | Acc. (%) | $\Delta$ |
| Initial | 35.4 | - | 41.0 | - | 47.1 | - | 53.2 | - | 59.6 | - |
| E-SOT | 55.4 | ↑56.41% | 61.1 | ↑48.94% | 67.1 | ↑42.55% | 56.7 | ↑6.54% | 57.7 | ↓3.22% |
| E-SUOT | 64.5 | ↑82.32% | 78.2 | ↑90.50% | 74.5 | ↑58.28% | 79.8 | ↑49.92% | 77.7 | ↑30.42% |

| Method | $p(y=1)=0.6$ | | $p(y=1)=0.7$ | | $p(y=1)=0.8$ | | $p(y=1)=0.9$ | | $p(y=1)=1.0$ | |
|---|---|---|---|---|---|---|---|---|---|---|
| | Acc. (%) | $\Delta$ | Acc. (%) | $\Delta$ | Acc. (%) | $\Delta$ | Acc. (%) | $\Delta$ | Acc. (%) | $\Delta$ |
| Initial | 73.8 | - | 80.0 | - | 85.9 | - | 91.4 | - | 97.1 | - |
| E-SOT | 75.2 | ↑1.95% | 74.1 | ↓7.33% | 80.0 | ↓6.89% | 89.9 | ↓1.65% | 96.0 | ↓1.08% |
| E-SUOT | 79.1 | ↑7.24% | 84.0 | ↑5.05% | 87.8 | ↑2.18% | 91.8 | ↑0.45% | 97.6 | ↑0.54% |

*Kindly Note*: $\Delta$ denotes performance change percentage compared to E-SUOT with entropy regularization and KL divergence. For source domain, $p(y=1)=0.63$. The acronym E-SOT stands for entropy-regularized semi-dual optimal transport.

vanilla OT fluctuates strongly and lacks stability, whereas UOT consistently improves performance. These observations demonstrate that incorporating the unbalanced OT formulation provides a more robust and effective approach for handling domain adaptation under label distribution mismatch.

## F.5. Further Analysis of Table 4

We have reported the main results in Table 4 of the main content. However, the improvements on Office-Home appear to be marginal for FixBi and CoVi. Therefore, to better understand when E-SUOT is most effective, we conduct a more detailed discrepancy-based analysis in Table 10. In this table, we report the accuracy change after applying E-SUOT to two UDA backbones, FixBi and CoVi, and analyze the results using three discrepancy measures: the 2-Wasserstein distance $\mathcal{W}_2$, the classwise Wasserstein distance $\mathcal{CW}_2$, and the Gromov-Wasserstein distance $\mathcal{GW}$. The definitions of the classwise Wasserstein distance and the Gromov-Wasserstein distance are given as follows. Specifically, the classwise Wasserstein distance is computed by first estimating the empirical class-conditional distributions in the source and target domains, and then averaging the class-level Wasserstein distances:

$$\mathcal{CW}_2 \coloneqq \sum_{c \in C_{\text{valid}}} \omega_c \mathcal{W}_2^2 \left( \hat{\mu}_s^c, \hat{\mu}_t^c \right), \qquad \hat{\mu}_s^c = \frac{1}{n_s^c} \sum_{i:x_s^{(i)}=c} \delta_{x_s^{(i)}}, \quad \hat{\mu}_t^c = \frac{1}{n_t^c} \sum_{j:x_t^{(j)}=c} \delta_{x_t^{(j)}}, \tag{139}$$

where $\omega_c$ denotes the normalized class weight, and $C_{\text{valid}}$ contains the classes with enough samples in both domains. The Gromov-Wasserstein distance is obtained as follows:

$$\mathcal{GW}(\mu_s, \mu_t) \coloneqq \min_{\pi \in \Pi(a,b)} \sum_{i,i'=1}^{n_s} \sum_{j,j'=1}^{n_t} [\|x_s^{(i)} - x_s^{(i')}\|_2^2 - \|x_t^{(j)} - x_t^{(j')}\|_2^2]^2 \pi_{ij} \pi_{i'j'}. \tag{140}$$

The Wasserstein distance measures the overall distributional gap between the source and target embeddings, the classwise Wasserstein distance reflects class-level semantic mismatch, and the Gromov-Wasserstein distance characterizes the discrepancy in the relational geometry of the two domains.

From Table 10, we observe that E-SUOT is more beneficial when the source and target domains share relatively compatible global and structural geometry. In such cases, reducing the global transport discrepancy can effectively improve cross-domain alignment without substantially distorting the intrinsic feature structure. For example, on the challenging Ar→Cl transfer, E-SUOT improves FixBi and CoVi by 6.2% and 5.3%, respectively. This transfer has relatively low global discrepancy compared with several other directions, and its structural discrepancy is also not excessively large. Therefore, the static endpoint coupling optimized by E-SUOT provides an effective correction to the pretrained feature extractor.

In contrast, E-SUOT may bring limited or even negative gains when the two domains exhibit large class-level or structural mismatch. For instance, on Cl→Pr and Rw→Cl, the improvements become negative for both FixBi and CoVi. These cases are associated with either large classwise Wasserstein distances or large Gromov-Wasserstein discrepancies, indicating that the semantic correspondence or relational geometry between domains is less compatible. Under such conditions, reducing

*Table 10.* Accuracy (%) improvement over different UDA feature extractor backbones and corresponding statistical analysis.

| | Method | Ar→Cl | Ar→Pr | Ar→Rw | Cl→Ar | Cl→Pr | Cl→Rw | Pr→Ar | Pr→Cl | Pr→Rw | Rw→Ar | Rw→Cl | Rw→Pr | Avg. |
|---|---|---|---|---|---|---|---|---|---|---|---|---|---|---|
| FixBi | vanilla | 58.1 | 77.3 | 80.4 | 67.7 | 79.5 | 78.1 | 65.8 | 57.9 | 81.7 | 76.4 | 62.9 | 86.7 | 72.7 |
| | +E-SUOT | 61.7 | 79.1 | 81.7 | 67.6 | 77.6 | 78.2 | 67.3 | 61.3 | 82.7 | 76.0 | 62.5 | 85.3 | 73.4 |
| | $\Delta$ | ↑6.2% | ↑2.3% | ↑1.6% | ↓0.1% | ↓2.4% | ↑0.1% | ↑2.3% | ↑5.9% | ↑1.2% | ↓0.5% | ↓0.6% | ↓1.6% | ↑1.0% |
| | $\mathcal{W}_2^2$ | 8.1 | 12.3 | 8.1 | 12.5 | 8.7 | 4.1 | 9.2 | 5.5 | 4.0 | 8.7 | 3.3 | 6.4 | - |
| | $\mathcal{CW}_2$ | 66.3 | 32.8 | 34.1 | 54.5 | 20.9 | 28.4 | 87.3 | 67.3 | 19.8 | 49.1 | 66.3 | 15.1 | - |
| | $\mathcal{GW}$ | 11.1 | 92.3 | 9.7 | 22.0 | 143.8 | 145.1 | 12.5 | 155.9 | 128.5 | 10.4 | 100.1 | 116.1 | - |
| CoVi | vanilla | 58.5 | 78.1 | 80.0 | 68.1 | 80.0 | 77.0 | 66.4 | 60.2 | 82.1 | 76.6 | 63.6 | 86.5 | 73.1 |
| | +E-SUOT | 61.6 | 79.3 | 81.8 | 67.6 | 77.7 | 78.1 | 67.4 | 61.2 | 82.9 | 76.3 | 62.5 | 85.2 | 73.5 |
| | $\Delta$ | ↑5.3% | ↑1.5% | ↑2.2% | ↓0.7% | ↓2.9% | ↑1.4% | ↑1.5% | ↑1.7% | ↑1.0% | ↓0.4% | ↓1.7% | ↓1.5% | ↑0.5% |
| | $\mathcal{W}_2^2$ | 6.8 | 11.0 | 8.7 | 12.2 | 8.8 | 3.9 | 11.0 | 5.8 | 6.4 | 6.7 | 4.9 | 4.8 | - |
| | $\mathcal{CW}_2$ | 55.5 | 23.4 | 31.6 | 60.6 | 22.9 | 25.9 | 81.1 | 83.5 | 22.3 | 54.3 | 72.5 | 14.2 | - |
| | $\mathcal{GW}$ | 15.1 | 96.1 | 78.9 | 20.5 | 130.1 | 167.0 | 76.1 | 144.2 | 101.5 | 77.7 | 132.5 | 82.7 | - |

*Kindly Note*: $\Delta$ denotes the relative accuracy change of E-SUOT over the corresponding vanilla UDA backbone.

the Wasserstein discrepancy alone may not be sufficient and can even lead to negative transfer, since samples from different classes may be incorrectly aligned or the intrinsic relational structure may be distorted.

It is also worth noting that E-SUOT does not rely on pseudo-labels or explicit class-level supervision during the transport optimization, and it does not directly exploit the relational geometry of the data. This is different from strong UDA methods such as FixBi and CoVi, which already incorporate additional regularization and task-specific adaptation mechanisms. Therefore, the improvement obtained by E-SUOT on top of these strong backbones is expected to be moderate. Nevertheless, the consistent average gains over both FixBi and CoVi suggest that E-SUOT can serve as a complementary module, especially for transfer tasks where the source and target domains have compatible global geometry and limited class-level mismatch.

### F.6. Computational Time Comparison

As shown in Figure 10, MMDSW, SMMD, and GOAT are generally among the most time-consuming methods, while GGF also incurs high cost on the smaller Portraits dataset. This is mainly due to their algorithmic designs: MMDSW and SMMD require repeated computation of distributional discrepancies based on pairwise distances, GOAT involves OT-related optimization, and GGF relies on a forward Euler discretization that typically requires many small steps to control simulation errors. In addition, AST and STDW have different computational bottlenecks. AST requires an additional backward pass to generate adversarial perturbations before each classifier update, while STDW combines sample evolution, rectified-flow updates, stochastic perturbations, and dynamically weighted source-target classifier training. These components increase their runtime compared with simple self-training. In contrast, E-SUOT remains computationally efficient and stable across datasets. It directly parameterises the transport map with a neural network and generates transported samples via a single forward pass. Together with the JKO scheme, which requires only a few backward discretization steps, E-SUOT achieves low runtime while maintaining the best adaptation performance.

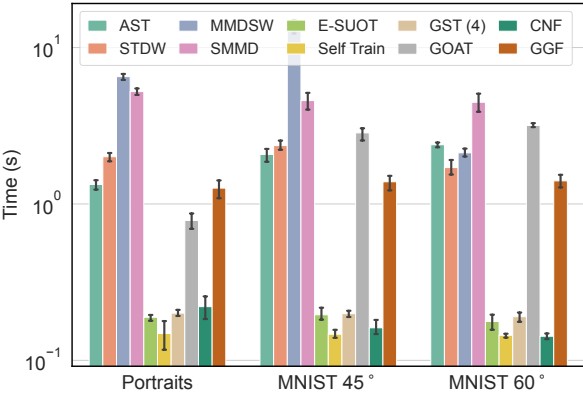

*Figure 10.* Computational time (s).

# G. Discussions on Limitations and Future Directions

The limitations and future research directions of this work can be summarized as follows:

- **Consideration of Label Information:** In this work, we focused primarily on feature adaptation and did not explicitly incorporate label or discriminator information into the adaptation process. As a result, the performance of the proposed E-SUOT framework may degrade under scenarios involving significant covariate shift (Sugiyama & Kawanabe, 2012; Sugiyama et al., 2007). An important direction for future research is to integrate label information into the transportation process, for example, classifier-guidance approach (Courty et al., 2017a; Dhariwal & Nichol, 2021; Zhuang et al., 2024), which could further enhance model robustness and adaptation performance.

- **Regularization of the Transportation Plan:** To facilitate computation, we introduced entropy regularization on the transport plan; however, this may introduce potential instability or blur sparsity in the map (Yin et al., 2025). Future work may explore alternative regularization strategies (Courty et al., 2014; 2017b), such as group sparsity (to better incorporate label priors) or Laplacian regularization (to preserve local relationships), in order to further stabilize training and improve the properties of the learned potential function $w$.

- **Exploration of Other Discrepancy Measures:** In this work, we adopted the Wasserstein distance as the primary metric for measuring domain discrepancy. However, other discrepancy measures, such as the Fisher-Rao distance (Wang et al., 2023; Zhang et al., 2022; Zhu, 2025) and Bures–Wasserstein discrepancy (Lyu et al., 2026; Wang et al., 2026a;b), could also be explored to enable more flexible or principled adaptation approaches. Future work may investigate the use of alternative metrics (Neklyudov et al., 2023; Skreta et al., 2025) to further improve the quality of intermediate domains and thereby enhance GDA performance.

- **Assumption of Label Invariance Along the Transportation Path:** The current formulation assumes that labels remain invariant during adaptation, i.e., $y_{t+1} \leftarrow y_t$, and thus primarily focuses on aligning the marginal feature distributions $p(x_t)$. This assumption may limit performance under pronounced label shift scenarios, where the conditional relationship $p(y|x)$ varies across domains, a case often encountered in unbalanced or fine-grained settings. Although our framework can be extended by incorporating classifier uncertainty or pseudo-label refinement (drawing inspiration from self-training schemes such as Equations (3) and (4) in reference (Kumar et al., 2020)), handling substantial concept drift remains an open challenge. Future work may consider integrating adaptive label transport (Courty et al., 2017a) or uncertainty-aware pseudo-labeling (Kumar et al., 2020; Zhuang et al., 2024) to explicitly account for label-shift dynamics along the adaptation trajectory.

- **Higher Efficiency Utilization of Neural Networks:** In our current design, each stage requires training three separate networks, namely $w_\phi$, $\boldsymbol{T}_\theta$, and $h_\omega$, to generate each intermediate domain. Although this strategy can save computation time compared to existing approaches that perform intermediate-domain generation online during the domain adaptation stage, it may still be suboptimal in terms of overall training efficiency in the offline stage. A promising future direction is to reformulate the training of $\boldsymbol{T}_\theta$ into a more parameter-efficient form, such as adopting a LoRA-style adaptation (Hu et al., 2022; Zhuang et al., 2025) or using the reparameterization trick to parameterize the differences between different stages (Choi et al., 2024). For $w_\phi$ and $h_\omega$, one possible improvement is to finetune the last layer with variational Bayesian techniques (Brunzema et al., 2025; Harrison et al., 2024; Rudner et al., 2022), which could further reduce training cost.

