# OpenReview forum: "Rethinking the Flow-based Gradual Domain Adaptation: A Semi-Dual Optimal Transport Perspective"
_ICML.cc/2026/Conference — ICML 2026 regular_

### Official Review · Reviewer_eCZE · 2026-03-13

**Soundness:** 2
**Presentation:** 2
**Significance:** 3
**Originality:** 2
**Overall Recommendation:** 5
**Confidence:** 5

**Summary:**

This paper revisits flow-based gradual domain adaptation (GDA) and argues that existing flow-based approaches are suboptimal because they typically rely on explicit estimation of the target-domain density or score, which can be inaccurate and degrade the quality of synthesized intermediate domains. To avoid this dependence, the paper reformulates the discrete gradient-flow step as a semi-dual optimal transport problem that uses only expectations over samples rather than explicit density evaluation. Because the resulting semi-dual problem has a $\sup$-$\inf$ structure that the authors argue is unstable and non-identifiable, they introduce an entropic regularization term and derive an entropy-regularized semi-dual objective depending only on a potential $w$. Empirically, the method is evaluated on Portraits, MNIST-$45^\circ$, MNIST-$60^\circ$, and Office-Home, where it outperforms several GDA/UDA baselines and shows favorable ablations on entropy regularization, choice of $f$-divergence, and UMAP embedding dimension.

**Compliance With Llm Reviewing Policy:**

Affirmed.

**Final Justification:**

I have independently implemented STDW and AST, and found the results are largely consistent with those reported in the paper. I have also reviewed the authors' code, and now consider the experimental validation of this work to be convincing.

**Key Questions For Authors:**

Please refer to the weaknesses.

**Limitations:**

Yes

**Strengths And Weaknesses:**

**Strengths.**
The paper targets a meaningful weakness of recent flow-based GDA methods: the practical dependence on explicit target-density or score estimation even when only target samples are available. Recasting the intermediate-domain generation step as a semi-dual transport problem is conceptually interesting, and the move from explicit density estimation to sample-based expectation objectives is a sensible direction. The overall algorithm is also relatively easy to understand once the derivation is accepted.

**Weaknesses.**
1. Theorem 3.5 is too weak to support the paper’s narrative that repeated transport progressively approaches the target distribution.
Theorem 3.5 states that the optimizer $\rho^\star(x)$ of Eq. (8) satisfies
$$
D_f[\rho^\star(x),p_T(x)]\le W_2(p(x_t),p_T(x)).
$$
But this is only a one-step inequality relating the divergence of the newly optimized distribution to the Wasserstein distance between the current distribution and the target. By itself, it does not imply a monotone decrease in either $D_f$ or $W_2$ across iterations, nor does it establish convergence of the iterated learned transport maps under function approximation. Yet the paper informally concludes from this theorem that “as $t$ increases, the transported density $\rho(x)$ progressively approaches the target distribution.” That interpretation seems stronger than the displayed bound supports.

2. The target-generalization theorem is too high-level and hides most of the substantive difficulty inside undefined aggregate terms.
Theorem 3.6 gives
$$
\varepsilon_{p_T}(h_T)\le \varepsilon_{p_0}(h_0)+\varepsilon_{p_0}(h_T^\star)+\iota\zeta C+S_{\mathrm stat},
$$
where $C$ “aggregates the cumulative domain transportation and label continuity costs” and $S_{\mathrm stat}$ is a statistical error term. This kind of bound is directionally reasonable, but in its current form it does not tell the reader much about what properties of E-SUOT actually control target performance, or how the proposed method compares to existing GDA bounds.

3. Please consider comparing your method with the following approaches:

[1] Self-Training with Dynamic Weighting for Robust Gradual Domain Adaptation, https://www.arxiv.org/abs/2510.13864

[2] Adversarial Self-Training Improves Robustness and Generalization for Gradual Domain Adaptation. Lianghe Shi, Weiwei Liu. NeurIPS 2023.

---

> ### Author Rebuttal · Authors · 2026-03-28
>
> **Thank you greatly for your time and valuable feedback on our manuscript!** Below is our response to your comments, and we promise that we will add these modifications in the main contents.
> ## W1 [Statement of Thm 3.5]
> - **Theoretical Analysis**:
>     - Based on Eq.(4.0.12) in Thm.4.0.4 of ref [1], we have $\frac{1}{2\eta}\mathcal{W}\_2^2(p(x\_{t+1}),p(x\_t))+\mathbb{D}\_f[p(x\_{t+1}),p\_T(x)]\le\mathbb{D}\_f[p(x\_t),p\_T(x)]$. Since the $\mathcal{W}\_2^2\ge0$, *we obtain: $\mathbb{D}\_f[p(x\_{t+1}),p\_T(x)]\le\mathbb{D}\_f[p(x\_t),p\_T(x)]$*. Thus, $\mathbb{D}\_f$ is *non-increasing* as $t$ increases.
>     - Thm. 3.5 in our manuscript provides a complementary interpretation: *the $\mathcal{W}\_2^2(p(x\_{t+1}),p(x\_t))$ controls the transport cost between consecutive iterates*, showing that each update trades off proximity to the previous iterate against reduction of the $\mathbb{D}\_f$ to the target.
>     - Based on the abovementioned analysis, we arrive at our statement in the sense of $\mathbb{D}\_f$.
> - **Case Study**:
>     - In additionally, *a solution process for a toy case study is provided for illustration*. Please refer to Fig. R5 at the link: https://anonymous.4open.science/r/ESUOTICML2026Rebbutal-CE63/ESUOTLink.md
> ## W2 [Statement of Thm 3.6]
> We treat flow-based GDA as "estimate-then-transport", and E-SUOT as "direct-transport".
> - **Bound Estimation**:
>
>   * $\iota$: Based on Assum. (1), we have $\iota<1$, which can be realized by weight normalization.
>   * $\zeta$: In our implementation, we have:$ \zeta\le\prod_{\ell=1}^L\|W_\ell\|_2$, where $L$ is classifier layer number, and $W$ is the weight.
>   * $\mathcal{C}$: This term consists of the Wasserstein distance between $p_t$ and $p_{t-1}$, together with the inter-step labeling-function shift term $\mathbb{E}\_{p_t} | q_t(x) - q_{t-1}(x)| $, which can be approximated using pseudo-labels.
>   * $\mathcal{S}\_{stat}:$ Based on sec. 9.2 of ref [2], we have: $S\_{stat}\le\frac{2CGD}{\sqrt{N}} (T-1)$, where $N$ is the training sample size, $G$ is the Lipschitz constant of the loss function, $D$ controls the function class complexity, and $C$ is a universal constant arising from the Rademacher complexity bound.
> - **Improvement for existing flow-based GDA**:
>
>   * The Wasserstein bound for the JKO proximal recursion is controlled by: $\frac{1}{2\eta}\mathcal{W}\_2^2(p(x\_{t+1}),p(x\_t))+\mathbb{D}\_f[p(x\_{t+1}),p\_T(x)]\le\mathbb{D}\_f[p(x\_t),p\_T(x)]+\varepsilon\_{density}+\varepsilon\_{opt}\le \mathcal{W}\_2^2(p(x\_{t+1}),p(x\_t))+\varepsilon\_{density}+\varepsilon\_{opt}$, where $\varepsilon\_{density}$ is the density estimation error, and $\varepsilon\_{opt}$ is numerical optimization error.
>   * For E-SUOT, we solve the duality problem and avoid the $\varepsilon\_{density}$, whereas flow-based GDA methods still suffer from $\varepsilon\_{density}$. As a result, the $\mathcal{C}$ is greater for flow-based GDA approach.
>   * The $\mathcal{W}$ is related to term $\mathcal{C}$ and E-SUOT improves this term compared with existing flow-based GDA.
> ## W3 [Additional Baselines Approaches]
> Based on your comments and those from Reviewer 8u2Z, we have added additional baselines including MMDSW [3], SMMD [4], STDW [5], and AST [6] on the GDA and UDA tasks.
> - GDA Task:
> Method|Portraits|MNIST45$^\circ$|MNIST60$^\circ$
> -|-|-|-|
> MMDSW|80.2|57.9|42.2
> SMMD|79.8|57.9|42.2
> STDW|84.3|60.3|43.9
> AST|84.2|58.3|41.3
> E-SUOT|86.4|72.1|51.0
> - UDA Task:
> |Method|Ar$\to$Cl|Ar$\to$Pr|Ar$\to$Rw|Cl$\to$Ar|Cl$\to$Pr|Cl$\to$Rw|Pr$\to$Ar|Pr$\to$Cl|Pr$\to$Rw|Rw$\to$Ar|Rw$\to$Cl|Rw$\to$Pr|Avg.
> -|-|-|-|-|-|-|-|-|-|-|-|-|-|
> MMDSW|59.2|75.6|81.7|67.6|77.4|77.9|67.4|61|82.6|75.9|62.4|85.4|72.8
> SMMD|58.8|74.8|81.7|67.6|77.4|77.9|67.4|61|82.6|75.9|62.4|85.4|72.7
> STDW|59.5|75.6|81.7|67.6|77.6|78|67.4|61|82.7|75.9|62.5|85.4|72.9
> AST|58.9|75.4|81.7|67.6|77.4|77.9|67.4|61|82.6|75.9|62.4|85.4|72.8
> E-SUOT|61.6|79.3|81.8|67.6|77.7|78.1|67.4|61.2|82.9|76.3|62.5|85.2|73.5
>
> **We hope our response have addressed your concerns. We would appreciate your kind reconsideration of the score. Thank you once again for your time and effort.**
>
> ---
> Refs:
> [1]. Gradient flows in metric spaces and in the space
> [2]. Learning Theory from First Principles
> [3]. Flowing Datasets with Wasserstein over Wasserstein Gradient Flows
> [4]. Generative Sliced MMD Flows with Riesz Kernels
> [5]. Self-Training with Dynamic Weighting for Robust Gradual Domain Adaptation
> [6]. Adversarial Self-Training Improves Robustness and Generalization for Gradual Domain Adaptation

---

> > ### Author Rebuttal · Reviewer_eCZE · 2026-04-02
> >
> > Thank you for your response. I have independently implemented STDW and AST, and found the results are largely consistent with those reported in the paper. I have also reviewed the authors' code, and now consider the experimental validation of this work to be convincing.
> > Although the rebuttal appears somewhat AI-assisted with occasional awkward phrasing, the overall quality of the paper and the experimental results are satisfactory. Therefore, I will raise my evaluation score.

---

> > > ### Author Response · Authors · 2026-04-03
> > >
> > > Dear Reviewer eCZE,
> > >
> > > **Thank you for your careful review** and for taking the time to independently implement STDW and AST. We greatly appreciate your thoughtful follow-up and are *glad that our response and experimental validation have addressed your concerns.*
> > >
> > > **Thank you again for your time and effort.**
> > >
> > > Sincerely,
> > > Authors of Submission 6648

---

### Official Review · Reviewer_ZJbU · 2026-03-13

**Soundness:** 3
**Presentation:** 3
**Significance:** 2
**Originality:** 3
**Overall Recommendation:** 5
**Confidence:** 4

**Summary:**

This work formulates the flow-based gradient domain adaptation as a dual Lagrangian in terms of Wasserstein-2 distance, to circument the requirement of explicit PDF estimation with flow-based methods. This term consists of optimizing the transport map T realized as a neural network and the continuous function W. An entropy term is introduced to stabilize the optimal transport formulation, which provides for a unique solution. The proposed approach is computationally expensive. It achieves state-of-the-art results over GDA  methods on four datasets including MNIST and office-home dataset for scalability.

**Compliance With Llm Reviewing Policy:**

Affirmed.

**Final Justification:**

The authors have addressed all my concerns. This paper is a good contribution to gradual adaptation or generation of intermediate  via a  sem-dual formulation of the optimal transport. The proposed approach as highlighted in the rebuttal on a toy example shows better sample quality. The approach, however, incurs much higher computational cost as is evident from the rebuttal.

**Key Questions For Authors:**

1. How are the classifiers for intermediate representations realized or trained? When are these classifiers considered to deliver oracle performance?
2. The computational complexity of the algorithm in the dual form seems to be very high compared to prior work. What explicit advantage does the approach offer over prior GDA-based approaches?
3.  What is the convergence criterion apart from the number of training epochs? How many samples compared to prior work are required to achieve convergence with the dual formulation?
4. What are the properties of the obtained transport maps with the proposed approach? The intermediate maps may not be affine or the smallest distance. How does entropy regularization affect the quality of the transport maps?

**Limitations:**

yes

**Strengths And Weaknesses:**

Strengths:
1. The paper is well-written and provides a solid theoretical framework for optimal transport-based gradual domain adaptation.
2. An entropy term is introduced to ensure that optimal transport yields a unique solution, which is mathematically sound.
3. The proposed GDA framework is supported by an empirical evaluation of MNIST and UDA  on the Office-Home datasets. The results are comparable, if not better than, the state of the art.

Weaknesses:
1. The proposed algorithm, given that it relies on the dual formulation, is computationally expensive.
2. The approach does not yield better results on UDA compared to prior work on UDA.
3. Figure 1 is not so clear. The evolution of the transport maps across different steps is difficult to infer from the figure.

---

> ### Author Rebuttal · Authors · 2026-03-30
>
> **Thank you greatly for your valuable feedback on our manuscript!** Below is our response to your comments. In addition, we relase an anonymous link to present figures: https://anonymous.4open.science/r/ESUOTICML2026Rebbutal-CE63/ESUOTLink.md.
>
> ### W1 & Q2 [Advantages]
> We agree that solving the dual formulation incurs higher per-iteration cost compared to prior GDA methods. However, this comparison overlooks key advantages in solution quality, stability, and extensibility.
> - **Solution Quality**: As represented by GOAT, which uses Sinkhorn-based OT to transport samples, introduces entropic bias and numerical issue due to exponential scaling, leading to suboptimal transport costs. *In our toy study (see next point), Sinkhorn yields significantly higher cost than the ground truth (e.g., 0.3971 vs 0.3350)*, while *dual approach achieves much closer estimates (0.3381)*, demonstrating improved solution quality.
> - **Density Estimation Error**: Second, compared to flow-based GDA methods, our method avoids explicit density estimation, (see response to `Reviewer eCZE`, W3), which yields better generalization error bound.
> - **Extrapolation:** When new samples arrive, we can reuse learned potentials without recomputing the full OT plan, which leads to favorable amortized efficiency.
>
> ### W2 [UDA]
> - Our method is *developed for the GDA setting*, and *strong UDA methods often rely on pseudo-labeling, which may implicitly alter the conditional distribution*, while E-SUOT did not make such assumptions.
> - Therefore, as stated in the preliminaries, our goal is not to prove that GDA methods are superior to UDA methods.
> - Nevertheless, as shown in Table 4, our method consistently improves multiple UDA backbones.
>
> |UDA Method| Avg. w/o E-SUOT|Avg. w E-SUOT|
> |-|-|-|
> |MSTN|65.7|71.0|
> |RSDA|70.9|73.3|
> |FixBi|72.7|73.4|
> |CoVi|73.1|73.5|
>
> ### W3 [Fig. 1]
> We provide an animated version of Fig. 1 to better illustrate the transport process; please refer to Fig. R2 in the above link.
>
> ### Q1 [Classifier]
> - **Training** The classifiers for intermediate domains are trained in a sequential manner. Specifically, the classifier at each stage is directly inherited from the previous stage and continues to be updated through the adaptation process on the current intermediate domain [1].
> - **Orcale Performance**: Regarding oracle performance, in the GDA setting (as in UDA), intermediate and target domains are unlabeled, and thus no ground-truth supervision is available. Therefore, an oracle classifier (i.e., trained with true target domain labels) is not defined in our setting.
>
> ### Q3 [Quality of Duality]
> - **Convergence criterion:** In practice, convergence is assessed by monitoring the transport cost. Since $\boldsymbol{T}^\star$ is learned to minimize this cost, we estimate it across iterations. The method is considered converged when $\Vert\boldsymbol{T}^\star(x)-x\Vert_2^2$ becomes stable.
> - **Sample size:** In theory, the primal and dual problems require the same sample size to converge. Below, we compare sample size and transport cost quality on a toy example. Additional visualizations of the dataset and convergence under different sample sizes are provided in Fig. R4 in the above link.
> |Size |Sinkhorn Time (ms) |Dual Time (ms) |Marginal Err (Sink) |Marginal Err (Dual) |Cost (Sink) |Cost (Dual) |Cost (Ground Truth)
> |-|-|-|-|-|-|-|-|
> |50|0.98±0.08|1014.7±69.1|2.11e-18|7.54e-06|0.3971|0.3381|0.3350|
> |100|2.95±2.08|1043.9±65.5|1.24e-18|3.42e-06|0.3667|0.3109|0.3078|
> |200|9.12±5.23|1005.0±80.5|6.36e-19|1.62e-06|0.3224|0.2734|0.2706|
> |500|25.50±17.38|1068.1±44.7|2.28e-19|5.53e-07|0.3298|0.2801|0.2770|
>
> ### Q4 [Transport Map]
> To better illustrate the effect of the transport plan, we added Fig. R3 in the link provided above. Below contents are our analysis:
> - **Plan Properties with Entropy Regularization:**
>    - The transport map tends to follow smoother paths in practice. Specifically, we denote the transport plan as $\pi$, and formulate the objective as $\inf_{\pi}\ell\coloneqq C^\top \pi+\mathbb{D}_{f}[\pi_y,\nu]+\epsilon\mathbb{H}(\pi)$. From the Lagrangian multiplier condition $\frac{\partial \ell}{\partial \pi}=0$, the optimal plan takes the Sinkhorn form: $\pi=u\exp(-C/\epsilon)v^\top$ [2], where $u$ and $v$ are normalization factors.
>    - Due to the entropy term, the solution tends to be dense, since $\exp(-C/\epsilon)>0$ always holds. As a result, the transport plan is typically *smooth* and *diffuse* rather than sparse.
> - **Influence of Entropy Regularization Strength:** As $\epsilon$ increases, the plan increasingly avoids zero entries, becoming more diffuse and spread out. See Fig. R3 (b) to (e) across different regularization strengths.
>
> **We hope our response have addressed your concerns. We would appreciate your kind reconsideration of the score. Thank you once again!**
>
> ---
> Refs.
> [1]. Gradual Domain Adaptation via Gradient Flow
> [2]. Gradient Flow Algorithms for Density Propagation in Stochastic Systems

---

> > ### Author Rebuttal · Reviewer_ZJbU · 2026-04-02
> >
> > The authors have addressed all my concerns. I will update my score
> > The figures and the extended analysis provided in the rebuttal should be included in the main paper or appendices.

---

> > > ### Author Response · Authors · 2026-04-03
> > >
> > > Dear Reviewer ZJbU,
> > >
> > > **Thank you for your time and effort in reviewing our manuscript.** We are glad that we have addressed your concerns. **We also appreciate your helpful suggestion** regarding the figures and extended analysis, and **we will incorporate them into the final version.**
> > >
> > > **Thank you once again!**
> > >
> > > Sincerely,
> > > Authors of Submission 6648

---

### Official Review · Reviewer_8u2Z · 2026-03-19

**Soundness:** 4
**Presentation:** 3
**Significance:** 2
**Originality:** 3
**Overall Recommendation:** 5
**Confidence:** 4

**Summary:**

This paper proposes  E-SUOT, a generative gradual domain adaptation method that generates intermediate domains using semi-dual optimal transport. Compared to previous gradient flow based methods, E-SUOT does not rely on explicit target density estimation, thus offering more stable performance. The paper provides theoretical guarantees for the convergence and generalization of the proposed method. Experiments on standard domain adaptation benchmarks show that the proposed method achieves the best average performance.

**Compliance With Llm Reviewing Policy:**

Affirmed.

**Final Justification:**

My concerns have been mostly addressed. I have increased my score.

**Key Questions For Authors:**

See Weaknesses 1-3.

**Limitations:**

Yes

**Strengths And Weaknesses:**

Strength
- It recasts flow evolution as an optimization
problem that combines an f -divergence term with a Wasserstein distance regularization term, leading to a novel GDA formulation.

- It improves the original semi-dual optimal transport by adding entropy regularization and provide additional theoretical analysis on the uniqueness of solution.

- Experiments show that E-SUOT has better average performance than existing methods, meaning that it is more stable.

- The presentation is clear with comprehensive background and theoretical analysis

Weakness
-  The main motivation that the author mentioned is that flow-based method suffers from poor density estimation of the target data. The motivating example, however, was from 2011 based on simple denoising AE for estimating density. It could be more convincing if more recent methods are used (ones that can estimate multimodal density or uses local estimation)
- The performance gain is not very strong on Office-Home. Compared to other baselines, it is the second best in 4/12 cases and appears weaker in the remaining cases. More explanations are needed regarding what kind of transfer tasks it performs well on, and those it does not.
- There exists other flow based methods that do not depend on density estimation. e.g. MMD flow based methods [1],   It is used in Flowing Datasets with Wasserstein over Wasserstein Gradient Flows by Bonet et. al., which also discusses domain adaption, and with classifier guidance. The author should compare the semi-dual OT approach with such alternatives.
- More challenging, realistic datasets should be considered, e.g. DomainNet or datasets with larger domain gaps, that could justify the need for gradual domain adaptation more. .


[1] Generative Sliced MMD Flows with Riesz Kernels by Hertrich et. al.

---

> ### Author Rebuttal · Authors · 2026-03-29
>
> **Thank you once again for your time and effort for reviewing our manuscript!** Below contents are our response to your comments:
> ### [W1] Approaches for "EstTrans"
> - We have added additional methods to validate efficacy, including RealNVP [1], Flow Matching (FM) [2], Kernel Stein Score Estimator (KSSE) [3], Spectral Stein Gradient Estimator (SSGE) [4], and Gaussian Kernel Density Estimation (GKDE) [3].
> - We report the overall Wasserstein distances (including DSM and OT-Map from our submitted manuscript) in the supplementary figures provided via this anonymous link: https://anonymous.4open.science/r/ESUOTICML2026Rebbutal-CE63/ESUOTLink.md.
>
> Method|DSM|RealNVP|FM|KSSE|SSGE|GKDE|OT Map|
> -|-|-|-|-|-|-|-
> $\mathcal{W}_2^2$|9.7E0|2.3E0|4.9E0|4.2E0|4.3E0|7.2E-1|7.8E-4
> ### [W2] Performance on UDA Task
> Based on part of Table 2, we further report the Wasserstein distance $\mathcal{W}_2$, the classwise Wasserstein distance $\mathcal{W}_2$ (computed per class), and the Gromov-Wasserstein distance $\mathcal{GW}$. The proposed E-SUOT does not use any pseudo-label information, and *does not exploit* the geometric structure of the data, *unlike existing methods such as CoVi, FixBi, and CST*.
> * **When perform well?** Our approach tends to be more effective when the source and target domains share *similar global geometry and structure*. In such cases, reducing global $\mathcal{W}_2^2$ is often associated with alignment without distorting the feature space. Empirically, tasks like Ar$\to$Cl show clear gains (e.g., +5.3%), where the structural discrepancy ($\mathcal{GW}$) remains relatively low, indicating that alignment is achieved without damaging the intrinsic geometry.
> * **When performance gains are negative?** On transfers like Cl$\to$Pr and Rw$\to$Cl, where the domains exhibit significant structural and semantic discrepancies, we observe higher $\mathcal{GW}$ and classwise $\mathcal{W}_2^2$, indicating larger structural and semantic discrepancies. In these cases, reducing global $\mathcal{W}_2^2$ alone does not necessarily translate into performance improvements, and may coincide with negative transfer. This is consistent with:
>     - misalignment at the class level (high classwise $\mathcal{W}_2^2$), and
>     - potential distortion of relational structure (large $\mathcal{GW}$ values, often >100).
>
> Method|Ar$\to$Cl|Ar$\to$Pr|Ar$\to$Rw|Cl$\to$Ar|Cl$\to$Pr|Cl$\to$Rw|Pr$\to$Ar|Pr$\to$Cl|Pr$\to$Rw|Rw$\to$Ar|Rw$\to$Cl|Rw$\to$Pr
> -|-|-|-|-|-|-|-|-|-|-|-|-|
> FixBi $\Delta$|$\uparrow$6.2\%|$\uparrow$2.3\%|$\uparrow$1.6\%|$\downarrow$0.1\%|$\downarrow$2.4\%|$\uparrow$0.1\%|$\uparrow$2.3\%|$\uparrow$5.9\%|$\uparrow$1.2\%|$\downarrow$0.5\%|$\downarrow$0.6\%|$\downarrow$1.6\%
> FixBi classwise $\mathcal{W}_2^2$|66.3|32.8|34.1|54.5|20.9|28.4|87.3|67.3|19.8|49.1|66.3|15.1
> FixBi $\mathcal{W}_2^2$|8.1|12.3|8.1|12.5|8.7|4.1|9.2|5.5|4|8.7|3.3|6.4
> FixBi $\mathcal{GW}$|11.1|92.3|9.7|22|143.8|145.1|12.5|155.9|128.5|10.4|100.1|116.1
> CoVi $\Delta$|$\uparrow$5.3\%|$\uparrow$1.5\%|$\uparrow$2.2\%|$\downarrow$0.7\%|$\downarrow$2.9\%|$\uparrow$1.4\%|$\uparrow$1.5\%|$\uparrow$1.7\%|$\uparrow$1.0\%|$\downarrow$0.4\%|$\downarrow$1.7\%|$\downarrow$1.5\%
> CoVi classwise $\mathcal{W}_2^2$|55.5|23.4|31.6|60.6|22.9|25.9|81.1|83.5|22.3|54.3|72.5|14.2
> CoVi $\mathcal{W}_2^2$|6.8|11|8.7|12.2|8.8|3.9|11|5.8|6.4|6.7|4.9|4.8
> CoVi $\mathcal{GW}$|15.1|96.1|78.9|20.5|130.1|167|76.1|144.2|101.5|77.7|132.5|82.7
>
> ### [W3] Flow-based Baseline Approaches
> Please see our response to Reviewer eCZE on `W3 [Additional Baseline Approaches]` for further details.
>
> **We hope our response have addressed your concerns. We would appreciate your kind reconsideration of the score. Thank you for your time and effort.**
>
> ### [W4] Datasets
> We agree that evaluating on more challenging datasets, such as DomainNet, would be valuable and could further justify the need for GDA. *As stated in our preliminaries, this work focuses on analyzing flow-based GDA itself*. The current Office-Home already exhibits non-trivial domain shifts and allows us to highlight the advantages of our method. Our method does not rely on dataset-specific assumptions, and we consider scaling to larger and more challenging datasets an important direction for future work.
>
> **We hope our response have addressed your concerns. Thank you once again for your time and effort.**
>
> ---
> Refs
> [1]. Density estimation using Real NVP
> [2]. Flow matching for generative modeling
> [3]. Gradient estimators for implicit models
> [4]. A spectral approach to gradient estimation for implicit distributions

---

> > ### Author Rebuttal · Reviewer_8u2Z · 2026-04-05
> >
> > I appreciate the effort and the detailed response provided by the authors. However, I still have unresolved concerns:
> >
> > W1: While the authors had added several baselines, including local density estimation method, yet none of them are  designed to handle multi-modal distributions (the second point i made). There are many works on these aspect that should be discussed: e.g.
> > - Semi-Supervised Learning with Normalizing Flows (2019)
> > - Variational Mixture of Normalizing Flows (2020)
> > - Gaussian Mixture Flow Matching Models (2025)
> >
> > W2: The authors explained that the relatively less significant improvement compared to related works is due to the fact that the proposed method does not use any pseudo-label information, and does not exploit the geometric structure of the data in UDA experiments. However, this design choice seems to be self-limiting, especially because the paper is focused on the applied problem of gradual domain adaptation rather than introducing new theory of optimal transport. In GDA, exploiting geometric structure is indeed a common strategy -- and assuming source and target domains share similar global geometry and structure goes against the current focus/challenge of the domain adaptation field. Maybe in the future the authors could show evidence that the proposed method can be incorporated with other techniques to exploit the geometric structure of data, or considering a more suitable application scenario than the generic UDA benmarks.
> >
> > W3: This question has been mostly addressed.
> > W4: My view is also linked to my response to W2. Even when we only focus on the OfficeHome dataset, the improvement doesn't seem to be significant enough.
> >
> > Due to these concerns, I am maintaining my already positive score.

---

> > > ### Author Response · Authors · 2026-04-08
> > >
> > > Dear Reviewer 8u2Z,
> > >
> > > **Thank you for your constructive feedback.** Our further responses to your comments are listed as follows:
> > >
> > > * **W1:** We appreciate your valuable feedback and suggested references. We have added FlowGMM [1] and GMFlow [2] as additional baselines, where FlowGMM is categorized under "EstTrans" and GMFlow under "DirTrans" in our motivation study. The updated figure is available via the abovementioned anonymous link. The corresponding $\mathcal{W}_2$ values are FlowGMM (2.22E0) and GMFlow (8.92E-1). By introducing the Gaussian mixture path, *we achieve a more accurate approximation to the target distribution compared with the vanilla denoising score matching loss.* We will include these methods to better organize and refine our motivation analysis.
> > > * **W3:** We are glad that our additional baselines have addressed your concern.
> > > * **W2 & W4:** We agree that the performance improvement over existing methods is relatively modest, which mainly stems from our design choice: the proposed E-SUOT approach does not adopt pseudo-labels or explicitly exploit the geometric structure of data. *We will add a clear discussion about this limitation in the revised paper.* Based on your suggestions, we will explore in future work how to integrate our framework with strategies such as leveraging geometric structure to further improve performance.
> > >
> > > **Once again, we sincerely appreciate your time and effort in reviewing our manuscript. We are grateful for your valuable suggestions and your supportive feedback.**
> > >
> > > Sincerely,
> > > Authors of Submission 6648
> > >
> > > ---
> > > Refs
> > > [1]. Variational Mixture of Normalizing Flows (2020)
> > > [2]. Gaussian Mixture Flow Matching Models (2025)

---

### Decision · Program_Chairs · 2026-04-30

**Decision:**

Accept (regular)

**Comment:**

The paper introduces a novel entropy-regularized semi-dual optimal transport framework for Gradual Domain Adaptation.
It reformulates flow-based GDA as a Lagrangian dual problem, avoiding explicit density estimation and improving stability via entropy regularization.

After the rebuttal, all reviewers agree that the paper brings a strong and novel contribution and that it deserves publications